# Nyström-Accelerated Primal LS-SVMs: Breaking the $O(an^3)$ Complexity Bottleneck for Scalable ODEs Learning

**Weikuo Wang**[1,2]**, Yue Liao**[3,4,*]**, Huan Luo**[1,2,*]

[1]College of Civil Engineering and Architecture, China Three Gorges University, Yichang, China
[2]Hubei Geological Disaster Prevention and Control Engineering Technology Research Center, Yichang, China
[3]College of Basic Medical Sciences, China Three Gorges University, Yichang, China
[4]Hubei Key Laboratory of Tumor Microenvironment and Immunotherapy, Yichang, China
{wkwang, liaoyue, hluo}@ctgu.edu.cn

## Abstract

A major problem of kernel-based methods (e.g., least squares support vector machines, LS-SVMs) for solving linear/nonlinear ordinary differential equations (ODEs) is the prohibitive $O(an^3)$ ($a = 1$ for linear ODEs and 27 for nonlinear ODEs) part of their computational complexity with increasing temporal discretization points $n$. We propose a novel Nyström-accelerated LS-SVMs framework that breaks this bottleneck by reformulating ODEs as primal-space constraints. Specifically, we derive for the first time an explicit Nyström-based mapping and its derivatives from one-dimensional temporal discretization points to a higher $m$-dimensional feature space ($1 < m \le n$), enabling the learning process to solve linear/nonlinear equation systems with $m$-dependent complexity. Numerical experiments on sixteen benchmark ODEs demonstrate: 1) $10 - 6000$ times faster computation than classical LS-SVMs and physics-informed neural networks (PINNs), 2) comparable accuracy to LS-SVMs ($< 0.13\%$ relative MAE, RMSE, and $\|y - \hat{y}\|_\infty$ difference) while maximum surpassing PINNs by 72% in RMSE, and 3) scalability to $n = 10^4$ time steps with $m = 50$ features. This work establishes a new paradigm for efficient kernel-based ODEs learning without significantly sacrificing the accuracy of the solution.

## 1 Introduction

Ordinary differential equations (ODEs) are foundational tools for modeling dynamical systems across scientific domains, including physics, engineering, and biology [1, 2]. Classical numerical methods, such as Runge-Kutta schemes and finite difference discretizations, have been the cornerstone of ODE solving due to their rigorous error analysis and convergence guarantees. However, these methods face limitations in achieving satisfactory accuracy with large time steps and incur high computational costs in long-time integration, particularly for stiff ODEs [3]. Recent advances in machine learning (ML) have introduced data-driven paradigms for solving ODEs , such as Physics-Informed Neural Networks (PINNs) [4], neural ODEs [5], and Gaussian process-based solvers [6]. These approaches demonstrate unique advantages in addressing inverse problems, enabling adaptive resolution, and directly incorporating observational data. These capabilities are often challenging for classical techniques.

Among ML-driven methods, kernel-based strategies—particularly Least Squares Support Vector Machines (LS-SVMs) [7]—have emerged as compelling alternatives for solving ODEs. By refor-

---

*Corresponding authors

39th Conference on Neural Information Processing Systems (NeurIPS 2025).

mulating the ODEs as a constrained optimization problem in a reproducing kernel Hilbert space (RKHS), LS-SVMs leverage kernel functions to implicitly capture nonlinear dynamics while ensuring regularization against overfitting. This framework has shown success in solving initial/boundary value problems [8] and parameter estimation tasks [9], benefiting from the inherent flexibility and theoretical soundness of kernel methods. However, a critical limitation persists: the computational complexity of kernel-based ODE solvers scales as $O((n+p)^3)$ when solving linear ODEs, and as $O((3n+p-2)^3)$ per Newton iteration in the case of nonlinear ODEs [8], where $n$ is the number of temporal discretization points and $p$ denotes the order of the ODE. This $O(an^3)$ ($a = 1$ for linear ODEs and 27 for nonlinear ODEs) scaling constitutes a major computational bottleneck, rendering traditional LS-SVMs impractical for long-time simulations or fine-grained temporal resolutions, and thereby undermining their utility in large-scale scientific applications.

To address this challenge, we propose a new Nyström-accelerated LS-SVMs framework that reduces the computational complexity to $O((m+p)^3)$ for linear ODEs and $O((m+p+n)^3)$ per Newton iteration for nonlinear ODEs, where $m \ll n$ represents the number of subsampled landmark points. The key innovation of our approach lies in its primal-domain formulation, which strategically integrates the Nyström method—traditionally used for low-rank kernel matrix approximations [10]—into the LS-SVMs optimization procedure. Unlike dual-domain kernel methods, our framework leverages the Nyström approximation to construct an explicit finite-dimensional feature map, enabling efficient computation of high-order derivatives essential for ODE operators. This not only preserves the theoretical benefits of kernel-based regularization but also facilitates a scalable surrogate model for high-resolution ODE systems. The main contributions include: (1) a systematic integration of the Nyström approximation within the LS-SVMs ODE solver, ensuring stability through error-controlled subspace selection; (2) a novel application of the Nyström method to obtain explicit feature mappings and their derivatives, enabling accurate representation of differential operators; and (3) comprehensive numerical validation on sixteen benchmark problems—including stiff and nonlinear systems—demonstrating order-of-magnitude speedups without sacrificing solution fidelity.

This work bridges the gap between the expressive power of kernel methods and the scalability demands of modern scientific computing. By mitigating the $O(an^3)$ complexity barrier, our method unlocks the potential for LS-SVMs to tackle large-scale ODE problems prevalent in multi-physics simulations, biological network modeling, and real-time control systems, where traditional kernel solvers were previously deemed infeasible.

## 1.1 Related Works

The development of solvers for ordinary differential equations (ODEs) navigates a trade-off between computational efficiency, theoretical robustness, and application flexibility. This section reviews the landscape of numerical, neural, and kernel-based methods, outlining the distinct challenges that motivate our work.

**Classical and Enhanced Numerical ODE Solvers**: Classical solvers, such as Runge-Kutta schemes [11] and linear multistep methods [12], form the bedrock of ODE simulation. Their principal strength lies in well-established convergence guarantees and explicit error control through adaptive time-stepping. However, their performance is intrinsically linked to the temporal discretization. Stiff ODE systems usually necessitate exceedingly small time steps to maintain stability, leading to prohibitive computational costs in long-time integration. While probabilistic enhancements [13] and Bayesian filters [14] introduce valuable uncertainty quantification, they often exacerbate computational burdens. Even ML-enhanced controllers [15, 16, 17] primarily optimize existing parameters without overcoming the fundamental constraints of discretization, leaving scalability challenges for high-resolution systems largely unresolved.

**Neural Networks for ODE Systems**: Neural network-based approaches represent a paradigm shift towards ODEs solving [18, 19]. Neural ODEs [5] offer a continuous-depth framework that adapts flexibly to irregular time series, and Physics-Informed Neural Networks (PINNs) [4] enable the seamless solution of inverse problems by incorporating physical laws as soft constraints. Despite their flexibility, these methods are prone to convergence issues due to non-convex optimization, often resulting in local minima and unpredictable training outcomes [20]. Furthermore, they can exhibit numerical instability when applied to stiff systems [21] and suffer from spectral bias [22], which impedes the accurate resolution of high-frequency solutions. Although differentiable solvers [23] improve gradient flow, they inherit the iterative cost of the underlying numerical schemes.

**Kernel Methods in ODE Solving**: Kernel methods, including formulations based on Least-Squares Support Vector Machines (LS-SVMs) [7, 8] and Gaussian processes [6, 24, 25], provide a mathematically rigorous alternative. Operating within Reproducing Kernel Hilbert Spaces (RKHS), they offer convex optimization landscapes that guarantee convergence to a global minimum, avoiding the pitfalls of non-convex training endemic to neural networks. This framework yields closed-form solutions with strong theoretical error bounds, as evidenced in kernel collocation techniques [26]. The primary limitation, however, is their computational scalability: solving the resulting dense linear system requires matrix inversion with $O(an^3)$ complexity, which becomes prohibitive for large-scale or long-time ODE simulations.

**Scalability in Kernel-Based ODE Solvers**: Accelerating kernel methods has been a central focus in supervised learning, where techniques like the Nyström method [10, 27, 28] and its advanced variants [29, 30] successfully reduce complexity to $O(m^2 n)$ for $m \ll n$ landmark points via low-rank matrix approximations. However, these innovations have seen limited translation to the ODE context. Exceptions, such as reduced-rank Kalman filters [31], exploit low-rank structures but remain disconnected from kernelized optimization frameworks like LS-SVMs. Consequently, a significant gap persists: no prior work has systematically reduced the $O(an^3)$ bottleneck of LS-SVMs for ODE solving while preserving their regularization benefits. Our method addresses this by integrating Nyström approximation directly into the LS-SVMs optimization process.

## 1.2 Novelty and Contributions

Our work bridges the gap between kernelized ODE solvers' theoretical strengths and their practical scalability constraints. In contrast to classical Nyström methods—which are typically applied in the dual formulation of kernel-based classification and regression problems using the kernel trick [10]—our method operates entirely within the primal formulation. This provides an explicit representation of the nonlinear feature mapping and its high-order derivatives, which is specifically tailored to the temporal structure of ODEs and ensures stability in stiff systems (Figure 1). This aligns with trends in data-driven scientific computing but specifically targets the under-explored challenge of $O(an^3)$ complexity in kernel-based ODE solvers. Our framework could be extended to large-scale problems in systems biology, control theory, and multi-agent dynamics, where traditional kernel-based ODEs solvers were previously impractical.

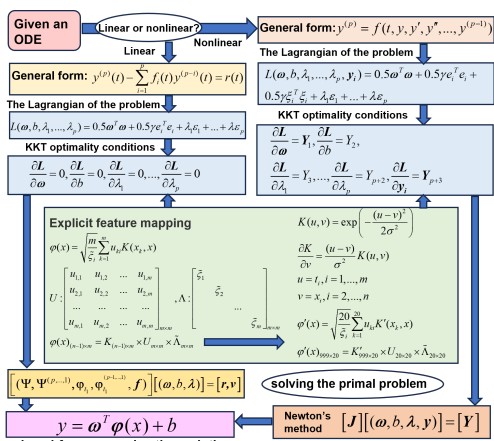

Figure 1: Flowchart of Nyström-accelerated LS-SVMs for Efficient ODE Learning.

## 2 Preliminaries

This section describes the problem statement, a short introduction to LS-SVMs (Appendix A.1) for ODEs solving is given to highlight the prohibitive $O(an^3)$ complexity problem considered in this paper. Consider a general $p$-th order linear/nonlinear ODE with the following form:

$$\mathcal{L}[y] \equiv y^{(p)} = f(t, y, y', \cdots, y^{(p-1)}), \quad t \in [t_1, t_n] \tag{1}$$

where $\mathcal{L}$ represents an $p$-th order linear/nonlinear differential operator depending on the linear/nonlinear function $f(t, y, y', \cdots, y^{(p-1)})$, $[t_1, t_n]$ is the problem domain, $t$ is the input signal, and $y^{(\ell)}(t)$ denotes the $\ell$-th derivative of $y$ with respect to $t$, $\ell \in [0, p]$. The $p$ necessary initial or boundary conditions for solving the above ODE are

$$\text{IVP: } \mathcal{IV}_\mu[y] = v_\mu, \text{BVP: } \mathcal{BV}_\mu[y] = q_\mu, \quad \mu = 0, \ldots, p-1 \tag{2}$$

where $\mathcal{IV}_\mu$ are the initial conditions (all constraints are applied at the initial value of the independent variable i.e., $t = t_1$) and $\mathcal{BV}_\mu$ are the boundary conditions (the constraints are applied at multiple

values of the independent variable $t$, typically at the ends of the interval $[t_1, t_n]$ in which the solution is sought). $v_\mu$ and $q_\mu$ are given scalars.

When the function $f(t, y, y', \cdots, y^{(p-1)})$ in Eq.(1) is linear, the $p$-th order ODE to be solved is linear. The optimization problem based on LS-SVMs for learning the $p$-th order linear ODE can be formulated by the method in Ref [8]. Its solution requires solving a system of $(n+p)$ linear equations with $O((n+p)^3)$ computational complexity. When the function is nonlinear, the $p$-th order ODE is nonlinear and written in the following form (initial value problems for illustration):

$$y^{(p)} = f(t, y, y', \cdots, y^{(p-1)}), \quad y(t_1) = v_0, \cdots, y^{(p-1)}(t_1) = v_{p-1}, \quad t_1 \leq t \leq t_n \quad (3)$$

Assume the approximate solution: $\hat{y}(t) = \boldsymbol{\omega}^T \boldsymbol{\varphi}(t) + b$. Additional unknowns $y_i$ are introduced to keep the constraints linear in $\boldsymbol{\omega}$. This yields the following nonlinear optimization problem:

$$\underset{\boldsymbol{\omega}, b, e_i, \xi_i, y_i}{\text{minimize}} \quad \frac{1}{2}\boldsymbol{\omega}^T\boldsymbol{\omega} + \frac{\gamma}{2}\sum_{i=2}^n e_i^2 + \frac{\gamma}{2}\sum_{i=2}^n \xi_i^2$$

$$\text{s.t. } y^p = \boldsymbol{w}^T \boldsymbol{\varphi}^{(p)}(t_i) = f\left(t_i, y_{t_i}, y'_{t_i}, \cdots, y^{(p-1)}_{t_i}\right) + e_i, \quad i = 2, \ldots, n \quad (4)$$

$$y(t_1) = \boldsymbol{w}^T \boldsymbol{\varphi}(t_1) + b = v_1, y^{(\ell-1)}(t_1) = \boldsymbol{\omega}^T \boldsymbol{\varphi}^{(\ell-1)}(t_1) = v_\ell, \quad \ell = 2, \ldots, p$$

$$y_i = \boldsymbol{w}^T \boldsymbol{\varphi}(t_i) + b + \xi_i, \quad i = 2, \ldots, n.$$

By using classical kernel trick, Eq.(4) can be effectively solved. For detailed mathematical derivation, see Appendix A.2. The nonlinear system, which consists of $(3n + p - 2)$ equations with $(3n + p - 2)$ unknowns, is solved by Newton's method [32]. For each Newton iteration, solving for unknowns requires the computational complexity of $O((3n + p - 2)^3)$. In cases of a long-time interval with more than $10^4$ time steps, the prohibitive $O(27n^3)$ part of the computational complexity causes the expensive computations or even may stop the run.

## 3   Proposed Nyström-accelerated LS-SVMs (NLS-SVMs)

We consider the explicit model, $\hat{y}(t) = \boldsymbol{\omega}^T \boldsymbol{\varphi}(t) + b$, as a closed-form approximation solution to the ODE (i.e., Eq.(1)). A key innovation of the NLS-SVMs approach is its application of the Nyström method to directly approximate the high-dimensional nonlinear feature map $\boldsymbol{\varphi}(t)$ and its higher-order derivatives, enabling the direct substitution of the approximate solution $\hat{y}(t)$ and its derivatives into the governing equation. Therefore, the NLS-SVMs method bypasses the kernel trick, allowing the model parameters to be efficiently estimated in primal form, with computational complexity dependent only on the dimension of the approximated features. Leveraging Mercer's theorem, the derivatives of the feature map $\boldsymbol{\varphi}(t)$ can be analytically expressed using derivatives of the kernel function. To this end, we define the following differential operator:

$$[\nabla_c^a K](t, s) = [\nabla_c^a K(u, v)]|_{u=t, v=s}, [\Omega_c^a]_{i,j} = \nabla_c^a [K(u, v)]|_{u=t_i, v=t_j} = \frac{\partial^{c+a} K(u,v)}{\partial u^c \partial v^a}\Big|_{u=t_i, v=t_j} \quad (5)$$

where $[\Omega_c^a]_{i,j}$ denotes the $(i, j)$-th entry of matrix $\Omega_c^a$. Given a long-time interval $\{t_i\}_{i=1}^n$, the Nyström method, as defined in Eq.(34) (Appendix A.3), is employed to construct an explicit finite-dimensional feature mapping $\boldsymbol{\varphi}(t) \in R^m$ with $(m \ll n)$. This allows the model parameters $\boldsymbol{\omega} \in R^m$ and $b \in R$ to be efficiently estimated in the primal space, circumventing the computational burden associated with the dual formulation. Throughout this paper, we adopt the radial basis function (RBF) kernel $K(u, v) = exp(-(u - v)^2/\sigma^2)$. Using the differential relations established in Eq.(5), the first- and second-order derivatives of the kernel function are given by:

$$\nabla_1^0[K(u, v)] = -\frac{2(u - v)}{\sigma^2} K(u, v), \nabla_2^0[K(u, v)] = \left[\frac{4(u - v)^2}{\sigma^4} - \frac{2}{\sigma^2}\right] K(u, v) \quad (6)$$

By extending Eq.(34), the $p$-th order derivative of the feature map $\boldsymbol{\varphi}(t)$ can be explicitly written as:

$$\varphi_k^{(p)}(t_i) = \sqrt{\hat{\varsigma}_k}\phi_k^{(p)}(t_i) = \frac{\sqrt{m}}{\sqrt{\hat{\varsigma}_k}}\sum_{k=1}^m \hat{\Phi}_{sk} K^{(p)}(t_k, t_i) \quad (7)$$

These explicit expressions for the feature map and its derivatives enable the direct numerical treatment of $p$-th order linear and nonlinear ODEs introduced in Ref [8] and Eq.(4), entirely within the primal framework. As a result, the dominant computational complexity is substantially reduced from $O(an^3)$ to $O(m^3)$ for linear ODEs and $O(n^3)$ for nonlinear ODEs, where $m \ll n$.

## 3.1 NLS-SVMs for learning $p$-th Order Linear ODE

Consider the general $p$-th order linear ODE with initial value problem (IVP) in [8]:

$$y^{(p)}(t)-\sum_{k=1}^{p} f_k(t)y^{(p-k)}(t)=r(t), \quad t\in[t_1,t_n], \quad y(t_1)=v_1, \quad y^{(k-1)}(t_1)=v_k, \quad k=2,\ldots,p$$

$$(8)$$

The approximate solution can be obtained by solving the following optimization problem:

$$\underset{\boldsymbol{\omega},b,e_i}{\text{minimize}} \quad \frac{1}{2}\boldsymbol{\omega}^T\boldsymbol{\omega}+\frac{\gamma}{2}\sum_{i=1}^{n}e_i^2 \tag{9}$$

$$\text{s.t.} \quad y^{(p)}(t_i)=\boldsymbol{\omega}^T\boldsymbol{\varphi}^{(p)}(t_i)=\boldsymbol{\omega}^T\left[\sum_{k=1}^{p}f_k(t_i)\boldsymbol{\varphi}_i^{(p-k)}\right]+f_p(t_i)b+r(t_i)+e_i,, \quad i=2,\ldots,n$$

$$y(t_1)=\boldsymbol{\omega}^T\boldsymbol{\varphi}(t_1)+b=v_1, y^{(\ell-1)}(t_1)=\boldsymbol{\omega}^T\boldsymbol{\varphi}^{(\ell-1)}(t_1)=v_i, \quad \ell=2,\ldots,p$$

**Lemma 1**: Given a differentiable positive definite kernel function $K:\mathbb{R}\times\mathbb{R}\to\mathbb{R}$, $m$ landmark points subsampled from a dataset of $n$ time points, where $m\ll n$, and a regularization parameter $\gamma\in\mathbb{R}^+$, the solution to optimization problem Eq.(9) is obtained by solving the following primal problem.

$$@\begin{bmatrix}\begin{bmatrix} \boldsymbol{E}+\mathbf{A}\left(\boldsymbol{\Psi}^{(p)}-\boldsymbol{f_1}\boldsymbol{\Psi}^{(p-1)}-...-\boldsymbol{f_p}\boldsymbol{\Psi}\right) & -\mathbf{A}\left(\boldsymbol{f_p}\cdot\mathbf{1}\right) & \boldsymbol{\varphi}^T(t_1) & ... & (\boldsymbol{\varphi}^{(p-1)}(t_1))^T \\ \boldsymbol{V}\left(\boldsymbol{\Psi}^{(p)}-\boldsymbol{f_1}\boldsymbol{\Psi}^{(p-1)}-...-\boldsymbol{f_p}\boldsymbol{\Psi}\right) & -\boldsymbol{V}\left(\boldsymbol{f_p}\cdot\mathbf{1}\right) & 1 & ... & 0 \\ \boldsymbol{\varphi}(t_1) & 1 & 0 & ... & 0 \\ ... & ... & ... & ... & ... \\ \boldsymbol{\varphi}^{(p-1)}(t_1) & 0 & 0 & ... & 0 \end{bmatrix}\end{bmatrix}$$

$$@\begin{bmatrix} \boldsymbol{\omega} \\ b \\ \lambda_1 \\ ... \\ \lambda_p \end{bmatrix}=\begin{bmatrix} \mathbf{A}\boldsymbol{r} \\ \boldsymbol{V}\boldsymbol{r} \\ v_1 \\ ... \\ v_p \end{bmatrix} \tag{10}$$

where $\boldsymbol{E}\in\mathbb{R}^{m\times m}$ is an identity matrix; $\boldsymbol{\Psi}=[\varphi(t_2),...,\varphi(t_n)]^T\in\mathbb{R}^{(n-1)\times m}$; $\boldsymbol{\Psi}'=[\varphi'(t_2),...,\varphi'(t_n)]^T\in\mathbb{R}^{(n-1)\times m}$; $\boldsymbol{\Psi}^{(p)}=[\varphi^{(p)}(t_2),...,\varphi^{(p)}(t_n)]^T\in\mathbb{R}^{(n-1)\times m}$; $\boldsymbol{f_1}=[f_1(t_2),...,f_1(t_n)]^T\in\mathbb{R}^{n-1}$; $\boldsymbol{f_p}=[f_p(t_2),...,f_p(t_n)]^T\in\mathbb{R}^{n-1}$; $\mathbf{1}=[1,\ldots,1]^T\in\mathbb{R}^{n-1}$; $\boldsymbol{\varphi}(t_1)\in\mathbb{R}^{1\times m}$; $\boldsymbol{\varphi}^{(p)}(t_1)\in\mathbb{R}^{1\times m}$; $\boldsymbol{A}=\gamma[(\boldsymbol{\Psi}^{(p)}-\boldsymbol{f_1}\boldsymbol{\Psi}^{(p-1)}-...-\boldsymbol{f_p}\boldsymbol{\Psi}]^T\in\mathbb{R}^{m\times(n-1)}$; $\boldsymbol{V}=\gamma[-\boldsymbol{f_p}\mathbf{1}]^T\in\mathbb{R}^{1\times(n-1)}$; $\boldsymbol{r}=[r(t_2),...,r(t_n)]^T\in\mathbb{R}^{n-1}$; $\boldsymbol{\omega}\in\mathbb{R}^{m\times 1}$. See Appendix A.4 for proof.

## 3.2 NLS-SVMs for Learning $p$-th Order Nonlinear ODE

We establish a new paradigm based on NLS-SVMs for solving Eq.(4).

**Lemma 2**: Given a differentiable positive definite kernel function $K:\mathbb{R}\times\mathbb{R}\to\mathbb{R}$, $m$ landmark points subsampled from a dataset of $n$ time points, where $m\ll n$, and a regularization parameter $\gamma\in\mathbb{R}^+$, the solution to optimization problem Eq.(4) is given by solving the following primal problem.

$$\begin{bmatrix} \frac{\partial\boldsymbol{Y_1}}{\partial\boldsymbol{\omega}} & \frac{\partial\boldsymbol{Y_1}}{\partial b} & \frac{\partial\boldsymbol{Y_1}}{\partial\lambda_1} & \cdots & \frac{\partial\boldsymbol{Y_1}}{\partial\boldsymbol{y_i}} \\ \frac{\partial Y_2}{\partial\boldsymbol{\omega}} & \frac{\partial Y_2}{\partial b} & \frac{\partial Y_2}{\partial\lambda_1} & \cdots & \frac{\partial Y_2}{\partial\boldsymbol{y_i}} \\ \frac{\partial Y_3}{\partial\boldsymbol{\omega}} & \frac{\partial Y_3}{\partial b} & \frac{\partial Y_3}{\partial\lambda_1} & \cdots & \frac{\partial Y_3}{\partial\boldsymbol{y_i}} \\ ... & ... & ... & ... & ... \\ \frac{\partial\boldsymbol{Y_{p+3}}}{\partial\boldsymbol{\omega}} & \frac{\partial\boldsymbol{Y_{p+3}}}{\partial b} & \frac{\partial\boldsymbol{Y_{p+3}}}{\partial\lambda_1} & \cdots & \frac{\partial\boldsymbol{Y_{p+3}}}{\partial\boldsymbol{y_i}} \end{bmatrix}\begin{bmatrix} \boldsymbol{\omega} \\ b \\ \lambda_1 \\ ... \\ \lambda_p \\ \boldsymbol{y_i} \end{bmatrix}=\begin{bmatrix} \boldsymbol{Y_1} \\ Y_2 \\ Y_3 \\ ... \\ Y_{p+2} \\ \boldsymbol{Y_{p+3}} \end{bmatrix} \tag{11}$$

where $\boldsymbol{Y_1}\in\mathbb{R}^{m\times 1}$; $Y_2\in\mathbb{R}^{1\times 1}$; $Y_3\in\mathbb{R}^{1\times 1}$;...;$Y_{p+2}\in\mathbb{R}^{1\times 1}$; $\boldsymbol{Y_{p+3}}\in\mathbb{R}^{(n-1)\times 1}$. The non-linear system, which consists of equations with unknowns $(\boldsymbol{\omega},b,\lambda_1,...,\lambda_p,\boldsymbol{y_i})\in\mathbb{R}^{(m+n+p)}$, is solved by Newton's method. While more efficient iterative schemes ([33, 34, 35]) exist, their exploration falls outside the scope of this paper. It is worth noting that when learning a class of $p$-th order nonlinear ODEs, the changes in the ODEs primarily affect the $\frac{\gamma}{2}\left(\boldsymbol{\omega}^T\boldsymbol{\varphi}^{(p)}(t_i)-f(t_i,y_{t_i},y'_{t_i},\cdots,y_{t_i}^{(p-1)})\right)^T\left(\boldsymbol{\omega}^T\boldsymbol{\varphi}^{(p)}(t_i)-f(t_i,y_{t_i},y'_{t_i},\cdots,y_{t_i}^{(p-1)})\right)$ function, as indicated by the Lagrangian loss function. Therefore, the Jacobian matrix needs to be updated based on the derivation of the four functions $\frac{\partial\boldsymbol{Y_1}}{\partial\boldsymbol{\omega}}$, $\frac{\partial\boldsymbol{Y_1}}{\partial\boldsymbol{y_i}}$, $\frac{\partial\boldsymbol{Y_{p+3}}}{\partial\boldsymbol{\omega}}$ and $\frac{\partial\boldsymbol{Y_{p+3}}}{\partial\boldsymbol{y_i}}$. See Appendix A.5 for proof.

**Convergence analysis**: Since the tunable parameters (i.e., regularization parameter $\gamma$ and kernel parameter $\sigma^2$) in the nonlinear system Eq.(11) influence the structure of the Jacobian matrix, we theoretically analyze their impact on the convergence behavior of the Newton solver.

*Theorem*: Consider the system of nonlinear equations in Eq.(11) represented by $F(\boldsymbol{x}; \sigma^2, \gamma) = 0$, where $F : \mathbb{R}^N \to \mathbb{R}^N (N = m + n + p)$ depends on a kernel parameter $\sigma^2$ and a regularization parameter $\gamma$. Let $J_F(\boldsymbol{x}; \sigma^2, \gamma)$ denote its Jacobian matrix. Assume a solution $\boldsymbol{x}^*$ exists such that $F(\boldsymbol{x}^*; \sigma^2, \gamma) = 0$. The Newton iteration is given by:

$$\boldsymbol{x}_{k+1} = \boldsymbol{x}_k - [J_F(\boldsymbol{x}_k; \sigma^2, \gamma)]^{-1} F(\boldsymbol{x}_k; \sigma^2, \gamma) \tag{12}$$

Suppose the following conditions hold in a ball $B(\boldsymbol{x}^*, \delta)$ of radius $\delta > 0$ around $\boldsymbol{x}^*$:

1. *Lipschitz continuity of the Jacobian*: There exists a constant $L(\sigma^2, \gamma) > 0$ such that

$$\|J_F(\boldsymbol{x}; \sigma^2, \gamma) - J_F(\boldsymbol{y}; \sigma^2, \gamma)\| \leq L(\sigma^2, \gamma)\|\boldsymbol{x} - \boldsymbol{y}\|, \quad \forall \boldsymbol{x}, \boldsymbol{y} \in B(\boldsymbol{x}^*, \delta) \tag{13}$$

2. *Bounded inverse of the Jacobian*: The Jacobian matrix $J_F(\boldsymbol{x}; \sigma^2, \gamma)$ is non-singular for all $\boldsymbol{x} \in B(\boldsymbol{x}^*, \delta)$, and there exists a constant $\beta(\sigma^2, \gamma) > 0$ such that

$$\|[J_F(\boldsymbol{x}; \sigma^2, \gamma)]^{-1}\| \leq \beta(\sigma^2, \gamma), \quad \forall \boldsymbol{x} \in B(\boldsymbol{x}^*, \delta) \tag{14}$$

Then, for any initial guess $\boldsymbol{x}_0 \in B(\boldsymbol{x}^*, \delta)$ satisfying

$$\|\boldsymbol{x}_0 - \boldsymbol{x}^*\| < \min\left(\delta, \frac{1}{K}\right), \quad \text{where} \quad K = \frac{1}{2}\beta(\sigma^2, \gamma)L(\sigma^2, \gamma) \tag{15}$$

the iteration converges quadratically to $\boldsymbol{x}^*$, with the error $\boldsymbol{e}_k = \boldsymbol{x}_k - \boldsymbol{x}^*$ satisfying:

$$\|\boldsymbol{e}_{k+1}\| \leq K\|\boldsymbol{e}_k\|^2 \tag{16}$$

*Proof*: According to Eq.(12), the error at the next step $\boldsymbol{e}_{k+1} = \boldsymbol{x}_{k+1} - \boldsymbol{x}^*$ can be expressed as (parameters $\gamma$ and $\sigma^2$ are omitted for clarity):

$$\boldsymbol{e}_{k+1} = \boldsymbol{e}_k - [J_F(\boldsymbol{x}_k)]^{-1}F(\boldsymbol{x}_k) = [J_F(\boldsymbol{x}_k)]^{-1}(J_F(\boldsymbol{x}_k)\boldsymbol{e}_k - (F(\boldsymbol{x}_k) - F(\boldsymbol{x}^*))) \tag{17}$$

The term $F(\boldsymbol{x}_k) - F(\boldsymbol{x}^*)$ is estimated using Taylor expansion with $F(\boldsymbol{x}^*) = 0$:

$$F(\boldsymbol{x}_k) - F(\boldsymbol{x}^*) = \int_0^1 J_F(\boldsymbol{x}^* + t\boldsymbol{e}_k)\boldsymbol{e}_k \, dt \tag{18}$$

Substituting this back yields:

$$\boldsymbol{e}_{k+1} = [J_F(\boldsymbol{x}_k)]^{-1}\left(\int_0^1 [J_F(\boldsymbol{x}_k) - J_F(\boldsymbol{x}^* + t\boldsymbol{e}_k)]\boldsymbol{e}_k \, dt\right) \tag{19}$$

Taking norms and applying the Lipschitz condition (Eq.(13)) to the integrand gives:

$$\|J_F(\boldsymbol{x}_k) - J_F(\boldsymbol{x}^* + t\boldsymbol{e}_k)\| \leq L\|\boldsymbol{e}_k - t\boldsymbol{e}_k\| = L(1-t)\|\boldsymbol{e}_k\| \tag{20}$$

The integral $\int_0^1 (1-t)dt = 1/2$. Then, applying the bound on the inverse of the Jacobian (Eq.(14)), $\|[J_F(\boldsymbol{x}_k)]^{-1}\| \leq \beta$, we obtain the final result:

$$\|\boldsymbol{e}_{k+1}\| \leq \beta \cdot \frac{L}{2}\|\boldsymbol{e}_k\|^2 = K\|\boldsymbol{e}_k\|^2 \tag{21}$$

*Implications*: The convergence rate is governed by the product $\beta(\sigma^2, \gamma)L(\sigma^2, \gamma)$, which is jointly influenced by the parameters $\gamma$ and $\sigma^2$. Here, the regularization parameter $\gamma$ ensures the non-singularity of the Jacobian, while the kernel parameter $\sigma^2$ controls the smoothness of $F$. A larger $\sigma^2$ generally reduces the Lipschitz constant $L$, thereby promoting convergence at the potential cost of oversmoothing. Consequently, a careful balance between $\sigma^2$ and $\gamma$ is critical to achieving an optimal trade-off among convergence speed, solution accuracy, and numerical stability.

**Computational complexity reduction**: The computational complexity of the proposed NLS-SVMs method, as indicated in Eq.(10) and Eq.(11), is substantially reduced compared to traditional LS-SVMs ODE solvers. It scales with the dimension $m$ ($m \ll n$) of the high-dimensional feature $\boldsymbol{\varphi}(t_i)$. For linear ODEs, it decreases from $O((n+p)^3)$ to $O((m+p)^3)$, while for nonlinear ODEs, the per-Newton-iteration complexity is lowered from $O((3n+p-2)^3)$ to $O((m+p+n)^3)$. Empirical analysis demonstrates that the proposed approach significantly enhances computational efficiency while preserving prediction accuracy. Moreover, the memory requirement for linear ODEs decreases from $O((n+p)^2)$ to $O((m+p)^2)$ and for nonlinear ODEs decreases from $O((3n+p-2)^2)$ to $O((m+n+p)^2)$. Appendix A.6 covers the initial value problem (IVP) for the first-order linear ODE, Appendix A.7 presents the IVP and boundary value problem (BVP) for the second-order linear ODE, and Appendix A.8 provides the derivations for the first-order nonlinear ODE with IVP.

## 4 Numerical experiments

This section evaluates the performance of the proposed method using sixteen benchmark ODE problems (see Appendix B.1). These include six first-order, nine second-order, and one fourth-order ODE, covering a wide range of common types such as stiff, linear, nonlinear, and singular ODEs, as well as ODEs with time-varying input signals, undamped free vibration, and higher-order dynamics. Prior to the performance evaluation, we conducted a preliminary analysis involving kernel function selection and sampling strategies across all benchmark problems. Convergence behavior was further examined using Problem 4, a representative nonlinear ODE. Additionally, a comprehensive comparative study will be performed against suitable baseline models to thoroughly assess the efficacy of the proposed approach. All numerical experiments in this study were conducted on a computer system equipped with an Intel(R) Core(TM) i9-14900HX processor, 32 GB of RAM. Source code is available at: `https://github.com/AI4SciCompLab/NLS-SVMs`.

**RBF kernel justification**: The selection of the kernel function plays a critical role in the proposed methodology, as it directly governs the model's capacity to capture nonlinear patterns. To illustrate this, we examine Problem 1 as a representative case. As shown in Figure 2, the radial basis function (RBF) kernel projects samples into an infinite-dimensional feature space, offering universal approximation capabilities and robustness to variations in data scale. In comparison to linear and polynomial kernels, the RBF kernel demonstrates superior stability and accuracy when applied to complex nonlinear systems (see results for Problems 4, 5, 6, 14 and 15 in Table 4). Consequently, the RBF kernel is adopted in all subsequent experiments. Both the NLS-SVMs and the classical LS-SVMs are tuned via two hyperparameters: the kernel

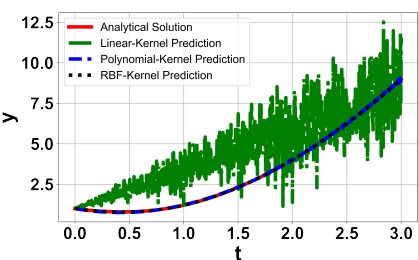

Figure 2: Numerical results for Problem 1 using the Linear, Polynomial and RBF kernel within the NLS-SVMs framework.

bandwidth $\sigma^2$ and regularization coefficient $\gamma$. A comprehensive quantitative analysis of kernel performance across all benchmark problems is provided in Appendix B.3 (Table 4).

**Sampling strategy analysis for Nyström landmarks**: To assess the influence of sampling strategy on the performance of the proposed NLS-SVMs method, we evaluate three distinct approaches that are equidistant sampling, ran-

Table 1: Performance comparison of NLS-SVMs with different sampling strategies on Problem 1

| ODEs | Model | MAE | RMSE | $\|y - \hat{y}\|_\infty$ | Time/s |
|---|---|---|---|---|---|
| $P\,1$ | Random | $8.70{\times}10^{-4}$ | $1.06{\times}10^{-3}$ | $3.06{\times}10^{-3}$ | 0.52 |
| | Leverage score | $7.94{\times}10^{-4}$ | $9.47{\times}10^{-4}$ | $1.87{\times}10^{-3}$ | 0.64 |
| | Equidistant | $7.94{\times}10^{-4}$ | $9.47{\times}10^{-4}$ | $1.87{\times}10^{-3}$ | 0.55 |

dom sampling, and leverage score sampling. The analysis for Problem 1, summarized in Table 1, reveals that all three sampling methods achieve a comparable level of prediction accuracy. However, equidistant sampling demonstrates notable advantages in computational efficiency and error reduction. Specifically, it improves computational speed by approximately 16% compared to leverage score sampling and reduces the RMSE by about 10.66% relative to simple random sampling. Comprehensive numerical results evaluating these sampling strategies across all benchmark problems are available in Appendix B.3 (see Table 5). These results demonstrate that the equidistant sampling strategy delivers robust and superior performance. It either matches or surpasses the other methods in the majority of test cases, particularly in challenging scenarios such as stiff systems (e.g., Problem 3), second-order ODEs (e.g., Problems 8, 9, 10, and 12), and problems with singularities (e.g., Problem 13). Furthermore, equidistant sampling exhibits exceptional stability, maintaining consistently low error levels across the entire test cases. While the computational efficiency is generally comparable across all methods, the equidistant strategy frequently attains superior accuracy without incurring a substantial computational penalty. In many cases, its runtime is nearly identical to that of the fastest alternative. These collective results indicate that the equidistant sampling strategy provides an optimal balance between numerical precision and computational expense.

**Convergence analysis and numerical validation**: Building upon the theoretical convergence analysis presented in Section 3.2, which established that the convergence of the Newton-type solver

is governed by the proper selection of the regularization parameter $\gamma$ and RBF kernel parameter $\sigma^2$, this section provides an empirical validation of these theoretical findings. The analysis demonstrated that local quadratic convergence is achieved when these parameters are chosen appropriately. The numerical experiments confirm the theoretical predictions. For the representative nonlinear system (Problem 4), the solver exhibits stable residual convergence, typically within 10 to 50 iterations, as depicted in Figure 3). This robust convergence behavior is consistently observed across a wide range of tested problems, with detailed results provided in Appendix B.4 (Table 8). The empirical evidence strongly corroborates the theoretical reliability of the solver for the considered class of problems. Furthermore, the numerical results validate the theoretically indicated parameter configurations. Robust and efficient convergence is achieved with $\gamma = 10^{(6/7)}$ for all sixteen benchmark problems. The kernel parameter is effectively set to $\sigma^2 = 1/8/10$ for first-order ODEs, and $\sigma^2 = 1$ for second-order and higher-order ODEs. See Appendix B.5 for more details. These results collectively demonstrate that the proposed solver delivers numerically stable and efficient convergence across diverse problem types, firmly aligning with the established theoretical framework.

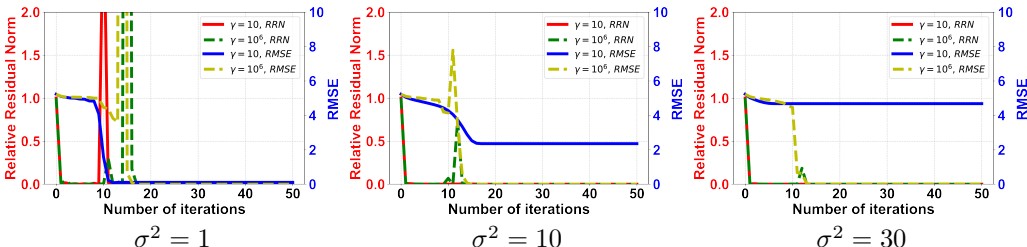

Figure 3: Numerical results of convergence analysis for Problem 4

**Comparative analysis with classical numerical methods: RK4 and EAB**: To validate the effectiveness of the proposed NLS-SVM method, we conducted a comparative study against two classical numerical techniques: the fourth-order Runge–Kutta method (RK4) and the Explicit Adams–Bashforth (EAB) method. The evaluation was performed on three representative test cases: Problem 3 (stiff ODE), Problem 6 (nonlinear ODE), and Problem 11 (singular ODE). The results are summarized in Table 2. On Problem 3, NLS-SVMs achieved an MAE of

Table 2: Performance comparison for solving Problems 3, 6, and 11 using NLS-SVMs, RK4, and EAB

| ODEs | Model | MAE | RMSE | Train/s | Predict/s |
|------|-------|-----|------|---------|-----------|
| **P 3** | NLS-SVMs | $2.06 \times 10^{-8}$ | $2.77 \times 10^{-8}$ | 0.460 | 0.001 |
| | RK4 | $3.87 \times 10^{-7}$ | $3.41 \times 10^{-7}$ | 0.880 | 0.880 |
| | EAB | $4.97 \times 10^{-8}$ | $2.90 \times 10^{-8}$ | 0.030 | 0.030 |
| **P 6** | NLS-SVMs | $8.89 \times 10^{-4}$ | $1.01 \times 10^{-3}$ | 1.230 | 0.001 |
| | RK4 | $1.59 \times 10^{-2}$ | $1.21 \times 10^{-2}$ | 0.021 | 0.021 |
| | EAB | $1.28 \times 10^{-6}$ | $1.11 \times 10^{-6}$ | 0.036 | 0.036 |
| **P 11** | NLS-SVMs | $2.36 \times 10^{-9}$ | $2.97 \times 10^{-9}$ | 0.100 | 0.001 |
| | RK4 | $3.11 \times 10^{-6}$ | $2.87 \times 10^{-6}$ | 0.010 | 0.010 |
| | EAB | $1.19 \times 10^{-2}$ | $6.86 \times 10^{-3}$ | 0.020 | 0.020 |

$2.06 \times 10^{-8}$, which is 18.7 times lower than that of RK4 ($3.87 \times 10^{-7}$) and 2.4 times lower than that of EAB ($4.97 \times 10^{-8}$), while being 880 and 30 times faster in prediction time, respectively. For Problem 6, the proposed method delivered an MAE of $8.89 \times 10^{-4}$, reflecting a 17.9-fold improvement over RK4 ($1.59 \times 10^{-2}$), along with a 21-fold speedup (0.001 s vs. 0.021 s) in predictions. In the case of the singular ODE (Problem 11), NLS-SVMs maintained high accuracy (MAE=$2.36 \times 10^{-9}$), whereas EAB failed with an error nearly 5 million times larger (MAE=$1.19 \times 10^{-2}$), establishing the proposed approach as a high-accuracy solver. Additional numerical results are provided in Appendix B.3 (Table 6). It is important to note that the proposed method is not universally superior but offers practical advantages in specific scenarios: (1) it rescues simulations where EAB may fail on singular systems; (2) it enables real-time control applications to stiff systems where RK4 may be computationally prohibitive; (3) it maintains reliability at practical step sizes where other methods require extreme refinement; and (4) it preserves accuracy across a range of step sizes, unlike the step-sensitive errors observed with RK4 and EAB.

**Physics-Informed Neural Network Baseline Selection**: To establish a rigorous baseline for comparison with the proposed NLS-SVMs method, a systematic model selection procedure was conducted for Physics-Informed Neural Networks (PINNs). The selection process comprised two main stages:

first, an architectural optimization within the Vanilla PINN framework, followed by a comparative evaluation against Fourier Feature PINN to determine the optimal PINN variant. *Vanilla PINN Architecture Selection*: An ablation study was performed to identify the most effective network architecture for Vanilla PINN. We compared a standard three-layer configuration against an eight-layer architecture using Problem 1 as a benchmark case. The results indicate that the deeper architecture (MAE $= 2.32 \times 10^{-3}$, RMSE $= 2.93 \times 10^{-3}$, Time $= 1275$ s ) provides no significant accuracy improvement over the three-layer counterpart (MAE $= 4.45 \times 10^{-3}$, RMSE $= 5.05 \times 10^{-3}$, Time $= 672$ s ), while incurring a substantial computational overhead (1275 s vs. 672 s). Consequently, the three-layer architecture was selected as it offers the optimal balance between accuracy and efficiency for the class of problems under consideration. This model utilizes identity activation functions, a learning rate of 0.001, and the Adam optimizer, with gradients computed via PyTorch's automatic differentiation. *Comparative Evaluation with Fourier Feature PINN*: The optimized three-layer Vanilla PINN was then evaluated against Fourier Feature PINN across all 16 benchmark ODEs. Fourier Feature PINN demonstrates superior accuracy in the vast majority of cases. For instance, in Problems 3, 11, and 15, it achieves dramatically lower errors (often by several orders of magnitude) in MAE, RMSE, and $L_\infty$ norm compared to the Vanilla PINN. Notably, it can also lead to substantially faster training times, as seen in Problems 11 and 15. However, this advantage is not universal. The Vanilla PINN outperforms its counterpart in specific cases like Problems 4, 7, and most notably Problem 16, where the Fourier Feature version fails completely with exceedingly high errors. Furthermore, the Fourier Feature PINN often requires longer computational time, though this is not always the case. Detailed comparison of numerical results are provided in Appendix B.3 (Table 7). Based on this comprehensive analysis, the three-layer Vanilla PINN is selected as the representative PINN baseline for subsequent comparisons with the proposed NLS-SVM method. The chosen model provides a meaningful benchmark for evaluating the relative performance of the proposed method, while Fourier Feature PINN remains recommended for more complex multi-physics or high-gradient scenarios. All subsequent comparisons with PINNs in this study refer to this optimized three-layer Vanilla PINN configuration.

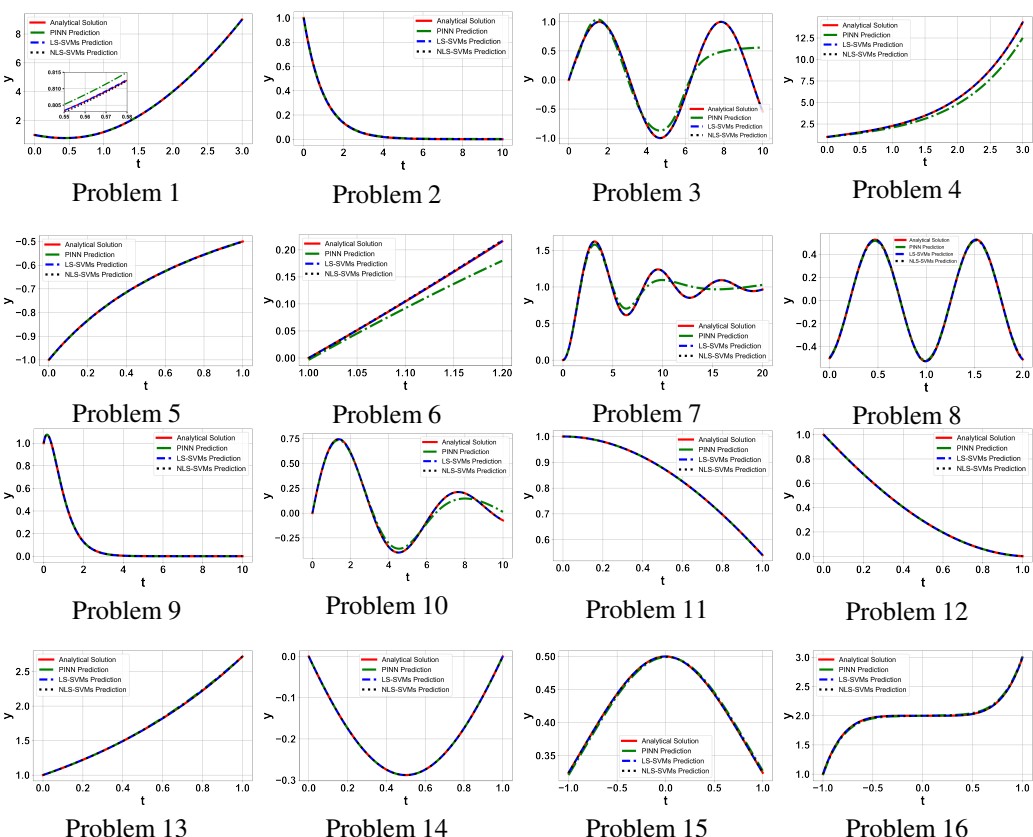

Figure 4: Comparison of model predictions against the analytical solution for Problems 1 to 16

# 5 Results and discussions

Figure 4 presents the comparision of solutions obtained by LS-SVMs, PINNs, and proposed NLS-SVMs for all 16 benchmark ODEs, with corresponding analytical solutions serving as ground truth. By observation, the proposed NLS-SVMs demonstrate significant improvements in computational efficiency while maintaining competitive solution accuracy relative to classical LS-SVMs. Table 3 quantifies the significant computational speedups (10–6000 times) under comparable accuracy regimes for key benchmarks. Across all test cases, NLS-SVMs achieved comparable solution accuracy to LS-SVMs ($< 0.13\%$ relative MAE, RMSE, and $\|y - \hat{y}\|_\infty$ difference). Notably, the computational time (including training and prediction time) of NLS-SVMs was reduced by a factor of 10 to 3355 compared to LS-SVMs, depending on the number of time steps $n$. In comparison to PINNs, the proposed NLS-SVMs model exhibited consistently superior accuracy with maximum reduction by 72% in RMSE, while runs 49-6426 times faster than PINNs (see Appendix B.8 for detailed results).

The scalability of NLS-SVMs is further validated on problems scaled to 100 seconds with $n = 5 \times 10^4$ time steps. Under these large-scale conditions, conventional LS-SVMs and PINNs failed to execute due to memory constraints on the tested hardware, whereas NLS-SVMs produced stable and accurate solutions with execution times between 17.5 and 460 seconds (see Appendix B.6 for details). This performance highlights the method's practical utility in resource-limited settings. While the framework is validated on ODEs, the underlying Nyström acceleration is readily extendable to large-scale partial differential equations by reformulating the differential operators and to other kernel-based ML tasks requiring efficient approximation. It is important to acknowledge a primary limitation of the current method: although it achieves a significant acceleration for nonlinear ODEs, the computational complexity remains at $O(n^3)$. Our future work will focus on breaking this complexity barrier.

Table 3: Comprehensive performance comparison of NLS-SVMs against PINN and LS-SVMs for solving ODEs: accuracy and efficiency

| ODEs | Model | $\mathbf{\Delta} R^2$ | $\mathbf{\Delta} MAE/\%$ | $\mathbf{\Delta} RMSE/\%$ | $\mathbf{\Delta} \|y - \hat{y}\|_\infty /\%$ | Speedup |
|------|-------|------|------|------|------|------|
| **P4** | NLS-SVMs vs PINN | +0.05 | $\downarrow 5.34 \times 10^1$ | $\downarrow 7.20 \times 10^1$ | $\downarrow 1.81 \times 10^2$ | $\uparrow 49$ |
| | NLS-SVMs vs LS-SVMs | +0.00 | $\uparrow 4.94 \times 10^{-2}$ | $\uparrow 6.14 \times 10^{-2}$ | $\uparrow 1.29 \times 10^{-1}$ | $\uparrow 370$ |
| **P5** | NLS-SVMs vs PINN | -0.00 | $\downarrow 1.51 \times 10^{-2}$ | $\uparrow 3.00 \times 10^{-4}$ | $\uparrow 5.40 \times 10^{-2}$ | $\uparrow 171$ |
| | NLS-SVMs vs LS-SVMs | +0.00 | $\downarrow 1.00 \times 10^{-3}$ | $\downarrow 4.00 \times 10^{-3}$ | $\downarrow 1.70 \times 10^{-2}$ | $\uparrow 10$ |
| **P16** | NLS-SVMs vs PINN | +0.00 | $\downarrow 1.24 \times 10^0$ | $\downarrow 1.35 \times 10^0$ | $\downarrow 1.79 \times 10^0$ | $\uparrow 6426$ |
| | NLS-SVMs vs LS-SVMs | +0.00 | $\downarrow 1.33 \times 10^{-2}$ | $\downarrow 1.64 \times 10^{-2}$ | $\downarrow 6.33 \times 10^{-2}$ | $\uparrow 58$ |

# 6 Conclusion

This study introduces a novel Nyström-accelerated Least Squares Support Vector Machines (NLS-SVMs) framework for solving ODEs, establishing a significant improvement in computational efficiency and scalability over classical ML methods such as standard LS-SVMs and physics-informed neural networks (PINNs). The method successfully overcomes the $O(an^3)$ ($a = 1$ for linear ODEs and 27 for nonlinear ODEs) complexity bottleneck inherent in classical LS-SVMs for ODE problems, achieving speedup factors ranging from 10 to 6000—depending on the problem size and characteristics—while consistently preserving solution accuracy. Extensive numerical experiments across 16 benchmark problems confirm the model's versatility and robustness, with relative errors remaining below 0.1% across all tested cases. Even for large-scale systems with fine temporal discretization, the solution time is bounded within 460 seconds, demonstrating the practical viability of the framework in real-time engineering applications such as dynamic system control and multi-physics simulation, where conventional kernel methods or neural models are often limited by memory or latency constraints.

## Acknowledgements

The authors acknowledge the funding support provided by the National Natural Science Foundation of China (No. 52208485), the Key Research Project of the Educational Commission of Hubei Province of China (Grant No. D20241203), and the Talent Research Startup Fund of China Three Gorges University (No. 2023RCKJ013).

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

# A    Technical Appendices

## A.1    Least Squares Support Vector Machines (LS-SVMs)

The learning objective of LS-SVMs is to find a nonlinear mapping relationship [7] between the input variable $\boldsymbol{x}_i \in R^d$ and the output variable $y_i \in R$ from the given training dataset $\{\boldsymbol{x}_i, y_i\}_{i=1}^n$. The goal in a regression problem is to estimate a model of the form $y(\boldsymbol{x}) = \boldsymbol{\omega}^T \boldsymbol{\varphi}(\boldsymbol{x}) + b$. A simple description of the mathematical model follows:

$$\underset{\boldsymbol{\omega}, b, e_i}{\text{minimize}} : J_p(\boldsymbol{\omega}, e_i) = \frac{1}{2}\boldsymbol{\omega}^T\boldsymbol{\omega} + \frac{\gamma}{2}\sum_{i=1}^n e_i^2 \tag{22}$$

$$\text{subject to} : y_i = \boldsymbol{\omega}^T \boldsymbol{\varphi}(\boldsymbol{x}_i) + b + e_i, \qquad i = 1, ..., n \tag{23}$$

where $e_i \in R$ denotes the error variable, $\gamma$ is a regularization parameter, $b \in R$ is the bias term, $\boldsymbol{\omega} \in R^h$ is the weight parameters, $\boldsymbol{\varphi}(\boldsymbol{x}_i) \in R^h$ is a high-dimensional feature variable of input data $\boldsymbol{x}_i$ , where $\varphi(\cdot) : R^d \to R^h$ is the feature map and $h$ is the dimension of the feature space, and $y_i$ indicates the output data. The dual solution is then given by

$$\left[\begin{array}{c|c} \boldsymbol{\Omega} + \mathbf{I}_{n \times n}/\gamma & \mathbf{1}_n \\ \hline \mathbf{1}_n{}^T & 0 \end{array}\right]\left[\begin{array}{c} \boldsymbol{\alpha} \\ b \end{array}\right] = \left[\begin{array}{c} \boldsymbol{y} \\ 0 \end{array}\right] \tag{24}$$

where $\boldsymbol{\Omega}_{ij} = K(\boldsymbol{x}_i, \boldsymbol{x}_j) = \boldsymbol{\varphi}(\boldsymbol{x}_i)^T \boldsymbol{\varphi}(\boldsymbol{x}_j)$ is the $ij$-th entry of the kernel matrix. $\mathbf{1}_n = [1, \ldots, 1]^T \in \mathbb{R}^n$, $\boldsymbol{\alpha} = [\alpha_1, \ldots, \alpha_n]^T$, $\mathbf{y} = [y_1, \ldots, y_n]^T$ and $\mathbf{I}_{n \times n}$ is the identity matrix. The model in the dual form becomes: $y(\boldsymbol{x}) = \sum_{i=1}^n \alpha_i K(\boldsymbol{x}, \boldsymbol{x}_i) + b$.

## A.2    LS-SVM for $p$-th order nonlinear ODE

The Lagrangian of the constrained optimization problem becomes

$$\begin{aligned} \mathcal{L}\left(\boldsymbol{\omega}, b, e_i, \xi_i, y_i, \alpha_i, \eta_i, \beta_i\right) = {}& \frac{1}{2}\boldsymbol{\omega}^T\boldsymbol{\omega} + \frac{\gamma}{2}\sum_{i=1}^n e_i^2 + \frac{\gamma}{2}\sum_{i=1}^n \xi_i^2 \\ & - \sum_{i=2}^N \alpha_i\left(\boldsymbol{\omega}^T\boldsymbol{\varphi}^{(p)}(t_i) - f\left(t_i, y_{t_i}^{(p-1)}, ..., y_{t_i}', y_{t_i}\right) - e_i\right) \\ & - \beta_1\left(\boldsymbol{\omega}^T\boldsymbol{\varphi}(t_1) + b - v_1\right) - ... - \beta_p\left(\boldsymbol{\omega}^T\boldsymbol{\varphi}^{(p-1)}(t_1) - v_p\right) \\ & - \sum_{i=2}^N \eta_i\left(y_i - \boldsymbol{\omega}^T\boldsymbol{\varphi}(t_i) - b - \xi_i\right) \end{aligned} \tag{25}$$

After obtaining KKT optimality conditions and making use of Mercer's theorem, the solution is obtained in the dual by solving the following nonlinear system of equations:

$$\left[\begin{array}{c|c|c|c|c|c} \widehat{\boldsymbol{\Omega}}_1^1 & \tilde{\boldsymbol{\Omega}}_0^1 & \boldsymbol{h}_1^T & \ldots & \mathbf{0}_{n-1} & 0_{(n-1)\times(n-1)} \\ \hline \left(\tilde{\boldsymbol{\Omega}}_0^1\right)^T & \widehat{\boldsymbol{\Omega}}_0^0 & \boldsymbol{h}_0^T & \ldots & \mathbf{1}_{n-1} & -I_{n-1} \\ \hline \boldsymbol{h_1} & \boldsymbol{h_0} & \left[\boldsymbol{\Omega}_0^0\right]_{1,1} & \ldots & 1 & \mathbf{0}_{n-1}^T \\ \hline \ldots & \ldots & \ldots & \ldots & \ldots & \ldots \\ \hline \mathbf{0}_{n-1}^T & \mathbf{1}_{n-1}^T & 1 & \ldots & 0 & \mathbf{0}_{n-1}^T \\ \hline D(\boldsymbol{y}) & I_{n-1} & \mathbf{0}_{n-1} & \ldots & \mathbf{0}_{n-1} & 0_{(n-1)\times(n-1)} \end{array}\right]\left[\begin{array}{c} \boldsymbol{\alpha} \\ \hline \boldsymbol{\eta} \\ \hline \beta_1 \\ \hline \ldots \\ \hline \beta_{p-1} \\ \hline b \\ \hline \boldsymbol{y} \end{array}\right] = \left[\begin{array}{c} \boldsymbol{f}(\boldsymbol{y}) \\ \hline \mathbf{0}_{n-1} \\ \hline v_1 \\ \hline \ldots \\ \hline v_{p-1} \\ \hline 0 \\ \hline \mathbf{0}_{n-1} \end{array}\right] \tag{26}$$

where

$$\widehat{\boldsymbol{\Omega}}_1^1 = \tilde{\boldsymbol{\Omega}}_1^1 + I_{n-1}/\gamma, \quad \widehat{\boldsymbol{\Omega}}_0^0 = \tilde{\boldsymbol{\Omega}}_0^0 + I_{n-1}/\gamma, D(\boldsymbol{y}) = \text{diag}\left(f'(\boldsymbol{y})\right)$$

$$\boldsymbol{f}(\boldsymbol{y}) = [f(t_2, y_2), \ldots, f(t_n, y_n)]^T, \boldsymbol{f}'(\boldsymbol{y}) = \left[\left.\frac{\partial f(t, y)}{\partial y}\right|_{t=t_2, y=y_2}, \ldots, \left.\frac{\partial f(t, y)}{\partial y}\right|_{t=t_n, y=y_n}\right]$$

$$\boldsymbol{\alpha} = [\alpha_2, \ldots, \alpha_n]^T, \boldsymbol{\eta} = [\eta_2, \ldots, \eta_n]^T, \boldsymbol{y} = [y_2, \ldots, y_n]^T, \tilde{\boldsymbol{\Omega}}_0^0 = \left[\boldsymbol{\Omega}_0^0\right]_{2:n, 2:n}$$

$$\tilde{\boldsymbol{\Omega}}_1^1 = \left[\boldsymbol{\Omega}_1^1\right]_{2:n, 2:n}, \tilde{\boldsymbol{\Omega}}_0^1 = \left[\boldsymbol{\Omega}_0^1\right]_{2:n, 2:n}, \boldsymbol{h_0} = \left[\left[\boldsymbol{\Omega}_0^0\right]_{1,2}, \ldots, \left[\boldsymbol{\Omega}_0^0\right]_{1,n}\right]$$

$$\boldsymbol{h_1} = \left[\left[\boldsymbol{\Omega}_0^1\right]_{1,2}, \ldots, \left[\boldsymbol{\Omega}_0^1\right]_{1,n}\right], \mathbf{0}_{n-1} = [0, \ldots, 0]^T \in \mathbb{R}^{n-1}$$

## A.3 Nyström Method for Feature Formulation

This appendix section presents the nonlinear feature mapping using Nyström method [7] . Firstly, given the input dataset $\{t_i\}_{i=1}^n$ , the corresponding kernel matrix consisting of Gaussian kernel functions is denoted as $\mathbf{\Omega}_{(n,n)} \in R^{n \times n}$ , where $\Omega_{i,j} = K(t_i, t_j)$, $i, j = 1, ..., n$ . According to Mercer's theorem, in a high-dimensional $h(n < h \le \infty)$ feature space, the following relation can be obtained:

$$K(t_i, t_j) = \sum_{k=1}^{h} \varsigma_k \phi_k(t_i) \phi_k(t_j) \tag{27}$$

where $\varsigma_1 \ge \varsigma_2 \ge \cdots \ge 0$ represent the eigenvalues; $\phi_1, \phi_2, \cdots, \phi_h$ represent the corresponding eigenfunctions. The eigenvalues and eigenfunctions are associated with the following integral equation:

$$\int K(\boldsymbol{z}, \boldsymbol{t}) \phi_k(t) p(t) dt = \varsigma_k \phi_k(\boldsymbol{z}) \tag{28}$$

where $p(t)$ represents the continuous-type probability density function of the input space $t$. In case of a large-scale training dataset $\{t_i, y_i\}_{i=1}^n$, it is assumed that the input space $\{t_i\}_{i=1}^n$ is distributed independently and identically between them, and all of them are sampled from $p(t)$. To approximate the integral equation of the above eigenfunction, $m$ sample data $\{t_s, y_s\}_{s=1}^m (m \ll n)$ can be selected from the large-scale training dataset $\{t_i, y_i\}_{i=1}^n$, In this case, $p(t)$ can be approximated by the sampled input dataset $\{t_s\}_{s=1}^m$ so that the continuous $p(t)$ can be characterized by the discrete $\{\boldsymbol{t_s}\}_{s=1}^m$ . Then, Eq.(28) can be written as follows:

$$\frac{1}{m} \sum_{s=1}^{m} K(\boldsymbol{z}, t_s) \phi_k(t_s) \approx \varsigma_k \phi_k(\boldsymbol{z}) \tag{29}$$

For the selected sample data $\{t_s\}_{s=1}^m$, the corresponding kernel matrix $\mathbf{\Omega}_{(m,m)}$ and the decomposition of its eigenvalues and eigenvectors can be described by the following relation:

$$\mathbf{\Omega}_{(m,m)} \hat{\mathbf{\Phi}}_{(m \times m)} = \hat{\mathbf{\Phi}}_{(m \times m)} \hat{\mathbf{\Lambda}}_{(m \times m)} \tag{30}$$

where $\mathbf{\Omega}_{(m,m)} \in R^{m \times m}$ represents the small-scale kernel matrix approximating the large-scale original kernel matrix $\mathbf{\Omega}_{(n,n)}$ by $\Omega_{(n,n)} \cong \Omega_{(n,m)} [\Omega_{(m,m)}]^{-1} \Omega_{(m,n)}$ with elements $\Omega_{sz} = K(t_s, t_z)$, $s, z = 1, ..., m$; $\hat{\mathbf{\Phi}}_{(m \times m)} \in R^{m \times m}$ represents a matrix consisting of the eigenvectors of the kernel matrix $\mathbf{\Omega}_{(m,m)}$ , where $\hat{\Phi}_{sk} = \hat{\phi}_k(t_s), s, k = 1, ..., m$, and $\hat{\mathbf{\Lambda}}_{(m \times m)} = diag([\hat{\varsigma}_1; ...; \hat{\varsigma}_m]) \in R^{m \times m}$ represents the diagonal matrix consisting of the eigenvalues of the kernel matrix $\mathbf{\Omega}_{(m,m)}$. Since $\mathbf{\Omega}_{(m,m)}$ is a symmetric semi-positive definite matrix, each element in $\hat{\mathbf{\Lambda}}_{(m \times m)}$ is no less than 0.

Then we can utilize the eigenvalues and eigenvectors of the small-scale kernel matrix $\boldsymbol{Q} = \mathbf{\Omega}_{(m,m)}$ to extract the features based on the proposed NLS-SVMs. By replacing $\boldsymbol{z}$ in Eq.(29) with $t_z$ and substituting $s, z, k = 1, ..., q$ all into Eq.(29) and writing the summary equation in the form of Eq.(30), the following equation is derived:

$$\phi_k(t_s) \approx \sqrt{m} \hat{\Phi}_{sk}; \qquad \varsigma_k = \frac{1}{m} \hat{\varsigma}_k \tag{31}$$

Substituting Eq.(31) into Eq.(29) leads to the $k$-th eigenfunction expression:

$$\phi_k(\boldsymbol{t_z}) \approx \frac{\sqrt{m}}{\hat{\varsigma}_k} \sum_{s=1}^{m} K(t_z, t_s) \hat{\Phi}_{sk} \tag{32}$$

The relation between the high-dimensional vector $\varphi(t_i)$ and the eigenfunction $\phi(t_i)$ can be derived as follows:

$$\varphi^T(t_i) \varphi(t_j) = \sum_{k=1}^{m} \sqrt{\hat{\varsigma}_k} \phi_k(t_i) \sqrt{\hat{\varsigma}_k} \phi_k(t_j) \tag{33}$$

Since $\varphi^T(t_i) \varphi(t_j) = \sum_{k=1}^{q} \varphi_k(t_i) \varphi_k(t_j)$, substituting this into Eq.(32) and combining it with Eq.(31), we get the following:

$$\varphi_k(t_i) = \sqrt{\hat{\varsigma}_k} \phi_k(t_i) = \frac{\sqrt{m}}{\sqrt{\hat{\varsigma}_k}} \sum_{k=1}^{m} \hat{\Phi}_{sk} K(t_k, t_i) \tag{34}$$

The high-dimensional feature vector $\boldsymbol{\varphi}(t_i) \in R^m$ ($m \ll n$) can be obtained from Eq.(34), where $\boldsymbol{\varphi}(t_i) = [\varphi_1(t_i), ..., \varphi_m(t_i)]^T$.

## A.4 NLS-SVMs for Lemma 1

**Proof 1**: The constraint conditions include $n$ time-discrete equations where each represents the discretized form of the ODE equation itself at each time point. The Lagrangian of the constrained optimization problem (Eq.(9)) becomes:

$$L(\boldsymbol{\omega}, b, \lambda_1, ..., \lambda_p) = \frac{1}{2}\boldsymbol{\omega}^T\boldsymbol{\omega} + \frac{\gamma}{2}\left(\boldsymbol{\omega}^T\boldsymbol{\varphi}^{(p)}(t_i) - (\boldsymbol{\omega}^T\left[\sum_{k=1}^{p}\boldsymbol{f}_k(t_i)\boldsymbol{\varphi}_i^{(p-k)}\right] + \boldsymbol{f}_p(t_i)b + r(t_i))\right)^T$$

$$\left(\boldsymbol{\omega}^T\boldsymbol{\varphi}^{(p)}(t_i) - (\boldsymbol{\omega}^T\left[\sum_{k=1}^{p}\boldsymbol{f}_k(t_i)\boldsymbol{\varphi}_i^{(p-k)}\right] + \boldsymbol{f}_p(t_i)b + r(t_i))\right)$$

$$+ \lambda_1\left(\boldsymbol{\omega}^T\boldsymbol{\varphi}(t_1) + b - v_1\right)$$

$$+ ... + \lambda_p\left(\boldsymbol{\omega}^T\boldsymbol{\varphi}^{(p-1)}(t_1) - v_p\right) \tag{35}$$

Then the Karush–Kuhn–Tucker (KKT) optimality conditions are as follows:

$$\begin{cases} \dfrac{\partial L(\boldsymbol{\omega}, b, \lambda_1, \ldots, \lambda_p)}{\partial \boldsymbol{\omega}} = \left[\boldsymbol{E} + \mathbf{A}\left(\boldsymbol{\Psi}^{(p)} - \boldsymbol{f_1}\boldsymbol{\Psi}^{(p-1)} - \cdots - \boldsymbol{f_p}\boldsymbol{\Psi}\right)\right]\boldsymbol{\omega} \\ \qquad\qquad - \left[\mathbf{A}\left(\boldsymbol{f_p}\cdot\mathbf{1}\right)\right]b + \boldsymbol{\varphi}^T(t_1)\lambda_1 + \cdots + \left(\boldsymbol{\varphi}^{(p-1)}(t_1)\right)^T\lambda_p - \mathbf{A}\boldsymbol{r} \\[2mm] \dfrac{\partial L(\boldsymbol{\omega}, b, \lambda_1, \ldots, \lambda_p)}{\partial b} = \left[\boldsymbol{V}\left(\boldsymbol{\Psi}^{(p)} - \boldsymbol{f_1}\boldsymbol{\Psi}^{(p-1)} - \cdots - \boldsymbol{f_p}\boldsymbol{\Psi}\right)\right]\boldsymbol{\omega} \\ \qquad\qquad - \left[\boldsymbol{V}\left(\boldsymbol{f_p}\cdot\mathbf{1}\right)\right]b + \lambda_1 - \boldsymbol{V}r \\[2mm] \dfrac{\partial L(\boldsymbol{\omega}, b, \lambda_1, \ldots, \lambda_p)}{\partial \lambda_1} = \boldsymbol{\varphi}(t_1)^T\boldsymbol{\omega} + b - v_1 \\[2mm] \qquad\qquad \vdots \\[2mm] \dfrac{\partial L(\boldsymbol{\omega}, b, \lambda_1, \ldots, \lambda_p)}{\partial \lambda_p} = \left(\boldsymbol{\varphi}^{(p-1)}(t_1)\right)^T\boldsymbol{\omega} - v_p \end{cases} \tag{36}$$

Therefore, these equations in the matrix from gives the linear system in Eq.(10). The model in the primal problem: $\hat{y}(t) = \boldsymbol{\omega}^T\boldsymbol{\varphi}(t) + b$

## A.5 Nyström Method for Lemma 2

**Proof 2**: The Lagrangian of the constrained optimization problem Eq.(4) in its primal form becomes:

$$L(\boldsymbol{\omega}, b, \lambda_1, ...\lambda_p, \boldsymbol{y}_i) = \frac{1}{2}\boldsymbol{\omega}^T\boldsymbol{\omega} + \frac{\gamma}{2}\left(\boldsymbol{\omega}^T\boldsymbol{\varphi}^{(p)}(t_i) - f(t_i, y_{t_i}^{(p-1)}, ..., y_{t_i}', y_{t_i})\right)^T$$

$$\left(\boldsymbol{\omega}^T\boldsymbol{\varphi}^{(p)}(t_i) - f(t_i, y_{t_i}^{(p-1)}, ..., y_{t_i}', y_{t_i})\right) + \frac{\gamma}{2}\left(\boldsymbol{y}_{t_i} - \boldsymbol{\omega}^T\boldsymbol{\varphi}(t_i) - b\right)^T$$

$$\left(\boldsymbol{y}_{t_i} - \boldsymbol{\omega}^T\boldsymbol{\varphi}(t_i) - b\right) + \lambda_p\left(\boldsymbol{\omega}^T\boldsymbol{\varphi}^{(p)}(t_1) - v_p\right)$$

$$+ ... + \lambda_1\left(\boldsymbol{\omega}^T\boldsymbol{\varphi}(t_1) + b - v_1\right) \tag{37}$$

Then the KKT optimality conditions are as follows:

$$
\begin{cases}
\dfrac{\partial L(\boldsymbol{\omega}, b, \lambda_1, \ldots, \lambda_p, \boldsymbol{y}_i)}{\partial \boldsymbol{\omega}} = \boldsymbol{Y}_1 \overset{Jacobian}{\rightarrow} \dfrac{\partial \boldsymbol{Y}_1}{\partial(\boldsymbol{\omega}, b, \lambda_1, \ldots, \lambda_p, \boldsymbol{y}_i)} \\[2mm]
\dfrac{\partial L(\boldsymbol{\omega}, b, \lambda_1, \ldots, \lambda_p, \boldsymbol{y}_i)}{\partial b} = Y_2 \overset{Jacobian}{\rightarrow} \dfrac{\partial Y_2}{\partial(\boldsymbol{\omega}, b, \lambda_1, \ldots, \lambda_p, \boldsymbol{y}_i)} \\[2mm]
\dfrac{\partial L(\boldsymbol{\omega}, b, \lambda_1, \ldots, \lambda_p, \boldsymbol{y}_i)}{\partial \lambda_1} = Y_3 \overset{Jacobian}{\rightarrow} \dfrac{\partial Y_3}{\partial(\boldsymbol{\omega}, b, \lambda_1, \ldots, \lambda_p, \boldsymbol{y}_i)} \\[2mm]
\qquad\qquad \vdots \\[2mm]
\dfrac{\partial L(\boldsymbol{\omega}, b, \lambda_1, \ldots, \lambda_p, \boldsymbol{y}_i)}{\partial \lambda_p} = Y_{p+2} \overset{Jacobian}{\rightarrow} \dfrac{\partial Y_{p+2}}{\partial(\boldsymbol{\omega}, b, \lambda_1, \ldots, \lambda_p, \boldsymbol{y}_i)} \\[2mm]
\dfrac{\partial L(\boldsymbol{\omega}, b, \lambda_1, \ldots, \lambda_p, \boldsymbol{y}_i)}{\partial \boldsymbol{y}_i} = \boldsymbol{Y}_{p+3} \overset{Jacobian}{\rightarrow} \dfrac{\partial \boldsymbol{Y}_{p+3}}{\partial(\boldsymbol{\omega}, b, \lambda_1, \ldots, \lambda_p, \boldsymbol{y}_i)}
\end{cases}
\tag{38}
$$

Therefore, these equations in the matrix from gives the nonlinear system in Eq.(11). The model in the primal form becomes $\hat{y}(t) = \boldsymbol{\omega}^T \boldsymbol{\varphi}(t) + b$.

## A.6 NLS-SVMs for Solving First-Order ODE with IVP

As a first example, consider the following first-order IVP:

$$
y'(t) - f_1(t)y(t) = r(t), \quad y(t_1) = v_1, \quad t_1 \le t \le t_n.
\tag{39}
$$

Then start by assuming the approximate solution to be of the form $\hat{y}(t) = \boldsymbol{\omega}^T \boldsymbol{\varphi}(t) + b$. In the NLS-SVMs framework, the approximate solution can be obtained by solving the following optimization problem:

$$
\begin{aligned}
\underset{\boldsymbol{\omega}, b, e_i}{\text{minimize}} \quad & \frac{1}{2}\boldsymbol{\omega}^T\boldsymbol{\omega} + \frac{\gamma}{2}\sum_{i=1}^n e_i^2 \\
\text{s.t.} \quad & y'(t_i) = \boldsymbol{\omega}^T\boldsymbol{\varphi}'(t_i) = f_1(t_i)\left[\boldsymbol{\omega}^T\boldsymbol{\varphi}(t_i) + b\right] + r(t_i) + \boldsymbol{e}_i, \quad i = 2, \ldots, n \\
& y(t_1) = \boldsymbol{\omega}^T\boldsymbol{\varphi}(t_1) + b = v_1
\end{aligned}
\tag{40}
$$

The Lagrangian of the constrained optimization problem becomes

$$
\begin{aligned}
\boldsymbol{L}(\boldsymbol{\omega}, b, \lambda) = {} & \frac{1}{2}\boldsymbol{\omega}^T\boldsymbol{\omega} + \frac{\gamma}{2}\left(\boldsymbol{\omega}^T\boldsymbol{\varphi}'(t_i) - f_1(t_i)\left[\boldsymbol{\omega}^T\boldsymbol{\varphi}(t_i) + b\right] - r(t_i)\right)^T \\
& \cdot \left(\boldsymbol{\omega}^T\boldsymbol{\varphi}'(t_i) - f_1(t_i)\left[\boldsymbol{\omega}^T\boldsymbol{\varphi}(t_i) + b\right] - r(t_i)\right) \\
& + \lambda\left(\boldsymbol{\omega}^T\boldsymbol{\varphi}(t_1) + b - v_1\right)
\end{aligned}
\tag{41}
$$

Then the Karush–Kuhn–Tucker (KKT) optimality conditions are as follows: $\frac{\partial L(\boldsymbol{\omega}, b, \lambda)}{\partial \boldsymbol{\omega}} = \boldsymbol{0}, \frac{\partial L(\boldsymbol{\omega}, b, \lambda)}{\partial b} = 0, \frac{\partial L(\boldsymbol{\omega}, b, \lambda)}{\partial \lambda} = 0$, therefore, the solution to Eq.(41) is obtained by solving the following primary problem:

$$
\begin{bmatrix}
\boldsymbol{E} + \mathbf{A}\left(\boldsymbol{\Psi}' - \boldsymbol{f} \cdot \boldsymbol{\Psi}\right) & -\mathbf{A}\left(\boldsymbol{f} \cdot \mathbf{1}\right) & -\boldsymbol{\varphi}^T(t_1) \\
\boldsymbol{V}\left(\boldsymbol{\Psi}' - \boldsymbol{f} \cdot \boldsymbol{\Psi}\right) & -\boldsymbol{V}\left(\boldsymbol{f} \cdot \mathbf{1}\right) & -1 \\
\boldsymbol{\varphi}(t_1) & 1 & 0
\end{bmatrix}
\begin{bmatrix}
\boldsymbol{\omega} \\
b \\
\lambda
\end{bmatrix}
=
\begin{bmatrix}
\mathbf{A}\boldsymbol{r} \\
\boldsymbol{V}\boldsymbol{r} \\
v_1
\end{bmatrix}
\tag{42}
$$

where $\boldsymbol{E} \in \mathbb{R}^{q \times q}$ is an identity matrix; $\boldsymbol{\Psi} = [\varphi(t_2), \ldots, \varphi(t_n)]^T \in \mathbb{R}^{(n-1) \times m}$; $\boldsymbol{\Psi}' = [\varphi'(t_2), \ldots, \varphi'(t_n)]^T \in \mathbb{R}^{(n-1) \times m}$; $\boldsymbol{f} = [f_1(t_2), \ldots, f_1(t_n)]^T \in \mathbb{R}^{n-1}$; $\mathbf{1} = [1, \ldots, 1]^T \in \mathbb{R}^{n-1}$; $\boldsymbol{\varphi}(t_1) \in \mathbb{R}^{1 \times m}; \mathbf{A} = \gamma[\boldsymbol{\Psi}' - \boldsymbol{f}\boldsymbol{\Psi}]^T \in \mathbb{R}^{m \times (n-1)}; \boldsymbol{V} = \gamma[-\boldsymbol{f}\mathbf{1}]^T \in \mathbb{R}^{1 \times (n-1)}; \boldsymbol{r} = [r(t_2), \ldots, r(t_n)]^T \in \mathbb{R}^{n-1}; \boldsymbol{\omega} \in \mathbb{R}^{m \times 1}; \gamma$ is a regularization parameter.

The model in the primary form becomes $\hat{y}(t) = \boldsymbol{\omega}^T\boldsymbol{\varphi}(t) + b$

## A.7 NLS-SVMs for Solving Second-Order ODEs with IVP and BVP

**IVP Case**     Let us consider a second-order IVP of the form:

$$y''(t) = f_1(t)y'(t) + f_2(t)y(t) + r(t), \quad y(t_1) = v_1, \quad y'(t_1) = v_2, \quad t_1 \leq t \leq t_n. \quad (43)$$

The approximate solution, $\hat{y}(t) = \boldsymbol{\omega}^T \boldsymbol{\varphi}(t) + b$, is then obtained by solving the following optimization problem:

$$\underset{\boldsymbol{\omega},b,e_i}{\text{minimize}} \quad \frac{1}{2}\boldsymbol{\omega}^T\boldsymbol{\omega} + \frac{\gamma}{2}\sum_{i=1}^n e_i^2$$

$$\text{s.t.} \quad y''(t_i) = \boldsymbol{\omega}^T\boldsymbol{\varphi}''(t_i) = f_1(t_i)\boldsymbol{\omega}^T\boldsymbol{\varphi}'(t_i) + f_2(t_i)\left[\boldsymbol{\omega}^T\boldsymbol{\varphi}(t_i) + b\right] + r(t_i) + \boldsymbol{e}_i$$

$$i = 2,\ldots,n$$

$$y'(t_1) = \boldsymbol{\omega}^T\boldsymbol{\varphi}'(t_1) = v_2$$

$$y(t_1) = \boldsymbol{\omega}^T\boldsymbol{\varphi}(t_1) + b = v_1 \quad (44)$$

The Lagrangian of the constrained optimization problem becomes

$$\boldsymbol{L}(\boldsymbol{\omega},b,\lambda_1,\lambda_2) = \frac{1}{2}\boldsymbol{\omega}^T\boldsymbol{\omega} + \frac{\gamma}{2}\left(\boldsymbol{\omega}^T\boldsymbol{\varphi}''(t_i) - f_1(t_i)\boldsymbol{\omega}^T\boldsymbol{\varphi}'(t_i) - f_2(t_i)\left[\boldsymbol{\omega}^T\boldsymbol{\varphi}(t_i) + b\right] - \boldsymbol{r}(t_i)\right)^T$$

$$\left(\boldsymbol{\omega}^T\boldsymbol{\varphi}''(t_i) - f_1(t_i)\boldsymbol{\omega}^T\boldsymbol{\varphi}'(t_i) - f_2(t_i)\left[\boldsymbol{\omega}^T\boldsymbol{\varphi}(t_i) + b\right] - \boldsymbol{r}(t_i)\right)$$

$$+ \lambda_1\left(\boldsymbol{\omega}^T\boldsymbol{\varphi}(t_1) + b - v_1\right) + \lambda_2\left(\boldsymbol{\omega}^T\boldsymbol{\varphi}'(t_1) - v_2\right) \quad (45)$$

Then the Karush–Kuhn–Tucker (KKT) optimality conditions are as follows: $\frac{\partial L(\boldsymbol{\omega},b,\lambda_1,\lambda_2)}{\partial \boldsymbol{\omega}} = \mathbf{0}, \frac{\partial L(\boldsymbol{\omega},b,\lambda_1,\lambda_2)}{\partial b} = 0, \frac{\partial L(\boldsymbol{\omega},b,\lambda_1,\lambda_2)}{\partial \lambda_1} = 0, \frac{\partial L(\boldsymbol{\omega},b,\lambda_1,\lambda_2)}{\partial \lambda_2} = 0$, therefore, the solution to Eq.(45) is obtained by solving the following primary problem:

$$\begin{bmatrix} \boldsymbol{E} + \boldsymbol{A}\left(\boldsymbol{\Psi}'' - \boldsymbol{f_1}\boldsymbol{\Psi}' - \boldsymbol{f_2}\boldsymbol{\Psi}\right) & -\boldsymbol{A}\left(\boldsymbol{f_2}\cdot\mathbf{1}\right) & \boldsymbol{\varphi}^T(t_1) & (\boldsymbol{\varphi}'(t_1))^T \\ \boldsymbol{V}\left(\boldsymbol{\Psi}'' - \boldsymbol{f_1}\boldsymbol{\Psi}' - \boldsymbol{f_2}\boldsymbol{\Psi}\right) & -\boldsymbol{V}\left(\boldsymbol{f_2}\cdot\mathbf{1}\right) & 1 & 0 \\ \boldsymbol{\varphi}(t_1) & 1 & 0 & 0 \\ \boldsymbol{\varphi}'(t_1) & 0 & 0 & 0 \end{bmatrix}\begin{bmatrix} \boldsymbol{\omega} \\ b \\ \lambda_1 \\ \lambda_2 \end{bmatrix} = \begin{bmatrix} \boldsymbol{Ar} \\ \boldsymbol{Vr} \\ v_1 \\ v_2 \end{bmatrix}$$

$$(46)$$

where $\boldsymbol{E} \in \mathbb{R}^{m\times m}$ is an identity matrix; $\boldsymbol{\Psi} = [\varphi(t_2),...,\varphi(t_n)]^T \in \mathbb{R}^{(n-1)\times m}$; $\boldsymbol{\Psi}' = [\varphi'(t_2),...,\varphi'(t_n)]^T \in \mathbb{R}^{(n-1)\times m}$; $\boldsymbol{\Psi}'' = [\varphi''(t_2),...,\varphi''(t_n)]^T \in \mathbb{R}^{(n-1)\times m}$; $\boldsymbol{f_1} = [f_1(t_2),...,f_1(t_n)]^T \in \mathbb{R}^{n-1}$; $\boldsymbol{f_2} = [f_2(t_2),...,f_2(t_n)]^T \in \mathbb{R}^{n-1}$; $\mathbf{1} = [1,...,1]^T \in \mathbb{R}^{n-1}$; $\boldsymbol{\varphi}(t_1) \in \mathbb{R}^{1\times q}$; $\boldsymbol{\varphi}'(t_1) \in \mathbb{R}^{1\times m}$; $\boldsymbol{A} = \gamma[\boldsymbol{\Psi}'' - \boldsymbol{f_1}\boldsymbol{\Psi}' - \boldsymbol{f_2}\boldsymbol{\Psi}]^T \in \mathbb{R}^{m\times(n-1)}$; $\boldsymbol{V} = \gamma[-\boldsymbol{f_2}\mathbf{1}]^T \in \mathbb{R}^{1\times(n-1)}$; $\boldsymbol{r} = [r(t_2),...,r(t_n)]^T \in \mathbb{R}^{n-1}$; $\boldsymbol{\omega} \in \mathbb{R}^{m\times1}$; $\gamma$ is a regularization parameter.

The model in the primary form becomes $\hat{y}(t) = \boldsymbol{\omega}^T\boldsymbol{\varphi}(t) + b$.

**BVP Case**     Consider the second-order BVP of ODEs of the form

$$y''(t) = f_1(t)y'(t) + f_2(t)y(t) + r(t), \quad y(t_1) = q_1, \quad y(t_n) = q_n, \quad t_1 \leq t \leq t_n. \quad (47)$$

Then the parameters of the closed-form approximation of the solution can be obtained by solving the following optimization problem:

$$\underset{\boldsymbol{\omega},b,e_i}{\text{minimize}} \quad \frac{1}{2}\boldsymbol{\omega}^T\boldsymbol{\omega} + \frac{\gamma}{2}\sum_{i=1}^n e_i^2$$

$$\text{s.t.} \quad y''(t_i) = \boldsymbol{\omega}^T\boldsymbol{\varphi}''(t_i) = f_1(t_i)\boldsymbol{\omega}^T\boldsymbol{\varphi}'(t_i) + f_2(t_i)\left[\boldsymbol{\omega}^T\boldsymbol{\varphi}(t_i) + b\right] + r(t_i) + \boldsymbol{e}_i$$

$$i = 2,\ldots,n-1$$

$$y(t_1) = \boldsymbol{\omega}^T\boldsymbol{\varphi}(t_1) + b = q_1$$

$$y(t_n) = \boldsymbol{\omega}^T\boldsymbol{\varphi}(t_n) + b = q_n \quad (48)$$

The same procedure can be applied to derive the Lagrangian and afterward the KKT optimality conditions. Then, one can show that the solution to Eq.(48) is obtained by solving the following linear system:

$$\begin{bmatrix} \boldsymbol{E} + \boldsymbol{A}\left(\boldsymbol{\Psi}'' - \boldsymbol{f_1}\boldsymbol{\Psi}' - \boldsymbol{f_2}\boldsymbol{\Psi}\right) & -\boldsymbol{A}\left(\boldsymbol{f_2}\cdot\mathbf{1}\right) & \boldsymbol{\varphi}^T(t_1) & \boldsymbol{\varphi}^T(t_n) \\ \boldsymbol{V}\left(\boldsymbol{\Psi}'' - \boldsymbol{f_1}\boldsymbol{\Psi}' - \boldsymbol{f_2}\boldsymbol{\Psi}\right) & -\boldsymbol{V}\left(\boldsymbol{f_2}\cdot\mathbf{1}\right) & 1 & 1 \\ \boldsymbol{\varphi}(t_1) & 1 & 0 & 0 \\ \boldsymbol{\varphi}(t_n) & 1 & 0 & 0 \end{bmatrix}\begin{bmatrix} \boldsymbol{\omega} \\ b \\ \lambda_1 \\ \lambda_2 \end{bmatrix} = \begin{bmatrix} \boldsymbol{Ar} \\ \boldsymbol{Vr} \\ q_1 \\ q_n \end{bmatrix}$$

$$(49)$$

where $\boldsymbol{E} \in \mathbb{R}^{m\times m}$ is an identity matrix; $\boldsymbol{\Psi} = [\varphi(t_2),...,\varphi(t_{n-1})]^T \in \mathbb{R}^{(n-2)\times m}$; $\boldsymbol{\Psi}' = [\varphi'(t_2),...,\varphi'(t_{n-1})]^T \in \mathbb{R}^{(n-2)\times m}$; $\boldsymbol{\Psi}'' = [\varphi''(t_2),...,\varphi''(t_{n-1})]^T \in \mathbb{R}^{(n-2)\times m}$; $\boldsymbol{f}_1 = [f_1(t_2),...,f_1(t_{n-1})]^T \in \mathbb{R}^{n-2}$; $\boldsymbol{f}_2 = [f_2(t_2),...,f_2(t_{n-1})]^T \in \mathbb{R}^{n-2}$; $\mathbf{1} = [1,...,1]^T \in \mathbb{R}^{n-2}$; $\boldsymbol{\varphi}(t_1) \in \mathbb{R}^{1\times m}$; $\boldsymbol{\varphi}(t_n) \in \mathbb{R}^{1\times m}$; $\boldsymbol{A} = \gamma[\boldsymbol{\Psi}'' - \boldsymbol{f}_1\boldsymbol{\Psi}' - \boldsymbol{f}_2\boldsymbol{\Psi}]^T \in \mathbb{R}^{m\times(n-2)}$; $\boldsymbol{V} = \gamma[-\boldsymbol{f}_2\mathbf{1}]^T \in \mathbb{R}^{1\times(n-2)}$; $\boldsymbol{r} = [r(t_2),...,r(t_n)]^T \in \mathbb{R}^{n-2}$; $\boldsymbol{\omega} \in \mathbb{R}^{m\times 1}$; $\gamma$ is a regularization parameter.

The model in the primary form becomes $\hat{y}(t) = \boldsymbol{\omega}^T\boldsymbol{\varphi}(t) + b$.

### A.8 NLS-SVMs for Solving First-Order nonlinear ODE

Here, we take Eq.(3) as the object of study. The Lagrangian of the constrained optimization problem becomes

$$\boldsymbol{L}(\boldsymbol{\omega}, b, \lambda, \boldsymbol{y}_i) = \frac{1}{2}\boldsymbol{\omega}^T\boldsymbol{\omega} + \frac{\gamma}{2}\left(\boldsymbol{\omega}^T\boldsymbol{\varphi}'(t_i) - f(t_i, y_i)\right)^T\left(\boldsymbol{\omega}^T\boldsymbol{\varphi}'(t_i) - f(t_i, y_i)\right)$$
$$+ \frac{\gamma}{2}\left(\boldsymbol{y}_i - \boldsymbol{\omega}^T\boldsymbol{\varphi}(t_i) - b\right)^T\left(\boldsymbol{y}_i - \boldsymbol{\omega}^T\boldsymbol{\varphi}(t_i) - b\right) + \lambda\left(\boldsymbol{\omega}^T\boldsymbol{\varphi}(t_1) + b - v_1\right) \quad (50)$$

Then the Karush–Kuhn–Tucker (KKT) optimality conditions are as follows: $\frac{\partial L(\boldsymbol{\omega}, b, \lambda, \boldsymbol{y}_i)}{\partial \boldsymbol{\omega}} = Y_1$, $\frac{\partial L(\boldsymbol{\omega}, b, \lambda, \boldsymbol{y}_i)}{\partial b} = Y_2$, $\frac{\partial L(\boldsymbol{\omega}, b, \lambda, \boldsymbol{y}_i)}{\partial \lambda} = Y_3$, $\frac{\partial L(\boldsymbol{\omega}, b, \lambda, \boldsymbol{y}_i)}{\partial \boldsymbol{y}_i} = Y_4$, therefore, the solution to Eq.(50) is obtained by solving the following primary problem:

$$\begin{bmatrix} \frac{\partial \boldsymbol{Y_1}}{\partial \boldsymbol{\omega}} & \frac{\partial \boldsymbol{Y_1}}{\partial b} & \frac{\partial \boldsymbol{Y_1}}{\partial \lambda} & \frac{\partial \boldsymbol{Y_1}}{\partial \boldsymbol{y}_i} \\ \frac{\partial Y_2}{\partial \boldsymbol{\omega}} & \frac{\partial Y_2}{\partial b} & \frac{\partial Y_2}{\partial \lambda} & \frac{\partial Y_2}{\partial \boldsymbol{y}_i} \\ \frac{\partial Y_3}{\partial \boldsymbol{\omega}} & \frac{\partial Y_3}{\partial b} & \frac{\partial Y_3}{\partial \lambda} & \frac{\partial Y_3}{\partial \boldsymbol{y}_i} \\ \frac{\partial \boldsymbol{Y_4}}{\partial \boldsymbol{\omega}} & \frac{\partial \boldsymbol{Y_4}}{\partial b} & \frac{\partial \boldsymbol{Y_4}}{\partial \lambda} & \frac{\partial \boldsymbol{Y_4}}{\partial \boldsymbol{y}_i} \end{bmatrix} \begin{bmatrix} \boldsymbol{\omega} \\ b \\ \lambda \\ \boldsymbol{y}_i \end{bmatrix} = \begin{bmatrix} \boldsymbol{Y_1} \\ Y_2 \\ Y_3 \\ \boldsymbol{Y_4} \end{bmatrix} \quad (51)$$

where $Y1 \in \mathbb{R}^{m\times 1}$; $Y2 \in \mathbb{R}^{1\times 1}$; $Y3 \in \mathbb{R}^{1\times 1}$; $Y4 \in \mathbb{R}^{(n-1)\times 1}$. The nonlinear system, which consists of equations with unknowns $(\boldsymbol{\omega}, b, \lambda, \boldsymbol{y}_i)$, is solved by Newton's method. The model in the primary form becomes $\hat{y}(t) = \boldsymbol{\omega}^T\boldsymbol{\varphi}(t) + b$.

# B  Supplementary Material

## B.1  Sixteen benchmark ODE problems

**Problem 1**: Consider the first-order ODE with time varying coefficient [8, 36, 37] $y'(t) + \left(t + \frac{1+3t^2}{1+t+t^3}\right)y = t^3 + 2t + t^2\left(\frac{1+3t^2}{1+t+t^3}\right), t \in [0,3], y(0) = 1$

**Problem 2**: Consider thd Dahlquist's problem as a non-stiff linear problem [38] given by $y'(t) = -y(t), t \in (0,10], y(0) = 1$

**Problem 3**: Consider a well-known stiff test problem, the Prothero–Robinson equation [38, 39], $y'(t) = -10^3(y(t) - \sin t) + \cos t, t \in (0,10]; y(0) = 0$

**Problem 4**: Consider the following nonlinear first-order ODE, which is studied by Junaid et al [40], $y'(t) = y(t) - \frac{t}{y(t)}, t \in (0,3], y(0) = 1$

**Problem 5**: Consider the following nonlinear first-order ODE [38] defined by $y'(t) = y^2(t), t \in (0,1], y(0) = -1$

**Problem 6**: Consider the following nonlinear first-order ODE [41] defined by $y'(t) = 2t^{-1}\sqrt{y(t) - \ln t} + t^{-1}, t \in (1,1.2], y(1) = 0$

**Problem 7**: Consider the following second-order IVP [42] defined by $y''(t) + 0.3y'(t) + y(t) = 1, t \in (0,20], y(0) = 0, y'(0) = 0$, the interval solution for this test problem is considered from 0 to 20, which is assumed to be a wide range for solutions.

**Problem 8**: Consider the second ODE having the initial conditions based on the laws of physics as follows [42]: $\frac{1}{2}y''(t) + 18y(t) = 0, t \in (0,2], y(0) = -\frac{1}{2}, y'(0) = 1$

**Problem 9**: Consider the following second order damped free vibration equation [36]: $y''(t) + 4y'(t) + 4y(t) = 0, t \in [0,10], \quad y(0) = 1, y'(0) = 1$

**Problem 10**: Consider the following second-order ODE with time-varying input signal [37]: $y''(t) + \frac{1}{5}y'(t) + y(t) = -\frac{1}{5}e^{-(t/5)}\cos t, t \in [0, 10], y(0) = 0, y'(0) = 1$

**Problem 11**: Consider a second-order singular ordinary differential equation [43]: $y''(t) + \frac{1}{t}y'(t) + \cos(t) + \frac{\sin(t)}{t} = 0, t \in [0, 1], y(0) = 1, y(1) = \cos(1)$

**Problem 12**: Consider the following second-order ODE [44]: $y''(t) + y(t) = 2, t \in [0, 1], y(0) = 1, y(1) = 0$

**Problem 13**: Consider a singular differential equation [45, 43]: $y''(t) + \frac{1-2t}{t}y'(t) + \frac{t-1}{t}y(t) = 0, t \in [0, 1], y(0) = 1, y(1) = e$

**Problem 14**: Consider one of the most famous second order ODE. It is given as follows [42, 46]: $y''(t) + (y'(t))^2 - 2e^{-y(t)} = 0, t \in (0, 1], y(0) = 0, y(1) = 0$

**Problem 15**: Consider the following nonlinear second-order ODE [41, 47] defined by $y''(t) = -y(t) + \frac{2(y'(t))^2}{y(t)}, t \in [-1, 1], y(-1) = y(1) = 0.324027137$

**Problem 16**: Consider a fourth-order linear ODE [48, 49] defined by $y^{(4)}(t) = 120t, t \in [-1, 1], y(-1) = 1, y'(-1) = 5, y(1) = 3, y'(1) = 5$

## B.2   NLS-SVMs for Solving Second-order ODE in Terms of Problem 15

$Problem 15 : y''(t) = -y(t) + \frac{2(y'(t))^2}{y(t)}, t \in [-1, 1], y(-1) = y(1) = 0.324027137$

$$\underset{\boldsymbol{\omega}, b, e_i, \xi_i, y_i}{\text{minimize}} \frac{1}{2}\boldsymbol{\omega}^T\boldsymbol{\omega} + \frac{\gamma}{2}\sum_{i=1}^{n}e_i^2 + \frac{\gamma}{2}\sum_{i=1}^{n}\xi_i^2 \tag{52}$$

$$\begin{aligned}
\text{s.t. } & y''(t_i) = \boldsymbol{\omega}^T\boldsymbol{\varphi}''(t_i) = -y(t_i) + 2(y'(t_i))^2/y(t_i) + e_i, \quad i = 2, \ldots, n-1 \\
& y'(t_i) = \boldsymbol{\omega}^T\boldsymbol{\varphi}'(t_i), \quad i = 2, \ldots, n-1 \\
& y(t_i) = \boldsymbol{\omega}^T\boldsymbol{\varphi}(t_i) + b + \xi_i, \quad i = 2, \ldots, n-1 \\
& \boldsymbol{\omega}^T\boldsymbol{\varphi}(t_1) + b = q_1 \\
& \boldsymbol{\omega}^T\boldsymbol{\varphi}(t_n) + b = q_n.
\end{aligned}$$

The Lagrangian of the constrained optimization problem becomes

$$\begin{aligned}
\boldsymbol{L}(\boldsymbol{\omega}, b, \lambda_1, \lambda_2, \boldsymbol{y}_i) = & \frac{1}{2}\boldsymbol{\omega}^T\boldsymbol{\omega} + \frac{\gamma}{2}\Big(\boldsymbol{w}^T\boldsymbol{\varphi}''(t_i) - (-y(t_i) + 2(y'(t_i))^2/y(t_i))\Big)^T \\
& \Big(\boldsymbol{w}^T\boldsymbol{\varphi}''(t_i) - (-y(t_i) + 2(y'(t_i))^2/y(t_i))\Big) + \frac{\gamma}{2}\left(\boldsymbol{y}_i - \boldsymbol{\omega}^T\boldsymbol{\varphi}(t_i) - b\right)^T \\
& \left(\boldsymbol{y}_i - \boldsymbol{\omega}^T\boldsymbol{\varphi}(t_i) - b\right) + \lambda_1\left(\boldsymbol{\omega}^T\boldsymbol{\varphi}(t_1) + b - q_1\right) + \lambda_2\left(\boldsymbol{\omega}^T\boldsymbol{\varphi}(t_N) + b - q_n\right)
\end{aligned} \tag{53}$$

Further simplified:

$$\begin{aligned}
\boldsymbol{L}(\boldsymbol{\omega}, b, \lambda_1, \lambda_2, \boldsymbol{y}_i) = & \frac{1}{2}\boldsymbol{\omega}^T\boldsymbol{\omega} + \frac{\gamma}{2}\Big(\boldsymbol{w}^T\boldsymbol{\varphi}''(t_i) - (-\boldsymbol{y}_i + 2(\boldsymbol{\omega}^T\boldsymbol{\varphi}'(t_i))^2/\boldsymbol{y}_i)\Big)^T \\
& \Big(\boldsymbol{w}^T\boldsymbol{\varphi}''(t_i) - (-\boldsymbol{y}_i + 2(\boldsymbol{\omega}^T\boldsymbol{\varphi}'(t_i))^2/\boldsymbol{y}_i)\Big) + \frac{\gamma}{2}\left(\boldsymbol{y}_i - \boldsymbol{\omega}^T\boldsymbol{\varphi}(t_i) - b\right)^T \\
& \left(\boldsymbol{y}_i - \boldsymbol{\omega}^T\boldsymbol{\varphi}(t_i) - b\right) + \lambda_1\left(\boldsymbol{\omega}^T\boldsymbol{\varphi}(t_1) + b - q_1\right) + \lambda_2\left(\boldsymbol{\omega}^T\boldsymbol{\varphi}(t_N) + b - q_n\right)
\end{aligned} \tag{54}$$

Then the Karush–Kuhn–Tucker (KKT) optimality conditions are as follows:

$$\boldsymbol{Y}1: \frac{\partial L(\boldsymbol{\omega}, b, \lambda_1, \lambda_2, \boldsymbol{y}_i)}{\partial \boldsymbol{\omega}} = \boldsymbol{\omega} + \gamma\big[(\boldsymbol{\varphi}''_{t_i})^T[\boldsymbol{\varphi}''_{t_i}\boldsymbol{\omega} - 2(\boldsymbol{\varphi}'_{t_i}\boldsymbol{\omega})^2/\boldsymbol{y}_i + \boldsymbol{y}_i] - 4(\boldsymbol{\varphi}'_{t_i})^T[(\boldsymbol{\varphi}'_{t_i}\boldsymbol{\omega}/\boldsymbol{y}_i)\odot$$
$$(\boldsymbol{\varphi}''_{t_i}\boldsymbol{\omega} - 2(\boldsymbol{\varphi}'_{t_i}\boldsymbol{\omega})^2/\boldsymbol{y}_i + \boldsymbol{y}_i)]\big] + \gamma[-\boldsymbol{\varphi}_{t_i}]^T[\boldsymbol{y}_i - \boldsymbol{\varphi}_{t_i}\boldsymbol{\omega} - b\boldsymbol{I}]$$
$$+ \lambda_1\boldsymbol{\varphi}_{t_1} + \lambda_2\boldsymbol{\varphi}_{t_n} \tag{55}$$

$$\boldsymbol{Y}2: \frac{\partial L(\boldsymbol{\omega}, b, \lambda_1, \lambda_2, \boldsymbol{y}_i)}{\partial b} = \gamma[-\boldsymbol{I}]^T[\boldsymbol{y}_i - \boldsymbol{\varphi}_{t_i}\boldsymbol{\omega} - b\boldsymbol{I}] + \lambda_1 + \lambda_2 \tag{56}$$

$$\boldsymbol{Y}3: \frac{\partial L(\boldsymbol{\omega}, b, \lambda_1, \lambda_2, \boldsymbol{y}_i)}{\partial \lambda_1} = \boldsymbol{\varphi}_{t_1}\boldsymbol{\omega} + b - q_1 \tag{57}$$

$$\boldsymbol{Y}4: \frac{\partial L(\boldsymbol{\omega}, b, \lambda_1, \lambda_2, \boldsymbol{y}_i)}{\partial \lambda_2} = \boldsymbol{\varphi}_{t_N}\boldsymbol{\omega} + b - q_n \tag{58}$$

$$\boldsymbol{Y}5: \frac{\partial L(\boldsymbol{\omega}, b, \lambda_1, \lambda_2, \boldsymbol{y}_i)}{\partial \boldsymbol{y}_i} = \gamma\big[2(\boldsymbol{\varphi}'_{t_i}\boldsymbol{\omega})^2/\boldsymbol{y}_i + 1\big] \odot \big[\boldsymbol{\varphi}''_{t_i}\boldsymbol{\omega} - 2(\boldsymbol{\varphi}'_{t_i}\boldsymbol{\omega})^2/\boldsymbol{y}_i + \boldsymbol{y}_i\big]$$
$$+ \gamma[\boldsymbol{y}_i - \boldsymbol{\varphi}_{t_i}\boldsymbol{\omega} - b\boldsymbol{I}] \tag{59}$$

Construct the Jacobian matrix and establish the matrix equation:

$$\begin{bmatrix} \frac{\partial \boldsymbol{Y_1}}{\partial \boldsymbol{\omega}} & \frac{\partial \boldsymbol{Y_1}}{\partial b} & \frac{\partial \boldsymbol{Y_1}}{\partial \lambda_1} & \frac{\partial \boldsymbol{Y_1}}{\partial \lambda_2} & \frac{\partial \boldsymbol{Y_1}}{\partial \boldsymbol{y}_i} \\ \frac{\partial Y_2}{\partial \boldsymbol{\omega}} & \frac{\partial Y_2}{\partial b} & \frac{\partial Y_2}{\partial \lambda_1} & \frac{\partial Y_2}{\partial \lambda_2} & \frac{\partial Y_2}{\partial \boldsymbol{y}_i} \\ \frac{\partial Y_3}{\partial \boldsymbol{\omega}} & \frac{\partial Y_3}{\partial b} & \frac{\partial Y_3}{\partial \lambda_1} & \frac{\partial Y_3}{\partial \lambda_2} & \frac{\partial Y_3}{\partial \boldsymbol{y}_i} \\ \frac{\partial Y_4}{\partial \boldsymbol{\omega}} & \frac{\partial Y_4}{\partial b} & \frac{\partial Y_4}{\partial \lambda_1} & \frac{\partial Y_4}{\partial \lambda_2} & \frac{\partial Y_4}{\partial \boldsymbol{y}_i} \\ \frac{\partial \boldsymbol{Y_5}}{\partial \boldsymbol{\omega}} & \frac{\partial \boldsymbol{Y_5}}{\partial b} & \frac{\partial \boldsymbol{Y_5}}{\partial \lambda_1} & \frac{\partial \boldsymbol{Y_5}}{\partial \lambda_2} & \frac{\partial \boldsymbol{Y_5}}{\partial \boldsymbol{y}_i} \end{bmatrix} \begin{bmatrix} \boldsymbol{\omega} \\ b \\ \lambda_1 \\ \lambda_2 \\ \boldsymbol{y}_i \end{bmatrix} = \begin{bmatrix} \boldsymbol{Y_1} \\ Y_2 \\ Y_3 \\ Y_4 \\ \boldsymbol{Y_5} \end{bmatrix} \tag{60}$$

Fill the results of the Jacobian matrix calculation into the equation matrix:

$$\begin{bmatrix} \frac{\partial \boldsymbol{Y_1}}{\partial \boldsymbol{\omega}} & \gamma(\boldsymbol{\varphi}'_{t_i})^T\boldsymbol{I} & (\boldsymbol{\varphi}'_1)^T & (\boldsymbol{\varphi}'_n)^T & \frac{\partial Y_1}{\partial \boldsymbol{y}_i} \\ \gamma\boldsymbol{I}^T\boldsymbol{\varphi}'_{t_i} & \gamma\boldsymbol{I}^T\boldsymbol{I} & 1 & 1 & \frac{\partial Y_2}{\partial \boldsymbol{y}_i} \\ \boldsymbol{\varphi}'_1 & 1 & 0 & 0 & \frac{\partial Y_3}{\partial \boldsymbol{y}_i} \\ \boldsymbol{\varphi}'_n & 1 & 0 & 0 & \frac{\partial Y_4}{\partial \boldsymbol{y}_i} \\ \frac{\partial \boldsymbol{Y_5}}{\partial \boldsymbol{\omega}} & -\gamma\boldsymbol{I} & \boldsymbol{0} & \boldsymbol{0} & \frac{\partial \boldsymbol{Y_5}}{\partial \boldsymbol{y}_i} \end{bmatrix} \begin{bmatrix} \boldsymbol{\omega} \\ b \\ \lambda_1 \\ \lambda_2 \\ \boldsymbol{y}_i \end{bmatrix} = \begin{bmatrix} \boldsymbol{Y_1} \\ Y_2 \\ Y_3 \\ Y_4 \\ \boldsymbol{Y_5} \end{bmatrix} \tag{61}$$

where

$$\frac{\partial \boldsymbol{Y}1}{\partial \boldsymbol{\omega}} = \boldsymbol{E} + \gamma(\boldsymbol{\varphi}''_{t_i})^T[\boldsymbol{\varphi}''_{t_i} - 4\operatorname{diag}\left[\frac{\boldsymbol{\varphi}'_{t_i}\boldsymbol{\omega}}{\boldsymbol{y}_i}\right]\boldsymbol{\varphi}'_{t_i}] - 4\gamma(\boldsymbol{\varphi}'_{t_i})^T\left[\operatorname{diag}\left[\frac{\boldsymbol{\varphi}'_{t_i}\boldsymbol{\omega}}{\boldsymbol{y}_i}\right]\left[\boldsymbol{\varphi}''_{t_i} - 4\operatorname{diag}\left[\frac{\boldsymbol{\varphi}'_{t_i}\boldsymbol{\omega}}{\boldsymbol{y}_i}\right]\boldsymbol{\varphi}'_{t_i}\right]\right]$$
$$+ \operatorname{diag}\left[[\boldsymbol{\varphi}''_{t_i}\boldsymbol{\omega} - \frac{2(\boldsymbol{\varphi}'_{t_i}\boldsymbol{\omega})^2}{\boldsymbol{y}_i} + \boldsymbol{y}_i]\operatorname{diag}\left[\frac{1}{\boldsymbol{y}_i}\right]\boldsymbol{\varphi}'_{t_i}\right] + \gamma(\boldsymbol{\varphi}_{t_i})^T\boldsymbol{\varphi}_{t_i} \tag{62}$$

$$\frac{\partial \boldsymbol{Y}1}{\partial \boldsymbol{y}_i} = \gamma\boldsymbol{\varphi}''_{t_i}{}^T\operatorname{diag}\left[\frac{2(\boldsymbol{\varphi}'_{t_i}\boldsymbol{\omega})^2}{\boldsymbol{y}_i^2} + 1\right]$$
$$- 4\gamma\boldsymbol{\varphi}'_{t_i}{}^T\operatorname{diag}\left[-\frac{\boldsymbol{\varphi}'_{t_i}\boldsymbol{\omega} \odot \left[\boldsymbol{\varphi}''_{t_i}\boldsymbol{\omega} - \frac{2(\boldsymbol{\varphi}'_{t_i}\boldsymbol{\omega})^2}{\boldsymbol{y}_i} + \boldsymbol{y}_i\right]}{\boldsymbol{y}_i^2} + \frac{\boldsymbol{\varphi}'_{t_i}\boldsymbol{\omega}}{\boldsymbol{y}_i} \odot \left[\frac{2(\boldsymbol{\varphi}'_{t_i}\boldsymbol{\omega})^2}{\boldsymbol{y}_i} + 1\right]\right] - \gamma\boldsymbol{\varphi}_{t_i}{}^T$$
$$\tag{63}$$

$$\frac{\partial \boldsymbol{Y}5}{\partial \boldsymbol{\omega}} = \gamma \left[ \operatorname{diag} \left[ \frac{2(\boldsymbol{\varphi}'_{t_i}\boldsymbol{\omega})^2}{\boldsymbol{y}_i} + 1 \right] \left[ \boldsymbol{\varphi}''_{t_i} - 4 \operatorname{diag} \left[ \frac{\boldsymbol{\varphi}'_{t_i}\boldsymbol{\omega}}{\boldsymbol{y}_i} \right] \boldsymbol{\varphi}'_{t_i} \right] \right]$$
$$+ \gamma \left[ \operatorname{diag} \left[ \boldsymbol{\varphi}''_{t_i}\boldsymbol{\omega} - \frac{2(\boldsymbol{\varphi}'_{t_i}\boldsymbol{\omega})^2}{\boldsymbol{y}_i} + \boldsymbol{y}_i \right] 4 \operatorname{diag} \left[ \frac{\boldsymbol{\varphi}'_{t_i}\boldsymbol{\omega}}{\boldsymbol{y}_i} \right] \boldsymbol{\varphi}'_{t_i} \right] - \gamma \boldsymbol{\varphi}_{t_i} \qquad (64)$$

$$\frac{\partial \boldsymbol{Y}5}{\partial \boldsymbol{y}_i} = \gamma \operatorname{diag} \left[ -\frac{2(\boldsymbol{\varphi}'_{t_i}\boldsymbol{\omega})^2}{\boldsymbol{y}_i^2} \left[ \boldsymbol{\varphi}''_{t_i}\boldsymbol{\omega} - \frac{2(\boldsymbol{\varphi}'_{t_i}\boldsymbol{\omega})^2}{\boldsymbol{y}_i} + \boldsymbol{y}_i \right] + \left[ \frac{2(\boldsymbol{\varphi}'_{t_i}\boldsymbol{\omega})^2}{\boldsymbol{y}_i} + 1 \right] \left[ \frac{2(\boldsymbol{\varphi}'_{t_i}\boldsymbol{\omega})^2}{\boldsymbol{y}_i^2} + 1 \right] + 1 \right]$$
$$(65)$$

Use Newton's iteration to solve for $\boldsymbol{\omega}$ and $b$, and then construct the prediction equation in the original space: $\hat{y}(t) = \boldsymbol{\omega}^T \boldsymbol{\varphi}(t) + b$.

## B.3 RBF kernel justification, Sampling for Nyström landmarks and PINN baseline comparison

Table 4: Comparative performance evaluation of Linear, Polynomial, and RBF Kernels for NLS-SVMs in Solving ODEs

| ODEs | Model | MAE | RMSE | $\|y - \hat{y}\|_\infty$ |
|---|---|---|---|---|
| *Problem 1* | Linear | $1.70 \times 10^0$ | $2.00 \times 10^0$ | $5.05 \times 10^0$ |
| | Poly | $1.92 \times 10^{-2}$ | $2.45 \times 10^{-2}$ | $1.29 \times 10^{-1}$ |
| | RBF | $7.94 \times 10^{-4}$ | $9.47 \times 10^{-4}$ | $1.87 \times 10^{-3}$ |
| *Problem 2* | Linear | $1.54 \times 10^0$ | $2.15 \times 10^0$ | $1.13 \times 10^1$ |
| | Poly | $3.37 \times 10^{-1}$ | $3.91 \times 10^{-1}$ | $7.44 \times 10^{-1}$ |
| | RBF | $3.16 \times 10^{-6}$ | $3.77 \times 10^{-6}$ | $9.75 \times 10^{-6}$ |
| *Problem 3* | Linear | $6.34 \times 10^{-1}$ | $7.15 \times 10^{-1}$ | $1.24 \times 10^0$ |
| | Poly | $1.19 \times 10^{-1}$ | $1.43 \times 10^{-1}$ | $4.80 \times 10^{-1}$ |
| | RBF | $2.06 \times 10^{-8}$ | $2.77 \times 10^{-8}$ | $1.40 \times 10^{-7}$ |
| *Problem 4* | Linear | $3.29 \times 10^0$ | $4.67 \times 10^0$ | $1.24 \times 10^1$ |
| | Poly | $3.16 \times 10^0$ | $4.45 \times 10^0$ | $1.16 \times 10^1$ |
| | RBF | $5.47 \times 10^{-4}$ | $6.92 \times 10^{-4}$ | $1.50 \times 10^{-3}$ |
| *Problem 5* | Linear | $1.07 \times 10^0$ | $1.44 \times 10^0$ | $5.82 \times 10^0$ |
| | Poly | $1.53 \times 10^{-3}$ | $1.76 \times 10^{-3}$ | $2.84 \times 10^{-3}$ |
| | RBF | $8.79 \times 10^{-4}$ | $1.07 \times 10^{-3}$ | $1.67 \times 10^{-3}$ |
| *Problem 6* | Linear | $4.01 \times 10^0$ | $5.05 \times 10^0$ | $1.67 \times 10^1$ |
| | Poly | $9.67 \times 10^{-4}$ | $1.10 \times 10^{-3}$ | $1.85 \times 10^{-3}$ |
| | RBF | $8.89 \times 10^{-4}$ | $1.01 \times 10^{-3}$ | $1.70 \times 10^{-3}$ |
| *Problem 7* | Linear | $4.68 \times 10^{-1}$ | $6.05 \times 10^{-1}$ | $1.56 \times 10^0$ |
| | Poly | $3.82 \times 10^{-1}$ | $5.52 \times 10^{-1}$ | $1.47 \times 10^0$ |
| | RBF | $1.43 \times 10^{-3}$ | $1.67 \times 10^{-3}$ | $3.02 \times 10^{-3}$ |
| *Problem 8* | Linear | $4.30 \times 10^{-1}$ | $5.23 \times 10^{-1}$ | $2.01 \times 10^0$ |
| | Poly | $3.42 \times 10^{-1}$ | $3.93 \times 10^{-1}$ | $6.70 \times 10^{-1}$ |
| | RBF | $5.66 \times 10^{-5}$ | $7.05 \times 10^{-5}$ | $2.23 \times 10^{-4}$ |
| *Problem 9* | Linear | $1.91 \times 10^0$ | $2.60 \times 10^0$ | $1.59 \times 10^1$ |
| | Poly | $8.05 \times 10^{-1}$ | $9.67 \times 10^{-1}$ | $1.73 \times 10^0$ |
| | RBF | $5.74 \times 10^{-5}$ | $6.92 \times 10^{-5}$ | $2.21 \times 10^{-4}$ |
| *Problem 10* | Linear | $1.17 \times 10^0$ | $1.61 \times 10^0$ | $8.50 \times 10^0$ |
| | Poly | $5.23 \times 10^{-1}$ | $6.25 \times 10^{-1}$ | $1.36 \times 10^0$ |
| | RBF | $4.29 \times 10^{-8}$ | $6.06 \times 10^{-8}$ | $3.09 \times 10^{-8}$ |
| *Problem 11* | Linear | $9.62 \times 10^{-2}$ | $1.27 \times 10^{-1}$ | $5.79 \times 10^{-1}$ |
| | Poly | $7.13 \times 10^{-4}$ | $8.90 \times 10^{-4}$ | $1.60 \times 10^{-3}$ |
| | RBF | $2.36 \times 10^{-9}$ | $2.97 \times 10^{-9}$ | $9.43 \times 10^{-9}$ |

| ODEs | Model | MAE | RMSE | $\|y - \hat{y}\|_\infty$ |
|---|---|---|---|---|
| *Problem* 12 | Linear | $9.75 \times 10^{-1}$ | $1.27 \times 10^{0}$ | $5.54 \times 10^{0}$ |
| | Poly | $2.75 \times 10^{-3}$ | $3.24 \times 10^{-3}$ | $5.02 \times 10^{-3}$ |
| | RBF | $3.15 \times 10^{-9}$ | $3.94 \times 10^{-9}$ | $1.31 \times 10^{-8}$ |
| *Problem* 13 | Linear | $4.55 \times 10^{0}$ | $6.58 \times 10^{0}$ | $2.84 \times 10^{1}$ |
| | Poly | $1.27 \times 10^{-3}$ | $1.59 \times 10^{-3}$ | $2.91 \times 10^{-3}$ |
| | RBF | $3.48 \times 10^{-3}$ | $4.36 \times 10^{-3}$ | $1.45 \times 10^{-6}$ |
| *Problem* 14 | Linear | $2.12 \times 10^{2}$ | $2.63 \times 10^{2}$ | $7.68 \times 10^{2}$ |
| | Poly | $2.38 \times 10^{-2}$ | $2.82 \times 10^{-2}$ | $4.42 \times 10^{-2}$ |
| | RBF | $1.17 \times 10^{-6}$ | $1.57 \times 10^{-6}$ | $3.85 \times 10^{-6}$ |
| *Problem* 15 | Linear | $1.08 \times 10^{-1}$ | $1.22 \times 10^{-1}$ | $1.76 \times 10^{-1}$ |
| | Poly | $2.93 \times 10^{-1}$ | $3.30 \times 10^{-1}$ | $7.96 \times 10^{-1}$ |
| | RBF | $1.68 \times 10^{-7}$ | $1.98 \times 10^{-7}$ | $3.46 \times 10^{-7}$ |
| *Problem* 16 | Linear | $4.27 \times 10^{15}$ | $6.18 \times 10^{15}$ | $3.53 \times 10^{16}$ |
| | Poly | $1.67 \times 10^{-1}$ | $1.92 \times 10^{-1}$ | $2.86 \times 10^{-1}$ |
| | RBF | $1.72 \times 10^{-5}$ | $2.06 \times 10^{-5}$ | $5.07 \times 10^{-5}$ |

Table 5: Comparative performance evaluation of NLS-SVMs utilizing Equidistant, Random, and Leverage-based sampling strategies for solving ODEs

| ODEs | Model | MAE | RMSE | $\|y - \hat{y}\|_\infty$ | Time/s |
|---|---|---|---|---|---|
| *Problem* 1 | Random | $8.70 \times 10^{-4}$ | $1.06 \times 10^{-3}$ | $3.06 \times 10^{-3}$ | 0.52 |
| | Leverage score | $7.94 \times 10^{-4}$ | $9.47 \times 10^{-4}$ | $1.87 \times 10^{-3}$ | 0.64 |
| | Equidistant | $7.94 \times 10^{-4}$ | $9.47 \times 10^{-4}$ | $1.87 \times 10^{-3}$ | 0.55 |
| *Problem* 2 | Random | $3.41 \times 10^{-6}$ | $4.06 \times 10^{-6}$ | $8.14 \times 10^{-6}$ | 0.52 |
| | Leverage score | $3.22 \times 10^{-6}$ | $3.82 \times 10^{-6}$ | $7.51 \times 10^{-6}$ | 0.57 |
| | Equidistant | $3.16 \times 10^{-6}$ | $3.77 \times 10^{-6}$ | $9.75 \times 10^{-6}$ | 0.55 |
| *Problem* 3 | Random | $1.31 \times 10^{-7}$ | $1.75 \times 10^{-7}$ | $8.99 \times 10^{-7}$ | 0.53 |
| | Leverage score | $3.12 \times 10^{-8}$ | $4.02 \times 10^{-8}$ | $1.79 \times 10^{-7}$ | 0.53 |
| | Equidistant | $2.06 \times 10^{-8}$ | $2.77 \times 10^{-8}$ | $1.40 \times 10^{-7}$ | 0.56 |
| *Problem* 4 | Random | $5.47 \times 10^{-4}$ | $6.92 \times 10^{-4}$ | $1.50 \times 10^{-3}$ | 9.53 |
| | Leverage score | $5.47 \times 10^{-4}$ | $6.92 \times 10^{-4}$ | $1.50 \times 10^{-3}$ | 14.1 |
| | Equidistant | $5.47 \times 10^{-4}$ | $6.92 \times 10^{-4}$ | $1.50 \times 10^{-3}$ | 10.2 |
| *Problem* 5 | Random | $8.79 \times 10^{-4}$ | $1.07 \times 10^{-3}$ | $1.67 \times 10^{-3}$ | 1.07 |
| | Leverage score | $8.79 \times 10^{-4}$ | $1.07 \times 10^{-3}$ | $1.67 \times 10^{-3}$ | 1.02 |
| | Equidistant | $8.79 \times 10^{-4}$ | $1.07 \times 10^{-3}$ | $1.67 \times 10^{-3}$ | 0.97 |
| *Problem* 6 | Random | $8.89 \times 10^{-4}$ | $1.01 \times 10^{-3}$ | $1.70 \times 10^{-3}$ | 0.80 |
| | Leverage score | $8.88 \times 10^{-4}$ | $1.01 \times 10^{-3}$ | $1.70 \times 10^{-3}$ | 1.45 |
| | Equidistant | $8.89 \times 10^{-4}$ | $1.01 \times 10^{-3}$ | $1.70 \times 10^{-3}$ | 0.69 |
| *Problem* 7 | Random | $1.43 \times 10^{-3}$ | $1.66 \times 10^{-3}$ | $3.02 \times 10^{-3}$ | 0.45 |
| | Leverage score | $1.43 \times 10^{-3}$ | $1.66 \times 10^{-3}$ | $3.02 \times 10^{-3}$ | 0.54 |
| | Equidistant | $1.43 \times 10^{-3}$ | $1.67 \times 10^{-3}$ | $3.02 \times 10^{-3}$ | 0.50 |
| *Problem* 8 | Random | $9.92 \times 10^{-5}$ | $1.29 \times 10^{-4}$ | $4.87 \times 10^{-4}$ | 0.44 |
| | Leverage score | $6.90 \times 10^{-5}$ | $8.79 \times 10^{-5}$ | $3.02 \times 10^{-4}$ | 0.46 |
| | Equidistant | $5.66 \times 10^{-5}$ | $7.05 \times 10^{-5}$ | $2.23 \times 10^{-4}$ | 0.44 |
| *Problem* 9 | Random | $8.43 \times 10^{-5}$ | $1.15 \times 10^{-4}$ | $3.88 \times 10^{-4}$ | 0.43 |
| | Leverage score | $6.07 \times 10^{-5}$ | $7.54 \times 10^{-5}$ | $2.45 \times 10^{-4}$ | 0.45 |
| | Equidistant | $5.74 \times 10^{-5}$ | $6.92 \times 10^{-5}$ | $2.21 \times 10^{-4}$ | 0.47 |
| *Problem* 10 | Random | $1.63 \times 10^{-7}$ | $2.25 \times 10^{-7}$ | $1.05 \times 10^{-6}$ | 0.44 |
| | Leverage score | $2.29 \times 10^{-7}$ | $2.65 \times 10^{-7}$ | $8.12 \times 10^{-7}$ | 0.43 |
| | Equidistant | $4.29 \times 10^{-8}$ | $6.06 \times 10^{-8}$ | $3.09 \times 10^{-8}$ | 0.49 |
| *Problem* 11 | Random | $9.02 \times 10^{-9}$ | $1.08 \times 10^{-8}$ | $2.90 \times 10^{-8}$ | 0.43 |
| | Leverage score | $2.46 \times 10^{-9}$ | $3.06 \times 10^{-9}$ | $8.66 \times 10^{-9}$ | 0.42 |

| ODEs | Model | MAE | RMSE | $\|y - \hat{y}\|_\infty$ | Time/s |
|---|---|---|---|---|---|
| | Equidistant | $2.36\times10^{-9}$ | $2.97\times10^{-9}$ | $9.43\times10^{-9}$ | 0.45 |
| *Problem* 12 | Random | $2.87\times10^{-8}$ | $3.59\times10^{-8}$ | $1.13\times10^{-7}$ | 0.42 |
| | Leverage score | $1.35\times10^{-7}$ | $1.68\times10^{-7}$ | $6.31\times10^{-7}$ | 0.42 |
| | Equidistant | $3.15\times10^{-9}$ | $3.94\times10^{-9}$ | $1.31\times10^{-8}$ | 0.46 |
| *Problem* 13 | Random | $8.76\times10^{-7}$ | $1.13\times10^{-6}$ | $4.29\times10^{-6}$ | 0.42 |
| | Leverage score | $5.48\times10^{-7}$ | $6.89\times10^{-7}$ | $2.28\times10^{-6}$ | 0.42 |
| | Equidistant | $3.48\times10^{-7}$ | $4.36\times10^{-7}$ | $1.45\times10^{-6}$ | 0.48 |
| *Problem* 14 | Random | $2.07\times10^{-6}$ | $2.67\times10^{-6}$ | $7.13\times10^{-6}$ | 1.38 |
| | Leverage score | $1.95\times10^{-6}$ | $2.63\times10^{-6}$ | $6.66\times10^{-6}$ | 1.39 |
| | Equidistant | $1.17\times10^{-6}$ | $1.57\times10^{-6}$ | $3.85\times10^{-6}$ | 1.39 |
| *Problem* 15 | Random | $1.99\times10^{-7}$ | $2.35\times10^{-7}$ | $4.40\times10^{-7}$ | 3.54 |
| | Leverage score | $1.73\times10^{-7}$ | $2.04\times10^{-7}$ | $3.60\times10^{-7}$ | 3.19 |
| | Equidistant | $1.68\times10^{-7}$ | $1.98\times10^{-7}$ | $3.46\times10^{-7}$ | 3.25 |
| *Problem* 16 | Random | $3.57\times10^{-4}$ | $4.42\times10^{-4}$ | $1.54\times10^{-3}$ | 0.43 |
| | Leverage score | $2.09\times10^{-5}$ | $2.57\times10^{-5}$ | $8.33\times10^{-5}$ | 0.45 |
| | Equidistant | $1.72\times10^{-5}$ | $2.06\times10^{-5}$ | $5.07\times10^{-5}$ | 0.42 |

Table 6: Comparative performance evaluation of NLS-SVMs, RK4, and EAB methods for solving ODEs

| ODEs | Model | MAE | RMSE | Train/s | Predict/s |
|---|---|---|---|---|---|
| *Problem* 1 | NLS-SVMs | $7.94\times10^{-4}$ | $9.47\times10^{-4}$ | 0.550 | 0.001 |
| | RK4 | $2.20\times10^{-5}$ | $1.80\times10^{-5}$ | 0.011 | 0.011 |
| | EAB | $3.23\times10^{-4}$ | $1.74\times10^{-4}$ | 0.013 | 0.013 |
| *Problem* 2 | NLS-SVMs | $3.16\times10^{-6}$ | $3.77\times10^{-6}$ | 0.480 | 0.002 |
| | RK4 | $1.42\times10^{-7}$ | $8.96\times10^{-8}$ | 0.019 | 0.019 |
| | EAB | $2.93\times10^{-3}$ | $1.36\times10^{-3}$ | 0.021 | 0.021 |
| *Problem* 3 | NLS-SVMs | $2.06\times10^{-8}$ | $2.77\times10^{-8}$ | 0.460 | 0.001 |
| | RK4 | $3.87\times10^{-7}$ | $3.41\times10^{-7}$ | 0.880 | 0.880 |
| | EAB | $4.97\times10^{-8}$ | $2.90\times10^{-8}$ | 0.030 | 0.030 |
| *Problem* 4 | NLS-SVMs | $5.47\times10^{-4}$ | $6.92\times10^{-4}$ | 14.530 | 0.001 |
| | RK4 | $3.79\times10^{-6}$ | $2.78\times10^{-6}$ | 0.022 | 0.022 |
| | EAB | $9.41\times10^{-4}$ | $6.45\times10^{-4}$ | 0.016 | 0.016 |
| *Problem* 5 | NLS-SVMs | $8.79\times10^{-4}$ | $1.07\times10^{-3}$ | 4.590 | 0.001 |
| | RK4 | $3.58\times10^{-7}$ | $3.38\times10^{-7}$ | 0.013 | 0.013 |
| | EAB | $1.79\times10^{-4}$ | $1.41\times10^{-4}$ | 0.011 | 0.011 |
| *Problem* 6 | NLS-SVMs | $8.89\times10^{-4}$ | $1.01\times10^{-3}$ | 1.230 | 0.001 |
| | RK4 | $7.94\times10^{-2}$ | $5.61\times10^{-2}$ | 0.021 | 0.021 |
| | EAB | $1.28\times10^{-6}$ | $1.11\times10^{-6}$ | 0.036 | 0.036 |
| *Problem* 7 | NLS-SVMs | $1.43\times10^{-3}$ | $1.67\times10^{-3}$ | 0.150 | 0.001 |
| | RK4 | $1.66\times10^{-3}$ | $1.42\times10^{-3}$ | 0.012 | 0.012 |
| | EAB | $4.01\times10^{-3}$ | $3.49\times10^{-3}$ | 0.013 | 0.013 |
| *Problem* 8 | NLS-SVMs | $5.66\times10^{-5}$ | $7.05\times10^{-5}$ | 0.066 | 0.001 |
| | RK4 | $3.61\times10^{-3}$ | $2.81\times10^{-3}$ | 0.023 | 0.023 |
| | EAB | $1.26\times10^{-1}$ | $7.32\times10^{-2}$ | 0.018 | 0.018 |
| *Problem* 9 | NLS-SVMs | $5.74\times10^{-5}$ | $6.92\times10^{-5}$ | 0.170 | 0.001 |
| | RK4 | $8.21\times10^{-6}$ | $3.08\times10^{-6}$ | 0.022 | 0.022 |
| | EAB | $2.32\times10^{-5}$ | $1.11\times10^{-5}$ | 0.026 | 0.026 |
| *Problem* 10 | NLS-SVMs | $4.29\times10^{-8}$ | $6.06\times10^{-8}$ | 0.075 | 0.001 |
| | RK4 | $1.00\times10^{-6}$ | $1.00\times10^{-6}$ | 0.029 | 0.029 |
| | EAB | $5.48\times10^{-5}$ | $4.62\times10^{-5}$ | 0.018 | 0.018 |

| ODEs | Model | MAE | RMSE | Train/s | Predict/s |
|---|---|---|---|---|---|
| *Problem* 11 | NLS-SVMs | $2.36\times10^{-9}$ | $2.97\times10^{-9}$ | 0.100 | 0.001 |
| | RK4 | $3.11\times10^{-6}$ | $2.87\times10^{-6}$ | 0.010 | 0.010 |
| | EAB | $1.19\times10^{-2}$ | $6.86\times10^{-3}$ | 0.020 | 0.020 |
| *Problem* 12 | NLS-SVMs | $3.15\times10^{-9}$ | $3.94\times10^{-9}$ | 0.043 | 0.001 |
| | RK4 | $5.75\times10^{-7}$ | $5.20\times10^{-7}$ | 0.016 | 0.016 |
| | EAB | $5.38\times10^{-6}$ | $4.61\times10^{-6}$ | 0.023 | 0.023 |
| *Problem* 13 | NLS-SVMs | $3.48\times10^{-7}$ | $4.36\times10^{-7}$ | 0.098 | 0.001 |
| | RK4 | $1.57\times10^{-6}$ | $1.21\times10^{-6}$ | 0.011 | 0.011 |
| | EAB | $4.79\times10^{-7}$ | $4.15\times10^{-7}$ | 0.024 | 0.024 |
| *Problem* 14 | NLS-SVMs | $1.17\times10^{-6}$ | $1.57\times10^{-6}$ | 1.390 | 0.001 |
| | RK4 | $9.42\times10^{-5}$ | $8.18\times10^{-5}$ | 0.025 | 0.025 |
| | EAB | $9.75\times10^{-4}$ | $7.57\times10^{-4}$ | 0.019 | 0.019 |
| *Problem* 15 | NLS-SVMs | $1.68\times10^{-7}$ | $1.98\times10^{-7}$ | 1.690 | 0.001 |
| | RK4 | $1.04\times10^{-4}$ | $9.10\times10^{-5}$ | 0.023 | 0.023 |
| | EAB | $1.35\times10^{-4}$ | $9.89\times10^{-5}$ | 0.020 | 0.020 |
| *Problem* 16 | NLS-SVMs | $1.72\times10^{-5}$ | $2.06\times10^{-5}$ | 0.410 | 0.001 |
| | RK4 | $3.75\times10^{-4}$ | $4.89\times10^{-4}$ | 0.019 | 0.019 |
| | EAB | $3.57\times10^{-3}$ | $4.36\times10^{-3}$ | 0.023 | 0.023 |

Table 7: Comparative performance evaluation of Vanilla PINN versus Fourier Feature-enhanced PINN for solving ODEs

| ODEs | Model | MAE | RMSE | $\|y-\hat{y}\|_\infty$ | Time/s |
|---|---|---|---|---|---|
| *Problem* 1 | Vanilla PINN | $4.45\times10^{-3}$ | $5.05\times10^{-3}$ | $9.91\times10^{-3}$ | 672.00 |
| | Fourier Feature PINN | $1.02\times10^{-2}$ | $1.55\times10^{-2}$ | $3.41\times10^{-2}$ | 549.00 |
| *Problem* 2 | Vanilla PINN | $2.03\times10^{-3}$ | $2.09\times10^{-3}$ | $3.26\times10^{-3}$ | 403.00 |
| | Fourier Feature PINN | $1.33\times10^{-5}$ | $1.64\times10^{-5}$ | $6.43\times10^{-5}$ | 762.00 |
| *Problem* 3 | Vanilla PINN | $1.82\times10^{-1}$ | $2.88\times10^{-1}$ | $1.03\times10^{0}$ | 1209.00 |
| | Fourier Feature PINN | $8.51\times10^{-4}$ | $1.15\times10^{-3}$ | $3.72\times10^{-3}$ | 1556.00 |
| *Problem* 4 | Vanilla PINN | $5.35\times10^{-1}$ | $7.21\times10^{-1}$ | $1.81\times10^{0}$ | 496.00 |
| | Fourier Feature PINN | $5.57\times10^{0}$ | $6.67\times10^{0}$ | $1.50\times10^{1}$ | 956.00 |
| *Problem* 5 | Vanilla PINN | $1.03\times10^{-3}$ | $1.04\times10^{-3}$ | $1.13\times10^{-3}$ | 166.00 |
| | Fourier Feature PINN | $1.90\times10^{-3}$ | $2.26\times10^{-3}$ | $4.41\times10^{-3}$ | 166.00 |
| *Problem* 6 | Vanilla PINN | $1.47\times10^{-2}$ | $1.76\times10^{-2}$ | $3.57\times10^{-2}$ | 75.60 |
| | Fourier Feature PINN | $2.32\times10^{-2}$ | $2.47\times10^{-2}$ | $3.86\times10^{1}$ | 217.00 |
| *Problem* 7 | Vanilla PINN | $6.58\times10^{-2}$ | $8.06\times10^{-2}$ | $1.63\times10^{-1}$ | 1021.00 |
| | Fourier Feature PINN | $7.88\times10^{-1}$ | $8.35\times10^{-1}$ | $1.42\times10^{0}$ | 1279.00 |
| *Problem* 8 | Vanilla PINN | $5.23\times10^{-3}$ | $5.92\times10^{-3}$ | $1.04\times10^{-2}$ | 1392.00 |
| | Fourier Feature PINN | $1.13\times10^{-1}$ | $1.26\times10^{-1}$ | $1.87\times10^{-1}$ | 2170.00 |
| *Problem* 9 | Vanilla PINN | $8.66\times10^{-5}$ | $9.96\times10^{-5}$ | $1.89\times10^{-4}$ | 776.00 |
| | Fourier Feature PINN | $8.35\times10^{-1}$ | $8.72\times10^{-1}$ | $9.55\times10^{-1}$ | 1534.00 |
| *Problem* 10 | Vanilla PINN | $3.12\times10^{-2}$ | $4.07\times10^{-2}$ | $8.62\times10^{-2}$ | 636.00 |
| | Fourier Feature PINN | $2.61\times10^{-1}$ | $3.31\times10^{-1}$ | $7.02\times10^{-1}$ | 2364.00 |
| *Problem* 11 | Vanilla PINN | $1.62\times10^{-5}$ | $1.89\times10^{-5}$ | $3.32\times10^{-5}$ | 817.00 |
| | Fourier Feature PINN | $2.81\times10^{-8}$ | $3.48\times10^{-8}$ | $1.16\times10^{-7}$ | 16.30 |
| *Problem* 12 | Vanilla PINN | $2.28\times10^{-5}$ | $2.29\times10^{-5}$ | $2.60\times10^{-5}$ | 969.00 |
| | Fourier Feature PINN | $1.56\times10^{-8}$ | $1.93\times10^{-8}$ | $5.91\times10^{-8}$ | 16.70 |
| *Problem* 13 | Vanilla PINN | $4.22\times10^{-3}$ | $5.72\times10^{-3}$ | $1.05\times10^{-2}$ | 1475.00 |
| | Fourier Feature PINN | $2.77\times10^{-7}$ | $3.42\times10^{-7}$ | $9.56\times10^{-7}$ | 33.60 |
| *Problem* 14 | Vanilla PINN | $8.22\times10^{-6}$ | $8.77\times10^{-6}$ | $1.34\times10^{-5}$ | 112.40 |

| ODEs | Model | MAE | RMSE | $\|y - \hat{y}\|_\infty$ | Time/s |
|---|---|---|---|---|---|
| | Fourier Feature PINN | $7.38\times10^{-7}$ | $9.17\times10^{-7}$ | $2.02\times10^{-6}$ | 36.00 |
| *Problem* 15 | Vanilla PINN | $2.42\times10^{-3}$ | $2.62\times10^{-3}$ | $3.62\times10^{-3}$ | 230.00 |
| | Fourier Feature PINN | $2.14\times10^{-8}$ | $2.66\times10^{-8}$ | $6.68\times10^{-7}$ | 85.40 |
| *Problem* 16 | Vanilla PINN | $1.24\times10^{-2}$ | $1.35\times10^{-2}$ | $1.80\times10^{-2}$ | 2699.00 |
| | Fourier Feature PINN | $7.51\times10^{0}$ | $8.74\times10^{0}$ | $2.38\times10^{1}$ | 697.00 |

## B.4 Convergence analysis of iterative solver

Table 8: Convergence behavior and computational efficiency of NLS-SVMs across varying iteration counts for solving ODEs

| ODEs | Index | 50 | 100 | 200 | 300 | 400 | 500 |
|---|---|---|---|---|---|---|---|
| *Problem* 4 | Residuals | $5.1\times10^{-5}$ | $3.9\times10^{-5}$ | $3.5\times10^{-5}$ | $1.6\times10^{-5}$ | $6.4\times10^{-5}$ | $1.1\times10^{-4}$ |
| | RMSE | $6.92\times10^{-4}$ | $6.92\times10^{-4}$ | $6.92\times10^{-4}$ | $6.92\times10^{-4}$ | $6.92\times10^{-4}$ | $6.92\times10^{-4}$ |
| | Time/s | 10.2 | 24.1 | 53.0 | 81.2 | 107.4 | 135.0 |
| *Problem* 5 | Residuals | $5.1\times10^{1}$ | $5.1\times10^{1}$ | $5.1\times10^{1}$ | $5.1\times10^{1}$ | $5.1\times10^{1}$ | $5.1\times10^{1}$ |
| | RMSE | $1.07\times10^{-3}$ | $1.07\times10^{-3}$ | $1.07\times10^{-3}$ | $1.07\times10^{-3}$ | $1.07\times10^{-3}$ | $1.07\times10^{-3}$ |
| | Time/s | 1.8 | 12.0 | 30.5 | 48.8 | 65.5 | 84.0 |
| *Problem* 6 | Residuals | $6.0\times10^{-2}$ | $6.0\times10^{-2}$ | $6.0\times10^{-2}$ | $6.0\times10^{-2}$ | $6.0\times10^{-2}$ | $6.0\times10^{-2}$ |
| | RMSE | $1.01\times10^{-3}$ | $1.01\times10^{-3}$ | $1.01\times10^{-3}$ | $1.01\times10^{-3}$ | $1.01\times10^{-3}$ | $1.01\times10^{-3}$ |
| | Time/s | 0.7 | 0.9 | 1.7 | 2.9 | 5.4 | 7.7 |
| *Problem* 14 | Residuals | $5.1\times10^{6}$ | $5.6\times10^{5}$ | $6.4\times10^{3}$ | $7.3\times10^{1}$ | $8.3\times10^{-1}$ | $9.5\times10^{-3}$ |
| | RMSE | $4.97\times10^{-4}$ | $5.36\times10^{-5}$ | $1.84\times10^{-6}$ | $1.58\times10^{-6}$ | $1.57\times10^{-6}$ | $1.57\times10^{-6}$ |
| | Time/s | 0.6 | 0.7 | 0.8 | 1.0 | 1.3 | 1.4 |
| *Problem* 15 | Residuals | $1.9\times10^{-3}$ | $1.4\times10^{-6}$ | $8.6\times10^{-6}$ | $2.9\times10^{-6}$ | $2.5\times10^{-6}$ | $7.1\times10^{-6}$ |
| | RMSE | $1.98\times10^{-7}$ | $1.98\times10^{-7}$ | $1.98\times10^{-7}$ | $1.98\times10^{-7}$ | $1.98\times10^{-7}$ | $1.98\times10^{-7}$ |
| | Time/s | 0.8 | 0.9 | 1.3 | 2.2 | 3.3 | 6.6 |

## B.5 Hyperparameters for Training

Table 9: Hyperparameter configurations for PINN, LS-SVMs and NLS-SVMs in solving ODEs

| ODEs | Model | Total sample | Sub-sample | Epoch/Iteration | $\sigma^2$ | $\gamma$ |
|---|---|---|---|---|---|---|
| *Problem* 1 | PINN | 1000 | - | 10000 | - | - |
| | LS-SVMs | 1000 | - | - | 10 | $10^7$ |
| | NLS-SVMs | 100 | 150 | - | 10 | $10^7$ |
| *Problem* 2 | PINN | 10000 | - | 10000 | - | - |
| | LS-SVMs | 10000 | - | - | 10 | $10^7$ |
| | NLS-SVMs | 10000 | 50 | - | 10 | $10^7$ |
| *Problem* 3 | PINN | 10000 | - | 20000 | - | - |
| | LS-SVMs | 10000 | - | - | 10 | $10^7$ |
| | NLS-SVMs | 10000 | 50 | - | 10 | $10^7$ |
| *Problem* 4 | PINN | 1000 | - | 10000 | - | - |
| | LS-SVMs | 1000 | - | 50 | 10 | $10^6$ |
| | NLS-SVMs | 1000 | 50 | 50 | 10 | $10^6$ |
| *Problem* 5 | PINN | 600 | - | 10000 | - | - |
| | LS-SVMs | 600 | - | 25 | 8 | $10^6$ |
| | NLS-SVMs | 600 | 20 | 25 | 8 | $10^6$ |
| *Problem* 6 | PINN | 300 | - | 10000 | - | - |
| | LS-SVMs | 300 | - | 50 | 1 | $10^6$ |

| ODEs | Model | Total sample | Sub-sample | Epoch | $\sigma^2$ | $\gamma$ |
|---|---|---|---|---|---|---|
| | NLS-SVMs | 300 | 20 | 50 | 1 | $10^6$ |
| *Problem* 7 | PINN | 1000 | - | 10000 | - | - |
| | LS-SVMs | 1000 | - | - | 1 | $10^7$ |
| | NLS-SVMs | 1000 | 50 | - | 1 | $10^7$ |
| *Problem* 8 | PINN | 1000 | - | 20000 | - | - |
| | LS-SVMs | 1000 | - | - | 1 | $10^7$ |
| | NLS-SVMs | 1000 | 50 | - | 1 | $10^7$ |
| *Problem* 9 | PINN | 1000 | - | 10000 | - | - |
| | LS-SVMs | 1000 | 1000 | - | 1 | $10^7$ |
| | NLS-SVMs | 1000 | 50 | - | 1 | $10^7$ |
| *Problem* 10 | PINN | 1000 | - | 10000 | - | - |
| | LS-SVMs | 1000 | - | - | 1 | $10^7$ |
| | NLS-SVMs | 1000 | 50 | - | 1 | $10^7$ |
| *Problem* 11 | PINN | 1000 | - | 10000 | - | - |
| | LS-SVMs | 1000 | - | - | 1 | $10^7$ |
| | NLS-SVMs | 1000 | 20 | - | 1 | $10^7$ |
| *Problem* 12 | PINN | 1000 | - | 10000 | - | - |
| | LS-SVMs | 1000 | - | - | 1 | $10^7$ |
| | NLS-SVMs | 1000 | 20 | - | 1 | $10^7$ |
| *Problem* 13 | PINN | 1000 | - | 10000 | - | - |
| | LS-SVMs | 1000 | - | - | 1 | $10^7$ |
| | NLS-SVMs | 1000 | 20 | - | 1 | $10^7$ |
| *Problem* 14 | PINN | 200 | - | 10000 | - | - |
| | LS-SVMs | 200 | - | 500 | 1 | $10^7$ |
| | NLS-SVMs | 200 | 10 | 500 | 1 | $10^7$ |
| *Problem* 15 | PINN | 300 | - | 20000 | - | - |
| | LS-SVMs | 300 | - | 400 | 1 | $10^7$ |
| | NLS-SVMs | 300 | 10 | 400 | 1 | $10^7$ |
| *Problem* 16 | PINN | 1000 | - | 10000 | - | - |
| | LS-SVMs | 1000 | - | - | 1 | $10^7$ |
| | NLS-SVMs | 1000 | 20 | - | 1 | $10^7$ |

Table 10: Hyperparameter configuration for NLS-SVMs in large-scale ODEs solving

| ODEs | Model | t | Total sample | Sub-sample | $\sigma^2$ | $\gamma$ |
|---|---|---|---|---|---|---|
| *Problem* 1 | NLS-SVMs | [0,100 ] | 50000 | 200 | 10 | $10^7$ |
| *Problem* 3 | NLS-SVMs | [0,100 ] | 50000 | 200 | 10 | $10^7$ |
| *Problem* 5 | NLS-SVMs | [0,100 ] | 10000 | 100 | 8 | $10^6$ |
| *Problem* 7 | NLS-SVMs | [0,100 ] | 50000 | 100 | 1 | $10^7$ |
| *Problem* 8 | NLS-SVMs | [0,50] | 20000 | 200 | 1 | $10^7$ |
| *Problem* 9 | NLS-SVMs | [0,100 ] | 50000 | 200 | 1 | $10^7$ |
| *Problem* 10 | NLS-SVMs | [0,100 ] | 50000 | 200 | 1 | $10^7$ |
| *Problem* 11 | NLS-SVMs | [0,100 ] | 50000 | 200 | 1 | $10^7$ |
| *Problem* 12 | NLS-SVMs | [0,100 ] | 50000 | 200 | 1 | $10^7$ |

## B.6 Assessment of NLS-SVMs for long-time integration of ODEs

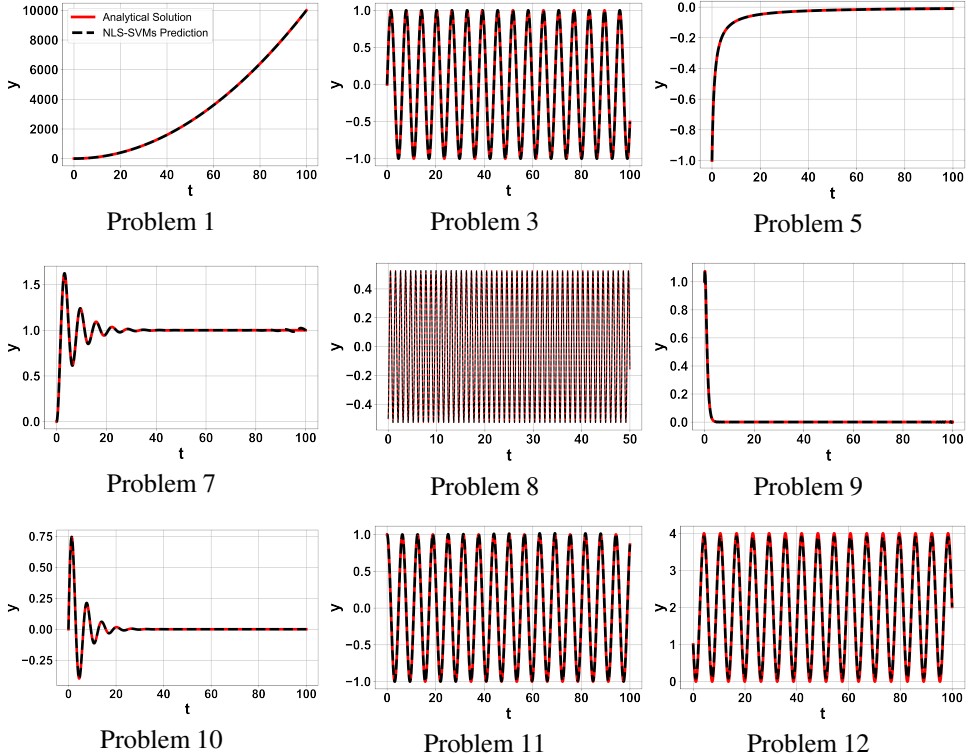

Figure 5: NLS-SVMs predictions against analytical solutions for twelve benchmark problems

## B.7 Performance comparison of PINN, LS-SVMs and NLS-SVMs

Table 11: Comparative performance evaluation of PINN, LS-SVMs, and NLS-SVMs in solving ODEs: accuracy metrics and computational efficiency

| ODEs | Model | $R^2$ | MAE | RMSE | $\|y - \hat{y}\|_\infty$ | Time/s |
|---|---|---|---|---|---|---|
| *Problem* 1 | PINN | 0.99999 | $4.45\times10^{-3}$ | $5.05\times10^{-3}$ | $9.91\times10^{-3}$ | 672.00 |
| | LS-SVMs | 0.99999 | $4.81\times10^{-4}$ | $5.72\times10^{-4}$ | $1.17\times10^{-3}$ | 9.79 |
| | NLS-SVMs | 0.99999 | $7.94\times10^{-4}$ | $9.47\times10^{-4}$ | $1.87\times10^{-3}$ | 0.55 |
| *Problem* 2 | PINN | 0.99989 | $2.03\times10^{-3}$ | $2.09\times10^{-3}$ | $3.26\times10^{-3}$ | 403.00 |
| | LS-SVMs | 0.99999 | $6.87\times10^{-7}$ | $8.55\times10^{-7}$ | $3.07\times10^{-6}$ | 1587.00 |
| | NLS-SVMs | 0.99999 | $3.16\times10^{-6}$ | $3.77\times10^{-6}$ | $9.75\times10^{-6}$ | 0.55 |
| *Problem* 3 | PINN | 0.74911 | $1.82\times10^{-1}$ | $2.88\times10^{-1}$ | $1.03\times10^{0}$ | 1209.00 |
| | LS-SVMs | 0.99999 | $2.55\times10^{-8}$ | $3.23\times10^{-8}$ | $1.36\times10^{-7}$ | 1879.00 |
| | NLS-SVMs | 0.99999 | $2.06\times10^{-8}$ | $2.77\times10^{-8}$ | $1.40\times10^{-7}$ | 0.56 |
| *Problem* 4 | PINN | 0.94554 | $5.35\times10^{-1}$ | $7.21\times10^{-1}$ | $1.81\times10^{0}$ | 496.00 |
| | LS-SVMs | 0.99999 | $5.26\times10^{-5}$ | $7.80\times10^{-5}$ | $2.13\times10^{-4}$ | 3774.00 |
| | NLS-SVMs | 0.99999 | $5.47\times10^{-4}$ | $6.92\times10^{-4}$ | $1.50\times10^{-3}$ | 10.20 |
| *Problem* 5 | PINN | 0.99995 | $1.03\times10^{-3}$ | $1.04\times10^{-3}$ | $1.13\times10^{-3}$ | 166.00 |
| | LS-SVMs | 0.99994 | $8.89\times10^{-4}$ | $1.11\times10^{-3}$ | $1.84\times10^{-3}$ | 9.71 |
| | NLS-SVMs | 0.99994 | $8.79\times10^{-4}$ | $1.07\times10^{-3}$ | $1.67\times10^{-3}$ | 0.97 |
| *Problem* 6 | PINN | 0.89021 | $1.47\times10^{-2}$ | $1.76\times10^{-2}$ | $3.57\times10^{-2}$ | 75.60 |
| | LS-SVMs | 0.99993 | $4.47\times10^{-4}$ | $5.11\times10^{-4}$ | $8.63\times10^{-4}$ | 13.20 |
| | NLS-SVMs | 0.99974 | $8.89\times10^{-4}$ | $1.01\times10^{-3}$ | $1.70\times10^{-3}$ | 0.69 |

Continued on next page

| ODEs | Model | $R^2$ | MAE | RMSE | $\|y - \hat{y}\|_\infty$ | Time/s |
|---|---|---|---|---|---|---|
| *Problem* 7 | PINN | 0.91449 | $6.58\times10^{-2}$ | $8.06\times10^{-2}$ | $1.63\times10^{-1}$ | 1021.00 |
| | LS-SVMs | 0.99997 | $1.43\times10^{-3}$ | $1.66\times10^{-3}$ | $3.02\times10^{-3}$ | 22.90 |
| | NLS-SVMs | 0.99997 | $1.43\times10^{-3}$ | $1.67\times10^{-3}$ | $3.02\times10^{-3}$ | 0.50 |
| *Problem* 8 | PINN | 0.99973 | $5.23\times10^{-3}$ | $5.92\times10^{-3}$ | $1.04\times10^{-2}$ | 1392.00 |
| | LS-SVMs | 0.99999 | $7.14\times10^{-6}$ | $8.93\times10^{-6}$ | $2.86\times10^{-5}$ | 20.80 |
| | NLS-SVMs | 0.99999 | $5.66\times10^{-5}$ | $7.05\times10^{-5}$ | $2.23\times10^{-4}$ | 0.44 |
| *Problem* 9 | PINN | 0.99999 | $8.66\times10^{-5}$ | $9.96\times10^{-5}$ | $1.89\times10^{-4}$ | 776.00 |
| | LS-SVMs | 0.99999 | $1.60\times10^{-5}$ | $1.92\times10^{-5}$ | $6.20\times10^{-5}$ | 20.20 |
| | NLS-SVMs | 0.99999 | $5.74\times10^{-5}$ | $6.92\times10^{-5}$ | $2.21\times10^{-4}$ | 0.47 |
| *Problem* 10 | PINN | 0.98291 | $3.12\times10^{-2}$ | $4.07\times10^{-2}$ | $8.62\times10^{-2}$ | 636.00 |
| | LS-SVMs | 0.99999 | $9.40\times10^{-9}$ | $1.34\times10^{-8}$ | $3.99\times10^{-8}$ | 24.70 |
| | NLS-SVMs | 0.99999 | $4.29\times10^{-8}$ | $6.06\times10^{-8}$ | $3.09\times10^{-8}$ | 0.49 |
| *Problem* 11 | PINN | 0.99999 | $1.62\times10^{-5}$ | $1.89\times10^{-5}$ | $3.32\times10^{-5}$ | 817.00 |
| | LS-SVMs | 0.99999 | $2.81\times10^{-8}$ | $3.48\times10^{-8}$ | $1.16\times10^{-7}$ | 16.30 |
| | NLS-SVMs | 0.99999 | $2.36\times10^{-9}$ | $2.97\times10^{-9}$ | $9.43\times10^{-9}$ | 0.45 |
| *Problem* 12 | PINN | 0.99999 | $2.28\times10^{-5}$ | $2.29\times10^{-5}$ | $2.60\times10^{-5}$ | 969.00 |
| | LS-SVMs | 0.99999 | $1.56\times10^{-8}$ | $1.93\times10^{-8}$ | $5.91\times10^{-8}$ | 16.70 |
| | NLS-SVMs | 0.99999 | $3.15\times10^{-9}$ | $3.94\times10^{-9}$ | $1.31\times10^{-8}$ | 0.46 |
| *Problem* 13 | PINN | 0.99987 | $4.22\times10^{-3}$ | $5.72\times10^{-3}$ | $1.05\times10^{-2}$ | 1475.00 |
| | LS-SVMs | 0.99999 | $2.77\times10^{-7}$ | $3.42\times10^{-7}$ | $9.56\times10^{-7}$ | 33.60 |
| | NLS-SVMs | 0.99999 | $3.48\times10^{-7}$ | $4.36\times10^{-7}$ | $1.45\times10^{-6}$ | 0.48 |
| *Problem* 14 | PINN | 0.99999 | $8.22\times10^{-6}$ | $8.77\times10^{-6}$ | $1.34\times10^{-5}$ | 112.40 |
| | LS-SVMs | 0.99999 | $7.38\times10^{-7}$ | $9.17\times10^{-7}$ | $2.02\times10^{-6}$ | 36.00 |
| | NLS-SVMs | 0.99999 | $1.17\times10^{-6}$ | $1.57\times10^{-6}$ | $3.85\times10^{-6}$ | 1.39 |
| *Problem* 15 | PINN | 0.99774 | $2.42\times10^{-3}$ | $2.62\times10^{-3}$ | $3.62\times10^{-3}$ | 230.00 |
| | LS-SVMs | 0.99999 | $2.14\times10^{-8}$ | $2.66\times10^{-8}$ | $6.68\times10^{-7}$ | 85.40 |
| | NLS-SVMs | 0.99999 | $1.68\times10^{-7}$ | $1.98\times10^{-7}$ | $3.46\times10^{-7}$ | 3.25 |
| *Problem* 16 | PINN | 0.99812 | $1.24\times10^{-2}$ | $1.35\times10^{-2}$ | $1.80\times10^{-2}$ | 2699.00 |
| | LS-SVMs | 0.99999 | $1.50\times10^{-4}$ | $1.88\times10^{-4}$ | $6.84\times10^{-4}$ | 24.30 |
| | NLS-SVMs | 0.99999 | $1.72\times10^{-5}$ | $2.06\times10^{-5}$ | $5.07\times10^{-5}$ | 0.42 |

Table 12: Performance evaluation of NLS-SVMs for long-time integration of ODEs: accuracy and computational efficiency

| ODEs | Model | $R^2$ | MAE | RMSE | $\|y - \hat{y}\|_\infty$ | Time/s |
|---|---|---|---|---|---|---|
| *Problem* 1 | NLS-SVMs | 0.99999 | $1.38\times10^{-4}$ | $4.23\times10^{-4}$ | $4.40\times10^{-3}$ | 34.80 |
| *Problem* 3 | NLS-SVMs | 0.99999 | $5.95\times10^{-8}$ | $1.18\times10^{-7}$ | $1.76\times10^{-6}$ | 24.20 |
| *Problem* 5 | NLS-SVMs | 0.99999 | $9.96\times10^{-5}$ | $2.36\times10^{-4}$ | $2.70\times10^{-3}$ | 413 |
| *Problem* 7 | NLS-SVMs | 0.99911 | $1.59\times10^{-3}$ | $4.01\times10^{-3}$ | $2.36\times10^{-2}$ | 299.00 |
| *Problem* 8 | NLS-SVMs | 0.99999 | $4.09\times10^{-4}$ | $5.08\times10^{-4}$ | $1.71\times10^{-3}$ | 24.70 |
| *Problem* 9 | NLS-SVMs | 0.99996 | $2.55\times10^{-4}$ | $6.31\times10^{-4}$ | $6.33\times10^{-3}$ | 459.00 |
| *Problem* 10 | NLS-SVMs | 0.99999 | $2.72\times10^{-5}$ | $5.47\times10^{-5}$ | $3.49\times10^{-4}$ | 460.00 |
| *Problem* 11 | NLS-SVMs | 0.99981 | $7.85\times10^{-3}$ | $9.62\times10^{-3}$ | $1.55\times10^{-2}$ | 372.00 |
| *Problem* 12 | NLS-SVMs | 0.99709 | $6.55\times10^{-2}$ | $7.27\times10^{-2}$ | $1.05\times10^{-1}$ | 379.00 |

## B.8 Performance improvements

Table 13: Benchmarking NLS-SVMs against PINN and LS-SVMs on ODE problems: accuracy and speed

| ODEs | Model | $\Delta R^2$ | $\Delta MAE/\%$ | $\Delta RMSE/\%$ | $\Delta \|y - \hat{y}\|_\infty /\%$ | Speedup |
|---|---|---|---|---|---|---|
| $Pro1$ | NLS-SVMs vs PINN | +0.00000 | $\downarrow 3.66 \times 10^{-1}$ | $\downarrow 4.10 \times 10^{-1}$ | $\downarrow 8.04 \times 10^{-1}$ | $\uparrow 1222$ |
| | NLS-SVMs vs LS-SVMs | +0.00000 | $\uparrow 3.13 \times 10^{-2}$ | $\uparrow 3.75 \times 10^{-2}$ | $\uparrow 7.00 \times 10^{-2}$ | $\uparrow 18$ |
| $Pro2$ | NLS-SVMs vs PINN | +0.00010 | $\downarrow 2.03 \times 10^{-1}$ | $\downarrow 2.09 \times 10^{-1}$ | $\downarrow 3.25 \times 10^{-1}$ | $\uparrow 733$ |
| | NLS-SVMs vs LS-SVMs | +0.00000 | $\uparrow 2.47 \times 10^{-4}$ | $\uparrow 2.92 \times 10^{-4}$ | $\uparrow 6.68 \times 10^{-4}$ | $\uparrow 2885$ |
| $Pro3$ | NLS-SVMs vs PINN | +0.25088 | $\downarrow 1.82 \times 10^{1}$ | $\downarrow 2.88 \times 10^{1}$ | $\downarrow 1.03 \times 10^{2}$ | $\uparrow 2159$ |
| | NLS-SVMs vs LS-SVMs | +0.00000 | $\downarrow 4.90 \times 10^{-7}$ | $\downarrow 4.60 \times 10^{-7}$ | $\uparrow 4.00 \times 10^{-7}$ | $\uparrow 3355$ |
| $Pro4$ | NLS-SVMs vs PINN | +0.05445 | $\downarrow 5.34 \times 10^{1}$ | $\downarrow 7.20 \times 10^{1}$ | $\downarrow 1.81 \times 10^{2}$ | $\uparrow 49$ |
| | NLS-SVMs vs LS-SVMs | +0.00000 | $\uparrow 4.94 \times 10^{-2}$ | $\uparrow 6.14 \times 10^{-2}$ | $\uparrow 1.29 \times 10^{-1}$ | $\uparrow 370$ |
| $Pro5$ | NLS-SVMs vs PINN | -0.00001 | $\downarrow 1.51 \times 10^{-2}$ | $\uparrow 3.00 \times 10^{-4}$ | $\uparrow 5.40 \times 10^{-2}$ | $\uparrow 171$ |
| | NLS-SVMs vs LS-SVMs | +0.00000 | $\downarrow 1.00 \times 10^{-3}$ | $\downarrow 4.00 \times 10^{-3}$ | $\downarrow 1.70 \times 10^{-2}$ | $\uparrow 10$ |
| $Pro6$ | NLS-SVMs vs PINN | -0.10953 | $\downarrow 1.38 \times 10^{0}$ | $\downarrow 1.66 \times 10^{0}$ | $\downarrow 3.40 \times 10^{0}$ | $\uparrow 190$ |
| | NLS-SVMs vs LS-SVMs | -0.00019 | $\uparrow 4.42 \times 10^{-2}$ | $\uparrow 4.99 \times 10^{-2}$ | $\uparrow 8.37 \times 10^{-2}$ | $\uparrow 19$ |
| $Pro7$ | NLS-SVMs vs PINN | +0.08548 | $\downarrow 6.44 \times 10^{0}$ | $\downarrow 7.89 \times 10^{0}$ | $\downarrow 1.60 \times 10^{1}$ | $\uparrow 2042$ |
| | NLS-SVMs vs LS-SVMs | +0.00000 | $\uparrow 0.00 \times 10^{0}$ | $\uparrow 1.00 \times 10^{-3}$ | $\uparrow 0.00 \times 10^{0}$ | $\uparrow 46$ |
| $Pro8$ | NLS-SVMs vs PINN | +0.00026 | $\downarrow 5.17 \times 10^{-1}$ | $\downarrow 5.85 \times 10^{-1}$ | $\downarrow 1.02 \times 10^{0}$ | $\uparrow 3164$ |
| | NLS-SVMs vs LS-SVMs | +0.00000 | $\uparrow 4.95 \times 10^{-2}$ | $\uparrow 6.16 \times 10^{-3}$ | $\uparrow 1.94 \times 10^{-2}$ | $\uparrow 47$ |
| $Pro9$ | NLS-SVMs vs PINN | +0.00000 | $\downarrow 2.92 \times 10^{-3}$ | $\downarrow 3.04 \times 10^{-3}$ | $\uparrow 3.20 \times 10^{-3}$ | $\uparrow 1651$ |
| | NLS-SVMs vs LS-SVMs | +0.00000 | $\uparrow 4.14 \times 10^{-3}$ | $\uparrow 5.00 \times 10^{-3}$ | $\uparrow 1.59 \times 10^{-2}$ | $\uparrow 43$ |
| $Pro10$ | NLS-SVMs vs PINN | +0.01708 | $\downarrow 3.12 \times 10^{0}$ | $\downarrow 4.07 \times 10^{0}$ | $\downarrow 8.62 \times 10^{0}$ | $\uparrow 1298$ |
| | NLS-SVMs vs LS-SVMs | +0.00000 | $\uparrow 3.35 \times 10^{-6}$ | $\uparrow 4.72 \times 10^{-6}$ | $\downarrow 9.00 \times 10^{-7}$ | $\uparrow 50$ |
| $Pro11$ | NLS-SVMs vs PINN | +0.00000 | $\downarrow 1.62 \times 10^{-3}$ | $\downarrow 1.90 \times 10^{-3}$ | $\downarrow 3.32 \times 10^{-3}$ | $\uparrow 1816$ |
| | NLS-SVMs vs LS-SVMs | +0.00000 | $\downarrow 2.57 \times 10^{-6}$ | $\downarrow 3.18 \times 10^{-6}$ | $\downarrow 1.07 \times 10^{-5}$ | $\uparrow 36$ |
| $Pro12$ | NLS-SVMs vs PINN | +0.00000 | $\downarrow 2.28 \times 10^{-3}$ | $\downarrow 2.29 \times 10^{-3}$ | $\downarrow 2.60 \times 10^{-3}$ | $\uparrow 2107$ |
| | NLS-SVMs vs LS-SVMs | +0.00000 | $\downarrow 1.25 \times 10^{-6}$ | $\downarrow 1.54 \times 10^{-6}$ | $\downarrow 4.60 \times 10^{-6}$ | $\uparrow 36$ |
| $Pro13$ | NLS-SVMs vs PINN | +0.00012 | $\downarrow 4.22 \times 10^{-1}$ | $\downarrow 5.72 \times 10^{-1}$ | $\downarrow 1.05 \times 10^{0}$ | $\uparrow 3073$ |
| | NLS-SVMs vs LS-SVMs | +0.00000 | $\uparrow 7.10 \times 10^{-6}$ | $\uparrow 9.40 \times 10^{-6}$ | $\uparrow 4.94 \times 10^{-5}$ | $\uparrow 70$ |
| $Pro14$ | NLS-SVMs vs PINN | +0.00000 | $\downarrow 7.05 \times 10^{-4}$ | $\downarrow 7.20 \times 10^{-4}$ | $\downarrow 9.55 \times 10^{-4}$ | $\uparrow 81$ |
| | NLS-SVMs vs LS-SVMs | +0.00000 | $\uparrow 4.32 \times 10^{-5}$ | $\uparrow 6.53 \times 10^{-5}$ | $\uparrow 1.83 \times 10^{-4}$ | $\uparrow 26$ |
| $Pro15$ | NLS-SVMs vs PINN | +0.00225 | $\downarrow 2.42 \times 10^{-1}$ | $\downarrow 2.62 \times 10^{-1}$ | $\downarrow 3.62 \times 10^{-1}$ | $\uparrow 71$ |
| | NLS-SVMs vs LS-SVMs | +0.00000 | $\uparrow 1.47 \times 10^{-5}$ | $\uparrow 1.71 \times 10^{-5}$ | $\downarrow 3.22 \times 10^{-5}$ | $\uparrow 26$ |
| $Pro16$ | NLS-SVMs vs PINN | +0.00187 | $\downarrow 1.24 \times 10^{0}$ | $\downarrow 1.35 \times 10^{0}$ | $\downarrow 1.79 \times 10^{0}$ | $\uparrow 6426$ |
| | NLS-SVMs vs LS-SVMs | +0.00000 | $\downarrow 1.33 \times 10^{-2}$ | $\downarrow 1.64 \times 10^{-2}$ | $\downarrow 6.33 \times 10^{-2}$ | $\uparrow 58$ |

