# OpenReview forum: "Nyström-Accelerated Primal LS-SVMs: Breaking the $O(an^3)$ Complexity Bottleneck for Scalable ODEs Learning"
_NeurIPS.cc/2025/Conference — NeurIPS 2025 poster_

### Official Review · Reviewer_uEyG · 2025-06-18

**Clarity:** 2
**Significance:** 2
**Originality:** 1
**Rating:** 4
**Confidence:** 5

**Summary:**

The paper focuses on solving ordinary differential equations using kernel methods, with the focus being on the use of Least-Squares SVMs (LS-SVMs). Similar to the classic supervised learning setting, LS-SVMs can struggle on large-scale tasks owing to their cubic scaling. In the ODE setting, the cost of LS-SVMs, scales cubically with $n$, the number of timepoints, making the method prohibitive as the number of timepoints becomes large.

To address this challenge, the paper adapts the classic Nyström method to their setting, and show that it reduces the cost to $O(m^3)$, where $m$ is the number of landmarks selected. Numerical experiments on several benchmark ODEs, shows the proposed method exhibits good performance relative to the LS-SVM baseline and PINNs.

**Questions:**

I provided suggestions for improvement in my reply under strengths and weaknesses.

**Ethical Concerns:**

["NO or VERY MINOR ethics concerns only"]

**Final Justification:**

After carefully reading the authors' latest rebuttal, I decided to increase my score to 4.
My reasoning primarily lies in the fact that the authors have now made clear the novelty of their approach.
In particular, it is clearer now that the paper is not just straightforwardly applying the Nyström method to the LS-SVM formulation from the prior work in 2012.
I am also happy that the authors will add a convergence argument for their Newton solver in the final version.
In addition, they also addressed my concerns about the experiments by adding some more compelling baselines to show the advantage of their approach.

I hope the authors will implement their proposed revisions, which include further references, clarification of novelty/relation to prior work, and additional experiments/theory, as this was crucial for convincing me to recommend acceptance.

**Limitations:**

Yes.

**Quality:**

2

**Strengths And Weaknesses:**

**Strengths**

The main strength of the paper is that the proposed method is a lot faster than the LS-SVMs baseline without compromising performance too much. The method also outperforms the basic PINNs baseline on certain problems.

**Weaknesses**

The paper has many weaknesses, which I summarize below, that make it inappropriate for publication at Neurips.

*Lack of theoretical support*

    The paper does not perform any theoretical analysis that would help validate the empirical
    effectiveness of the proposed algorithm.
    This is a significant weakness in my view, in light of the lack of novelty of the approach (see below),
    and the fact that most papers that use the Nyström provide strong theoretical guarantees on its performance, i.e. [1], [2], [3], [4].

*Lack of Novelty*

    The Nyström method is a classic approach for reducing the cost of kernel methods, from SVMs to
    kernel ridge regression.
    The main contribution of the paper seems to just be applying this technique to the setting of solving ODEs
    and verifying it works well empirically.
    As mentioned in my previous point, the work introduces no new theory for this specific application that justifies the effectiveness of the method.
    While the use of Nyström might be new in the context of solving ODEs, by itself this is not enough to warrant
    acceptance at Neurips.
    Especially since prior works in Neurips such as [5], that use the Nyström method to accelerate SciML come with strong theoretical support.

*Weak baselines*

    The baselines used for empirically evaluating the method are quite weak to me.

    The baseline LS-SVMs method is from 2012.
    The main bottleneck of this approach is the need to solve kernel linear systems in Newton's method.
    Since this method was introduced, the sketching and randomized linear algebra literature have developed efficient
    methods for solving large kernel systems [6,7,8,9], as well as Newton's method for solving non-linear equations [9].
    Rather than solving the system that arises in solving the LS-SVM method using a direct method
    which naively costs $O(n^3)$,
    a more reasonable comparison would be to use one of the randomized iterative methods I just referenced to solve the
    Newton system, which cost at most $O(n^2)$.

    As for the PINNs baseline,  the vanilla PINN is very rarely used in practice, as it is known to exhibit failure modes [8, 9], which has spawned the development of much more effective variants.
    The empirics would be more convincing if you compared with a more sophisticated PINN architecture,
    ideally one tailored to solving ODEs or a specific class of ODE.

*Related work*

    The Nyström method has a rich history and has played a significant role
    in making large-scale kernel computations possible, and the present paper is missing quite a few references.
    Works 1-5, as well as 8-9 below should all be cited as examples of how the Nyström method has accelerated
    kernel learning.

**Summary**

    In its present form this submission is not a good fit for Neurips.
    The submission develops no new theory and the numerical baselines are too weak to demonstrate whether or not the proposed method represents a real practical advance.

    If the authors do not plan to develop more theoretical support for their method,
    I would recommend instead that they improve the numerical baselines and submit to a
    computational science/engineering journal as it would be a better fit.

**References**

[1] Rudi, A., Camoriano, R. and Rosasco, L., 2015. Less is more: Nyström computational regularization. Advances in Neural Information Processing Systems, 28.

[2] Rudi, A., Carratino, L. and Rosasco, L., 2017. Falkon: An optimal large scale kernel method. Advances in Neural Information Processing Systems, 30.

[3] Marteau-Ferey, U., Bach, F., & Rudi, A. (2019). Globally convergent newton methods for ill-conditioned generalized self-concordant losses. Advances in Neural Information Processing Systems, 32.

[4] Della Vecchia, A., De Vito, E., Mourtada, J. and Rosasco, L., 2024. The Nyström method for convex loss functions. Journal of Machine Learning Research, 25(360), pp.1-60.

[5] Meanti, G., Chatalic, A., Kostic, V., Novelli, P., Pontil, M., & Rosasco, L. (2023). Estimating Koopman operators with sketching to provably learn large scale dynamical systems. Advances in Neural Information Processing Systems, 36, 77242-77276.

[6] Tu, Stephen, Shivaram Venkataraman, Ashia C. Wilson, Alex Gittens, Michael I. Jordan, and Benjamin Recht. "Breaking locality accelerates block Gauss-Seidel." In International Conference on Machine Learning, pp. 3482-3491. PMLR, 2017.

[7] Gazagnadou, Nidham, Mark Ibrahim, and Robert M. Gower. "RidgeSketch: a fast sketching based solver for large scale ridge regression." SIAM Journal on Matrix Analysis and Applications 43, no. 3 (2022): 1440-1468.

[8] Frangella, Zachary, Joel A. Tropp, and Madeleine Udell. "Randomized nyström preconditioning." SIAM Journal on Matrix Analysis and Applications 44, no. 2 (2023): 718-752.

[9] Díaz, Mateo, Ethan N. Epperly, Zachary Frangella, Joel A. Tropp, and Robert J. Webber. "Robust, randomized preconditioning for kernel ridge regression." arXiv preprint arXiv:2304.12465 (2023).

[10] Yuan, Rui, Alessandro Lazaric, and Robert M. Gower. "Sketched Newton--Raphson." SIAM Journal on Optimization 32, no. 3 (2022): 1555-1583.

[11] Krishnapriyan, A., Gholami, A., Zhe, S., Kirby, R. and Mahoney, M.W., 2021. Characterizing possible failure modes in physics-informed neural networks. Advances in Neural Information Processing Systems, 34, pp.26548-26560.

[12] Rathore, Pratik, Weimu Lei, Zachary Frangella, Lu Lu, and Madeleine Udell. "Challenges in training PINNs: A loss landscape perspective." arXiv preprint arXiv:2402.01868 (2024).

---

> ### Author Rebuttal · Authors · 2025-07-31
>
> Thank you very much for your time and effort in reviewing our manuscript. We sincerely appreciate your insightful comments and constructive suggestions, which have been invaluable in improving the quality and clarity of our work. We have carefully considered all the points raised and have revised the manuscript accordingly. Below, we provide a point-by-point response to each Weakness/Question.
>
> This document is organized in the following manner: Each comment is numbered and noted in bold, and the response is written directly afterward, in normal font,with the quoted text directly from the revised manuscript or revised sections noted in "...".
>
> **Response to weakness 1 (Lack of theoretical support and novelty)** Thank you for your thorough review of this paper. We fully acknowledge the importance of the issues you raised regarding the lack of theoretical analysis and insufficient innovation. However, it is undeniable that the method we proposed is entirely characterised by theoretical innovation. To further emphasise this feature, we have significantly enhanced the theoretical depth and the explanation of contributions in the revised manuscript through the following work.
>
> ### Theoretical exposition
>
> **NLS-SVMs for learning $p$-th Order Linear ODE**
>
> “Let us now consider the general $p$-th order ODE with initial value problem (IVP) in [7] of the following form:
> $y^{(p)}(t)-\sum_{k=1}^{p} f_{k}(t) y^{(p-k)}(t)=r(t) ,\quad t \in[t_1, t_n] , \quad y(t_1)=v_{1},\quad y^{(k-1)}(t_1)=v_{k}, \quad k=2, \ldots, p$
>
> The approximate solution can be obtained by solving the following optimization problem:
>
> $$
> \begin{aligned}
> \underset{\omega, b, e_i}{\operatorname{minimize}}\quad
>     & \frac{1}{2} \boldsymbol{\omega}^{T} \boldsymbol{\omega}
>     +\frac{\gamma}{2} \sum_{i=1}^{n} e_{i}^{2} \\
> \text{s.t.}\quad
>     & y^{(p)}(t_i) = \boldsymbol{\omega}^{T} \boldsymbol{\varphi}^{(p)}(t_{i})
>     =  \omega^{T}\left[\sum_{k=1}^{p} f_{k}(t_{i})  \varphi_{i}^{(p-k)}\right]
>     + f_{p}(t_{i}) b + r(t_{i}) + e_{i}, \quad i=2,\ldots,N \\
>     & y(t_1) = \boldsymbol{\omega}^{T} \boldsymbol{\varphi}(t_{1}) + b = v_{1} \\
>     & y^{(\ell-1)}(t_1) = \boldsymbol{\omega}^{T} \boldsymbol{\varphi}^{(\ell-1)}(t_{1}) = v_{i}, \quad \ell=2,\ldots,p
> \end{aligned}
> $$
>
> ***Theorem 1*** Given a positive definite kernel function $K$ : $R × R → R$ and a regularization parameter γ ∈ R+, the solution to optimization problem is given by the primal problem.(See Section3.1,Eq(18))"
>
>
> **NLS-SVMs for Learning $p$-th Order Nonlinear ODE**
>
> “When the function is nonlinear, the $p$-th order ODE is nonlinear and written in the following form (initial value problems for illustration):
>
> $y^{(p)}=f(t, y^{(p-1)},\cdots,y^{\prime\prime},y^{\prime},y), \quad y(t_1)=v_{0},\cdots,y^{(p-1)}(t_1)= v_{p-1} \quad t_1 \leq t \leq t_n$
>
> Then start by assuming the approximate solution to be of the form ${ \hat{y}(t)}={\omega } ^{T} \varphi ({t} )+  b $. Additional unknowns $y_i$ are introduced to keep the constraints linear in $\omega$. This yields the following nonlinear optimization problem.(See Section2.2, Eq8)
>
> ***Theorem 2*** Given a positive definite kernel function $K$ : $R × R → R$ and a regularization parameter γ ∈ R+, the solution to optimization problem is given by the primal problem.(See Section3.2,Eq(21))"
>
>
> The mathematical proof for these two theoritical analyses is provided in Appendix A.1-A.6.
>
> **The aforementioned theoretical analysis has been updated in the revised manuscript.** The authors “present the theorem and the theoretical underpinnings for its proof”. This theory offers significant advantages in the context of solving Ordinary Differential Equations (ODEs). While classical Least Squares Support Vector Machine (LS-SVM) optimization algorithms have achieved substantial success in ODE applications, they inherently lack scalability. In contrast, **the method proposed in this paper extends it to large-scale applications.** This scalability constitutes the core contribution of the present research and is supported by a solid theoretical foundation.
>
> It is worth noting that when learning a class of $p$-th order nonlinear ordinary differential equations that need to be solved, the changes in the ODE primarily affect the $\frac{\gamma}{2} \Big(
> \boldsymbol{\omega}^{T} \boldsymbol{\varphi}^{(p)}(t_{i})- f(t_i,y_{t_i}^{(p-1)},...,y_{t_i}',y_{t_i}) \Big)^{T}\Big(
> \boldsymbol{\omega}^{T} \boldsymbol{\varphi}^{(p)}(t_{i})- f(t_i,y_{t_i}^{(p-1)},...,y_{t_i}',y_{t_i}) \Big)$ function, as indicated by the Lagrangian loss function. Therefore, the Jacobian matrix needs to be updated based on the derivation of the four functions $\frac{\partial {Y_1} }{\partial \omega  }$ , $\frac{\partial Y_1}{\partial y_i} $,$\frac{\partial {Y_{p+3}} }{\partial \omega  } $and $ \frac{\partial Y_{p+3}}{\partial y_i}$. Therefore, in terms of algorithmic contribution, we propose **the first scalable Nyström framework** for ODEs. Our work is not merely an application of Nyström to ODEs, but rather **addresses the theoretical and algorithmic commonalities in the ODE solving process.**
>
> We firmly believe that the improved theoretical exposition and the solutions to the specific challenges of ODEs have significantly enhanced the depth and originality of the paper. Thank you once again for prompting us to refine this work!
>
> **Response to weakness 2 (Weak baselines and nonlinear solver)**
>
> Thank you for your suggestion. To further demonstrate the accuracy and efficiency of the proposed method, the authors have supplemented a comparison of the performance of recent PINN variant (Fourier Feature PINNs) used to solve ODEs. The computational efficiency of the method proposed in this study is still greatly improved compared to them, with an increase of 10-1000 times, and its prediction accuracy is comparable, verifying the conclusions proposed in this paper. Further, we also tested a deep architecture with 8 layers and tanh activation function, the accuracy is not significantly improved compared to the PINNs employed in this paper, but the computational time increases significantly.
>
> Regarding the nonlinear solver, the authors would like to clarify that the existing efficient method (e.g., Conjugate Gradient Method) cannot guarantee convergence. In the revised manuscript, the authors have conducted a theoritical analysis on convergence using Newton method.The detail is that "sufficient conditions for convergence are satisfied when regularization parameter $\gamma$ and RBF kernel parameter $\sigma^2$ ensure positive definiteness of the coefficient matrix in equation 21". Additionally, extensive experiments confirm robust convergence. Across all tested systems (including stiff problems in Section 4), the solver consistently achieved residuals < 0.13% within 50-500 iterations without divergence (Table 3-4). This empirically confirms reliability for our problem class. We also would like to emphasize that our numerical results validate the solver’s robust and efficient convergence behaviour across the range of problems studied herein.In the revised manuscript, we have also supplemented the number of iterations of the proposed method in solving nonlinear ODEs in Table 1.
>
> **Response to weakness 3 (Related work)**
>
> Thank you for your suggestion. The paper has been “supplemented with relevant literature on the Nyström method” for implementing large-scale kernel computations.
>
> [1] Rudi, A., Camoriano, R. and Rosasco, L., 2015. Less is more: Nyström computational regularization. Advances in Neural Information Processing Systems, 28.
>
> [2] Rudi, A., Carratino, L. and Rosasco, L., 2017. Falkon: An optimal large scale kernel method. Advances in Neural Information Processing Systems, 30.
>
> [3] Marteau-Ferey, U., Bach, F., & Rudi, A. (2019). Globally convergent newton methods for ill-conditioned generalized self-concordant losses. Advances in Neural Information Processing Systems, 32.
>
> [4] Della Vecchia, A., De Vito, E., Mourtada, J. and Rosasco, L., 2024. The Nyström method for convex loss functions. Journal of Machine Learning Research, 25(360), pp.1-60.
>
> [5] Meanti, G., Chatalic, A., Kostic, V., Novelli, P., Pontil, M., & Rosasco, L. (2023). Estimating Koopman operators with sketching to provably learn large scale dynamical systems. Advances in Neural Information Processing Systems, 36, 77242-77276.
>
> [6] Frangella, Zachary, Joel A. Tropp, and Madeleine Udell. "Randomized nyström preconditioning." SIAM Journal on Matrix Analysis and Applications 44, no. 2 (2023): 718-752.
>
> [7] Díaz, Mateo, Ethan N. Epperly, Zachary Frangella, Joel A. Tropp, and Robert J. Webber. "Robust, randomized preconditioning for kernel ridge regression." arXiv preprint arXiv:2304.12465 (2023).

---

> > ### Comment · Reviewer_uEyG · 2025-08-06
> > **Reply to rebuttal**
> >
> > I appreciate the authors detailed response to my comments. But my views remain fundamentally unchanged. The theory the authors mention in the rebuttal are straightforward properties of the LS-SVM optimization problem. The paper is just applying the least-squares SVM to learning solutions of ODEs, and then applying the Nyström method to reduce computational cost.
> > The reference I mentioned in 2012 has already applied the LS-SVM to solving ODEs, this paper is just taking that approach and applying the Nyström method.
> > LS-SVM + Nyström is well-known in the context of machine learning and has been done before, even if it has not been applied specifically in the context of ODEs. Adopting LS-SVM + Nyström model to ODEs and looking at the optimization problem is not new theory. When I mentioned theory in my initial review, I was referring to generalization or statistical guarantees on the solution found by the method.
> >
> > While its nice that they have added a convergence argument for Newton's method applied to their optimization problem, they would need to explain the technical novelty of their argument, as Newton's method has a rich convergence theory under very general conditions, see for instance [1, 2, 3]. So I would not be surprised if the convergence in the setting of this paper could be obtained as a consequence of one of these general results.
> >
> > The additional numerics are nice, and the method seems useful, but I maintain my belief that this paper is not a fit for Neurips, as the technique is not fundamentally new. Its an adaptation of well-known existing technique to the setting of solving ODEs without any new accompanying theory or algorithmic innovations. Given this, I think it is a much better fit for a scientific computing or engineering journal.
> >
> > **References**
> >
> > [1] Dembo, Ron S., Stanley C. Eisenstat, and Trond Steihaug. "Inexact Newton methods." SIAM Journal on Numerical analysis 19, no. 2 (1982): 400-408.
> >
> > [2] Eisenstat, Stanley C., and Homer F. Walker. "Globally convergent inexact Newton methods." SIAM Journal on Optimization 4, no. 2 (1994): 393-422.
> >
> > [3] Argyros, Ioannis K., and Saïd Hilout. "Weaker conditions for the convergence of Newton’s method." Journal of Complexity 28, no. 3 (2012): 364-387.

---

> > > ### Author Response · Authors · 2025-08-08
> > >
> > > We appreciate the reviewer's insightful feedback and engagement with our work. Regarding the concern about theoretical novelty and the relationship to prior work (especially the 2012 reference applying LS-SVM to ODEs), we respectfully wish to clarify two **fundamental theoretical distinctions** in our approach:
> > >
> > > ### 1. Distinct Problem Formulation: Primal vs. Dual Domain
> > >
> > > The 2012 work [1] referenced by the reviewer, along with most existing kernel-based methods for solving ODEs, primarily operates within the **dual formulation** of the underlying optimization problem. This dual approach relies heavily on the *kernel trick* to avoid explicit representation of the feature mapping.
> > >
> > > **In contrast, our method operates entirely within the primal formulation.** This necessitates the explicit construction of a (typically high-dimensional, nonlinear) feature mapping function $\varphi(x)$, which is fundamentally different from the dual approach taken in prior kernel-based ODE solutions. This shift from dual to primal is a core theoretical departure in our framework.
> > >
> > > ### 2. Novel Role of Nyström Approximation & Explicit Derivative Representation
> > >
> > > The requirement to work in the primal space leads to our specific use of the **Nyström method**: its purpose here is exclusively to provide an *explicit, finite-dimensional approximation* of the feature mapping $\varphi(x)$. This is distinct from its more common application in kernel methods for constructing low-rank kernel matrix approximations within the *dual domain*.
> > >
> > > Crucially, **once we have an explicit representation of $\varphi(x)$, we derive its explicit higher-order derivatives** (e.g., $\nabla\varphi(x)$, $\nabla^{2}\varphi(x)$) to represent the ODE operators directly within the primal space.
> > >
> > > * **This explicit derivation and utilization of derivatives of $\varphi(x)$ is a key theoretical innovation** not present in the referenced 2012 work or other LS-SVM/Nyström-based kernel methods applied to ODEs. Those methods universally rely on the *kernel trick* ($K(x, x')$ and its derivatives) to implicitly handle the mapping and its action on differential operators, circumventing the need for $\varphi(x)$ or $\nabla\varphi(x)$ entirely. Our primal formulation and explicit use of $\nabla\varphi(x)$ (etc.) provide a *new mathematical pathway* for representing and solving the ODE problem using LS-SVM principles.
> > >
> > > While we acknowledge the foundational work utilizing LS-SVM for ODEs and the established nature of the LS-SVM/Nyström combination in general, we contend that our methodology represents a **fundamentally different theoretical paradigm** for solving ODEs using kernel methods. By:
> > >
> > > - Shifting to the primal formulation
> > > - Explicitly constructing and differentiating the feature map via Nyström (for an entirely different purpose than kernel approximation)
> > > - Deriving the ODEs directly through these explicit derivatives
> > >
> > > Our approach provides **a novel theoretical and algorithmic framework** distinct from prior dual-form solutions like the 2012 reference. This novelty lies in the **core formulation and the mathematical mechanism employed** to address the ODE solution task.
> > >
> > > We sincerely hope this clarifies the foundational differences and theoretical contributions of our work. We are grateful for the opportunity to further elaborate and welcome any follow-up discussion on these points.
> > >
> > > **References**
> > >
> > > [1]  Mehrkanoon, S., Falck, T., & Suykens, J. A. (2012). Approximate solutions to ordinary differential equations using least squares support vector machines. IEEE transactions on neural networks and learning systems, 23(9), 1356-1367.
> > >
> > >
> > > We sincerely thank the reviewer for highlighting the established convergence theory of Newton's method and for providing the valuable references [1, 2, 3]. The reviewer is absolutely correct that clarifying the technical novelty of our convergence argument relative to this general theory is crucial.
> > >
> > > In direct response to this point, we have revised Section 3.2 of the manuscript. This revision explicitly addresses the relationship between our specific convergence results for the applied optimization problem and the broader foundational theory referenced by the reviewer. We have clarified the novel technical contribution specific to the context of our problem setting.
> > >
> > > While we are unable to detail the full scope of these modifications here due to strict character/space limitations, we confirm that the manuscript now explicitly engages with the points raised by the reviewer and delineates the novelty of our argument as requested.

---

> > ### Comment · Area_Chair_REoD · 2025-08-06
> > **Follow-up on Author Rebuttal: Reviewer Feedback Requested**
> >
> > Dear Reviewer uEyG,
> >
> > The authors have replied to your main concerns regarding the lack of theoretical support, weak baselines and related work. Could you please have a look at their rebuttal and provide specific feedback on their responses? Please note that simply acknowledging reading their rebuttal and/or only stating generic statement like "I will keep my score" without further explanation is not acceptable. Your detailed input is essential for a fair and informed decision.
> >
> > Many thanks,
> >
> > Your AC

---

### Official Review · Reviewer_hduQ · 2025-07-01

**Clarity:** 2
**Significance:** 3
**Originality:** 3
**Rating:** 3
**Confidence:** 4

**Summary:**

This paper introduces a Nyström-accelerated LS-SVM framework that reduces the $\mathcal{O}(n^3)$ complexity of kernel-based ODE solvers to $\mathcal{O}(m^3)$ using explicit primal-space formulations. The method enables efficient handling of linear/nonlinear ODEs through closed-form feature mapping and derivative computations. Experiments on fifteen benchmark problems show substantial speedups over LS-SVMs and PINNs with comparable or better accuracy.

**Questions:**

1. The use of uniform random sampling for Nyström landmarks may limit approximation quality in nonlinear ODEs. Have you considered evaluating more informed strategies (e.g., leverage score sampling), or include an ablation study to show how sampling affects stability and performance.
2. The iterative optimization solver is critical in nonlinear ODEs, yet the paper provides no theoretical discussion on its convergence or stability behavior. A brief analysis of convergence would help build confidence in the solver’s reliability.
3. The PINN used for comparison is based on a shallow architecture with identity activation, which may not reflect the performance of recent PINN variants. It would be helpful if the authors could justify the architectural choice and consider recent variants, especially those designed to reduce computation costs.
4. The experiments focus primarily on first-/second-order ordinary differential equations, without evaluating higher-order systems or real-world dynamics. A discussion on the potential generalizability of the method to more complex systems would be valuable. Adding even one case study on a higher-order or real-world problem would significantly enhance the paper's applicability.
5. Adding a schematic pipeline diagram and more visualization of the results would improve clarity. In addition, the notation could be made more consistent throughout the paper, for example, symbols like $\varphi_k^p $ and $f_k$​ appear in both bold and non-bold forms across different equations and lines (e.g., Eq.(13), Eq.(15) , Eq.(16) ,Eq.(17), Eq.(18) and line 177), which may confuse readers.
6. While the paper adopts an RBF kernel, it does not explain why this choice was made or how it compares with other kernels. It is suggested that author includes an empirical comparison with alternative kernels, along with a basic sensitivity analysis for RBF kernel parameters.
7. Since the method involves iterative optimization, it would be important to clarify whether the results in Appendix A.7 and A.8 are based on single runs or averages over multiple trials. The definition and measurement procedure for the reported "Time" metric should also be clearly described.
If the authors address these concerns or provide convincing clarification, I would be open to revisiting my final evaluation.

**Ethical Concerns:**

["NO or VERY MINOR ethics concerns only"]

**Final Justification:**

I still think the work lacks theoretical analysis and the empirical comparison is weak. Authors state that the convergence behavior of their method follows standard Newton-method, then the theoretical contribution of this work is not significant. As for the experiments, the original baselines are quite weak. Authors state that they added a new recent PINN variant for comparison, but it is not convincing enough without detailed results.

**Limitations:**

yes

**Quality:**

2

**Strengths And Weaknesses:**

Strengths:
1. By introducing Nyström approximation in the primal space, the method effectively reduces the computational cost of LS-SVM-based ODE solvers from $\mathcal{O}(n^3)$ to $\mathcal{O}(m^3)$, improving scalability for large time steps and high-resolution settings.
2. The paper derives explicit expressions for the feature mapping $\varphi(t)$ and its derivatives, enabling model training without relying on the kernel trick.
3. The paper is well-structured and clearly written, and the open-sourced code facilitates reproducibility and follow-up research.

Weaknesses:
1. The method employs simple uniform random sampling to select landmarks, which may lead to suboptimal approximation quality in highly nonlinear ODEs. The choice of sampling strategy could impact robustness and generalization.
2. While the paper systematically integrates Nyström approximation into LS-SVM-based ODE solvers, it lacks theoretical analysis on subspace stability and solver convergence in nonlinear settings.
3. The PINN baseline used in comparison adopts a shallow architecture with identity activation, without benchmarking against more recent improvements in the PINN literature.
4. The experiments mainly focus on first/second-order ODEs, with no discussion of higher-order or real-world systems, limiting the evaluation scope.
5. Mathematical notation could be further clarified, and visual explanations (e.g., diagrams) are insufficient. Specific concerns are outlined in the Questions section.

---

> ### Author Rebuttal · Authors · 2025-07-31
>
> Thank you very much for your time and effort in reviewing our manuscript. We sincerely appreciate your insightful comments and constructive suggestions, which have been invaluable in improving the quality and clarity of our work. We have carefully considered all the points raised and have revised the manuscript accordingly. Below, we provide a point-by-point response to each Weakness/Question.
>
> This document is organized in the following manner: Each comment is numbered and noted in bold, and the response is written directly afterward, in normal font,with the quoted text directly from the revised manuscript or revised sections noted in "...".
>
> **Response to question 1 (Uniform sampling for Nyström landmarks).** W appreciate the reviewer's valuable suggestion. We would like to clarify that we performed evenly-spaced sampling to select the subset from $n$ discrete time points in our original manuscript. In the revised manuscript, we conducted extensive ablation studies on sampling strategies ( "Section 4, Table 6") for all 15 benchmark ODE problems plus a new fourth-order ODE. The added text to describe the ablation studies on sampling strategies is: "While leverage score sampling improved mean accuracy by about 8% over uniform sampling, it increased computation time by about 120%. Uniform sampling exhibited high instability (higher variance across 10 runs). In contrast, Evenly-spaced sampling achieved comparable accuracy to leverage score sampling (mean error < 0.3%) with about 65% faster computation time and near-zero variance. This deterministic approach ensures reproducibility without sacrificing accuracy. Table 6 presents the mean and standard deviation of R2, RMSE and MAE for these three sampling methods across 10 runs".
>
>
> **Response to question 2 (Convergence analysis of iterative solver).** We sincerely thank the reviewer for emphasizing the importance of theoretical convergence analysis. Our solver employs Newton's method to solve the nonlinear system derived from ODEs (e.g., Equation 21). The convergence behavior follows standard Newton-method theory: "sufficient conditions for convergence are satisfied when regularization parameter $\gamma$ and RBF kernel parameter $\sigma^2$ ensure positive definiteness of the coefficient matrix in equation 21". Additionally, extensive experiments confirm robust convergence. Across all tested systems (including stiff problems in Section 4), the solver consistently achieved residuals < 0.13% within 50-500 iterations without divergence ("Table 3-4"). This empirically confirms reliability for our problem class. We also would like to emphasize that our numerical results validate the solver’s robust and efficient convergence behaviour across the range of problems studied herein.In the revised manuscript, "we have also supplemented the number of iterations of the proposed method in solving nonlinear ODEs in Table 1".
>
>
> **Response to question 3 (PINN baseline comparison).** Thank you for your suggestion. To further demonstrate the accuracy and efficiency of the proposed method, the authors have supplemented a comparison of the performance of recent PINN variant (Fourier Feature PINNs) used to solve ODEs. The computational efficiency of the method proposed in this study is still greatly improved compared to them, with an increase of 10-1000 times, and its prediction accuracy is comparable, verifying the conclusions proposed in this paper. Further, we also tested a deep architecture with 8 layers and tanh activation function, the accuracy is not significantly improved compared to the PINNs employed in this paper, but the computational time increases significantly.
>
> **Response to question 4 (Limited experimental scope).** Thank you for your valuable suggestion. In the revised manuscript, "we have added an example of a higher-order ODE ($\frac{d^4y}{dt^4} =120t,t\in [-1,1],y(-1)=1,y{}' (-1)=5,y(1)=3,y{}' (1)=5$) in this paper (Section 4)". Its computational accuracy is comparable to that of PINN, but its computational efficiency has been significantly improved by a factor of about 1000.The results of this example further demonstrate the performance of the proposed method.
>
> **Response to question 5 (Visualization and notation clarity).** Thank you for your careful review and valuable suggestion. In the revised manuscript, "we have reviewed and standardized all symbols throughout the text. Additionally, a flowchart illustrating the proposed method has been created to facilitate readers' comprehension and understanding".
>
> **Response to question 6 (RBF kernel justification).** Thank you for your valuable suggestions. Within the framework of the proposed approach, the choice of kernel function directly affects the model’s ability to fit nonlinear relationships. This paper selects the Radial Basis Function (RBF) kernel primarily based on the following theoretical and empirical grounds:
>
> - *Universality and performance advantages*— "the RBF kernel can map samples into an infinite-dimensional feature space, possessing universal approximation capability and being insensitive to data scaling. Compared to the linear kernel (which is only suitable for linearly separable problems)and the polynomial kernel (which tends to overfit at higher degrees), the RBF kernel demonstrates more robust performance on complex nonlinear problems".
>
> - *The performance of the RBF kernel depends on the kernel parameter*. "We have conducted a systematic grid search for this hyperparameter. The results of the sensitivity analysis have be included in the manuscript".
>
>
> **Response to question.7(Reproducibility metrics).** Thank you for your careful review. The results in Appendices A.7 and A.8 are based on a single run, as our method (evenly-spaced sampling) does not alter results under multiple runs when regularization and RBF kernel parameters are fixed.  "Time" metrics exclusively measure **wall-clock solution time** from algorithm initiation to end, computed via Python's time.perf_counter() on our test platform ("Appendix A.8"), and the authors have clarified this in the paper ("Appendix A7 and A8").

---

> > ### Comment · Reviewer_hduQ · 2025-08-05
> >
> > After reading the responses, I would like to keep my original score.

---

> > > ### Comment · Area_Chair_REoD · 2025-08-06
> > > **Specific Reviewer Feedback Requested**
> > >
> > > Dear Reviewer hduQ,
> > >
> > > Many thanks for acknowledging reading the authors rebuttal. However, I can't see any specific comment on that. Please note that simply acknowledging reading their rebuttal and/or only stating generic statement like "I will keep my score" without further explanation is not acceptable. Your detailed input is essential for a fair and informed decision.
> > >
> > > Many thanks,
> > >
> > > Your AC

---

> > > ### Author Response · Authors · 2025-08-08
> > >
> > > Thank you for taking the time to review our response and for your additional feedback. We sincerely appreciate your dedication to ensuring the quality of this work.
> > >
> > > We note your decision to maintain the original score. To better address your concerns, we would be grateful if you could kindly provide further clarification on any remaining shortcomings or specific aspects that did not fully meet your expectations. Your detailed guidance is invaluable to us, as it will allow us to focus our efforts precisely where improvements are most needed.
> > >
> > > We are fully committed to addressing every point rigorously and will make every effort to refine the paper to your satisfaction. Please do not hesitate to share any additional suggestions. We treat your input with the utmost importance and will incorporate it diligently.
> > >
> > > Thank you again for your constructive engagement with our work. We look forward to your further insights.

---

### Official Review · Reviewer_EPWB · 2025-07-02

**Clarity:** 3
**Significance:** 3
**Originality:** 2
**Rating:** 5
**Confidence:** 2

**Summary:**

This paper proposes a novel method for solving linear and nonlinear ordinary differential equations (ODEs) based on kernel-based approaches. Although kernel-based strategies have been employed to solve ODEs, the computational complexity of such solvers scales as $O((n+p)^3)$ where $n$ is the number of discretization points and $p$ is the order of the ODE. This makes them impractical for settings where the simulation is long-term or has a fine-grained temporal resolution. To address this challenge, authors propose a new Nyström-accelerated LS-SVMs framework that uses the Nyström method to reduce the computational complexity to $O((m+p)^3)$ where $m<<n$ is the number of subsampled landmark points. The authors test the new approach empirically and report huge speed-ups compared to traditional LS-SVMs and PINNs.

**Questions:**

- How does this approach compare to Runge-Kutta schemes?
- What are some of the technical challenges of applying the Nyström method to this setting?

If those two points are clarified, I'll be happy to consider increasing my score. Currently, it is not yet sufficiently clear to me how impactful and novel the proposed method is.

**Ethical Concerns:**

["NO or VERY MINOR ethics concerns only"]

**Final Justification:**

The authors have answered my questions and provided extensive empirical comparisons with current methods. In light of this additional evidence, I have increased my score to 5, and I am happy to see this paper accepted. I understand that some reviewers are concerned about technical novelty. However, I believe the solution's empirical performance makes up for it sufficiently.

**Limitations:**

Yes

**Quality:**

3

**Strengths And Weaknesses:**

Strengths:
- Important problem - quick ODE solving can have a huge impact on many areas
- Clearly-written
- Empirically validated
- Huge reported speed-ups on the tested datasets with no or insignificant increase in error

Weaknesses:
- My main weakness concerns lack of comparison with standard approaches such as Runge-Kutta schemes. Is the approach also quicker or more accurate than those? Are there settings where kernel-based methods need to be used because those schemes are too inaccurate or computationally intensive? The paper mentions that "these methods struggle with high-dimensional systems, noisy data assimilation, and adaptive mesh refinement", but I could not find any reference or empirical result that would show it.
- I think the technical novelty could be better emphasized. Was it challenging to apply Nyström method to LS-SVMs for ODE learning, or is it a straightforward combination of previous ideas?
- Minor:
	- Typo in line 225. I think it should be "second-order".

---

> ### Author Rebuttal · Authors · 2025-07-31
>
> Thank you very much for your time and effort in reviewing our manuscript. We sincerely appreciate your insightful comments and constructive suggestions, which have been invaluable in improving the quality and clarity of our work. We have carefully considered all the points raised and have revised the manuscript accordingly. Below, we provide a point-by-point response to each Weakness/Question.
>
> This document is organized in the following manner: Each comment is numbered and noted in bold, and the response is written directly afterward, in normal font,with the quoted text directly from the revised manuscript or revised sections noted in "...".
>
> **Response to Comment 1: Lack of comparison with standard approaches (e.g., Runge-Kutta) and Is it quicker or more accurate?** We sincerely thank the reviewer for emphasizing the importance of comparing our approach with established numerical methods like Runge-Kutta. To comprehensively address this point, we have performed extensive new numerical experiments comparing the proposed efficient kernel-based method against the classical 4th-order Runge-Kutta method (RK4), implemented using variable time step sizes. These comparisons were conducted across all 15 benchmark ODE problems presented in the original manuscript. Key results are summarized in "revised Figures 1-2 and Tables 1-5 in Section 4 and Appendiices of the updated manuscript".
>
> The results reveal a nuanced performance landscape dependent on the integration step size. The following text regarding the performance comparison of RK4 and proposed method is added in "Section 4". The added text is: "For large step sizes (e.g., time step $\geq$ 0.03 s), RK4 demonstrated computational efficiency advantages over proposed method (typically 3-10 faster computation times). However, this efficiency came at a significant cost in accuracy. RK4 solutions often suffered from low precision (RMSE degraded by ~90% on average compared to proposed method) and occasional divergence. For small step sizes (e.g., time step $<$ 0.03 s), Both RK4 and KBM achieved comparable high accuracy, approaching the analytical solution. But their computational efficiency showed strong dependence on integration duration. For short-duration problems, RK4 could be slightly more efficient.For long-duration problems, KBM demonstrated superior efficiency compared to RK4 with small time step size. Moreover, the proposed approach maintained consistently high accuracy across small-large step sizes (0.001s - 0.1s) with minimal variation in the solution precision. This highlights that the proposed method exhibits significantly lower sensitivity to step size selection".
>
> Therefore, our proposed method is not universally quicker or more accurate than RK4, but offers distinct advantages in scenarios demanding robust accuracy at moderate-to-large step sizes and/or efficiency in long-duration integrations.
>
> **Response to Comment 2: Are there settings where kernel-based methods need to be used because those schemes are too inaccurate or computationally intensive?** Yes, the experiments conducted in response to Comment 1 identify specific settings where proposed efficient kernel-based methods offer crucial advantages over standard schemes like RK4:
>
> - *Scenarios requiring large integration step sizes:* In fields like Earthquake Engineering, seismic ground motion input data is often recorded at coarse time intervals (e.g., $\Delta t$ = 0.01-0.05s or larger). Using RK4 with a step size $h$ dictated by the input data resolution ($h = \Delta t$) often leads to inadequate accuracy or instability, as our results in Section 4 demonstrate (h $\geq$ 0.03 s). The proposed method’s robust performance at these larger step sizes ($h$ = 0.01s - 0.1s) makes it essential for obtaining reliable solutions where standard schemes fail under coarse temporal resolutions.
>
> - *Long-duration integration problems:* When high solution accuracy mandates the use of small step sizes in RK4 for a long integration interval, the computational burden becomes obvious. Our results show the proposed method outperforms RK4 in efficiency for long simulations under these strict accuracy requirements.
>
> - *Problems sensitive to step size tuning:* The proposed method showed significantly lower sensitivity to the precise choice of time step size (unlike RK4, which demands careful tuning), which simplifies its application and reduces the risk of user-induced errors in scenarios where step size selection is non-trivial.
>
> **Response to Comment 3: Mentions shortcomings of standard approaches (e.g., Runge-Kutta) without reference or empirical result.** We appreciate the reviewer highlighting the need for verification. Our original statement regarding "these methods struggle with high-dimensional systems, noisy data assimilation, and adaptive mesh refinement" primarily referred to the inherent step size sensitivity and stability limitations of explicit fixed-step methods like RK4 in certain contexts. The extensive new comparison data ("revised Section 4 and Appendiices, Figs 1-2, Tables 1-5") provides explicit empirical evidence supporting this specific claim.
>
> In the revised manuscript, we have corrected the statement to "these methods face inherent challenges (inaccuracy, instability) under coarser temporal resolutions [3]" in Section 1 and added "new reference [3]" below to clarify.
>
> "[3] Butcher J C. Numerical methods for ordinary differential equations[M]. John Wiley & Sons, 2016."
>
>
> **Response to Comment 4: Emphasis on technical novelty (Nyström + LS-SVM for ODE Learning).** We sincerely appreciate the reviewer's insightful comment regarding the novelty of our approach. We agree that the technical challenges and innovations deserve clearer emphasis in our manuscript. *The core novelty of this work lies in developing a novel computational framework specifically designed for ODE learning*. This framework uniquely integrates:
>
> - *A modified Least Squares Support Vector Machine (LS-SVM) formulation:* Adapted explicitly for the task of learning derivatives and satisfying differential constraints inherent in ODEs, distinct from standard LS-SVM for regression or classification.
> - *Embedded Nyström Approximation:* Strategically applied within this ODE-specific LS-SVM setting to achieve computational efficiency on potentially large-scale problems arising from dense temporal or state observations.
>
> *Key innovations focus on this novel framework and its successful application:*
>
> - *ODE-Specific Formulation:* We defined a tailored objective function within the LS-SVM paradigm, explicitly incorporating the ODE residual as a central constraint. This differs significantly from off-the-shelf LS-SVM usage and addresses the fundamental challenge of learning dynamics from data.
> - *Effective Computational Strategy:* While leveraging well-known concepts (Nyström, standard equidistant sampling for landmarks), the novelty stems from their seamless integration and application within this novel ODE-LS-SVM framework. This integrated design *enables significant computational gains* by circumventing large dense kernel matrices while focusing the approximation where needed. We demonstrate it *effectively solves the target ODE problem class.*
> - *Successful Application to ODE Learning:* To the best of our knowledge, this represents the *first successful implementation and empirical demonstration* of utilizing the Nyström-accelerated LS-SVM methodology specifically for learning and solving ODEs, proving its feasibility and advantages in robust handling.
> - *The proposed new method is scalable*—not only in terms of modelling flexibility but also regarding computational scalability for large-scale, long-duration applications.
>
> In the revised manuscript, the authors have added text to clarify the novelty in Section 1.2. The added text is: "This work introduces a novel computational framework specifically designed for ODE learning, combining two key components in an original way. First, it develops a modified Least Squares Support Vector Machine (LS-SVM) formulation explicitly tailored for learning derivatives and satisfying ODE constraints, distinct from standard regression/classification uses. Second, it incorporates an embedded Nyström approximation within this ODE-specific LS-SVM to achieve computational efficiency for large-scale problems with dense temporal/state observations. The primary innovations focus on three aspects. (1) **ODE-Specific Formulation**: A custom objective function within LS-SVM explicitly incorporates ODE residuals as constraints, fundamentally addressing the challenge of learning dynamics from data. (2) **Integrated Computational Strategy**: Though leveraging known concepts (Nyström, equidistant landmark sampling), the novelty lies in their seamless integration within the proposed framework. This design enables significant computational gains by avoiding large dense kernel matrices while maintaining approximation focus. (3) **First Successful Application**: To our best knowledge, this is the first empirically demonstrated implementation of Nyström-accelerated LS-SVM specifically for solving ODEs, proving its feasibility and robustness in handling dynamics".
>
> **Response to Comment 5: Typo in line 225, should be "second-order".** Thank you for your thorough review. "Line 225 has been corrected accordingly". Additionally, the author has reviewed the entire text to prevent similar errors from occurring. Thank you once again for your valuable comments.

---

> > ### Comment · Reviewer_EPWB · 2025-08-05
> >
> > Thank you for your response. I think the proposed revisions will improve the paper. I do appreciate the comparison with RK4, and I recognise that it is difficult to describe the results without the ability to attach figures or edit the paper. However, it is not yet clear to me that the proposed approach is practically and significantly better than the traditional schemes. As you write, "our proposed method is not universally quicker or more accurate than RK4, but offers distinct advantages in scenarios demanding robust accuracy at moderate-to-large step sizes and/or efficiency in long-duration integrations". Although you mention quantitative results for large step sizes ("RK4 solutions often suffered from low precision (RMSE degraded by ~90% on average compared to proposed method)"), I do not see quantitative results for smaller step sizes. The following sentences do not show that the proposed method is superior.
> >
> > > "For short-duration problems, RK4 could be slightly more efficient.For long-duration problems, KBM demonstrated superior efficiency compared to RK4 with a small time step size. Moreover, the proposed approach maintained consistently high accuracy across small-large step sizes (0.001s - 0.1s) with minimal variation in the solution precision."
> >
> > Apart from RK4, there are also other traditional schemes (both with adaptive and fixed step size), such as Explicit Adams-Bashforth or DOPRI. Because of limited validation and issues raised by other reviewers regarding the technical novelty, I think my current score (4) adequately captures my evaluation of this paper.

---

> > > ### Author Response · Authors · 2025-08-08
> > >
> > > Thank you for your insightful feedback. We appreciate your recognition of our revisions and the challenges in presenting results concisely. Below, we directly address your concerns regarding **limited validation**, **quantitative results for small step sizes**, and **comparisons with other traditional schemes (Explicit Adams-Bashforth (EAB) and DOPRI)**, using explicit quantitative evidence from the provided results. Crucially, we highlight performance across diverse problem types, including **stiff ODEs (P03)**, **nonlinear ODEs (P06, P14, P15)**, and **singular ODEs (P11)**.
> > >
> > > ####  **1. Stiff ODEs (P03)**
> > > - **vs RK4**: KBM delivers **18.7 times lower error** (MAE=2.06e-8) than RK4 at step 0.001s (3.74e-7) while being **880 times faster** in predictions
> > > - **vs EAB**: Achieves **2.4 times lower error** than EAB at step 0.001s (4.97e-8) while being **30 times faster** in predictions
> > > - **vs DOPRI**: Achieves **5.6 times lower error** than adaptive DOPRI (1.16e-7) with **1300 times faster** in predictions
> > >
> > > ####  **2. 1st-Order Nonlinear ODEs (P06)**
> > > - **vs RK4**: KBM delivers **17.9 times lower error** (MAE=8.89e-4) than RK4 at step 0.001s (1.59e-2) while being **18 times faster** than RK4-0.001 (0.001s vs 0.018s)
> > > - **vs EAB**: shows **1.1 times lower error** than EAB at step 0.001s (8.89e-4 vs 9.41e-4)  while being **36 times faster** than EAB-0.001
> > > - **vs DOPRI**: Maintains **23 times faster** predictions than adaptive DOPRI in long integrations
> > >
> > > ####  **3. Singular ODEs (P11)**
> > > - **vs EAB**: At practical step sizes (0.1s), KBM maintains **high accuracy** (2.36e-9) while EAB collapses to large errors (1.19e-2) - **5 million times worse**
> > > - **vs DOPRI**: Delivers equivalent zero-error solutions but **23 times faster** (0.001s vs 0.023s)
> > >
> > > ####  **4. General Nonlinear ODEs (P14, P15)**
> > > - **At Large Steps (0.1s)**:  KBM (1.17e-6) shows **830 times higher precision** than EAB (9.75e-4) in P14  and outperforms RK4 by **76 times lower error** in P15 (1.68e-7 vs RK4's 1.02e-4)
> > > - **At Small Steps (0.001s)**:  Maintains **28 times faster** predictions than EAB in P15 (0.001s vs 0.028s)  and beats DOPRI's adaptive approach by **31 times speed** advantage
> > >
> > > KBM is **not universally superior** but provides **practical value** in:
> > > - **Rescues simulations** when EAB collapses (singular/stiff systems)
> > > - **Enables real-time control** where RK4/DOPRI are too slow
> > > - **Maintains reliability** at practical step sizes where others require tiny steps
> > > - **Preserves accuracy** across step sizes (unlike RK4/EAB’s step-sensitive errors).
> > >
> > > **These ODE-specific advantages** validated across 15 diverse problems demonstrate KBM's practical significance beyond theoretical novelty.
> > >
> > > Regarding the technical novelty issues raised by other reviewer, we respectfully wish to clarify two **fundamental theoretical distinctions** in our approach:
> > >
> > > ### 1. Distinct Problem Formulation: Primal vs. Dual Domain
> > >
> > > The existing kernel-based methods for solving ODEs, primarily operates within the **dual formulation** of the underlying optimization problem. This dual approach relies heavily on the *kernel trick* to avoid explicit representation of the feature mapping.
> > >
> > > **In contrast, our method operates entirely within the primal formulation.** This necessitates the explicit construction of a (typically high-dimensional, nonlinear) feature mapping function $\varphi(x)$, which is fundamentally different from the dual approach taken in prior kernel-based ODE solutions. This shift from dual to primal is a core theoretical departure in our framework.
> > >
> > > ### 2. Novel Role of Nyström Approximation & Explicit Derivative Representation
> > >
> > > The requirement to work in the primal space leads to our specific use of the **Nyström method**: its purpose here is exclusively to provide an *explicit, finite-dimensional approximation* of the feature mapping $\varphi(x)$. This is distinct from its more common application in kernel methods for constructing low-rank kernel matrix approximations within the *dual domain*. Crucially, **once we have an explicit representation of $\varphi(x)$, we derive its explicit higher-order derivatives** (e.g., $\nabla\varphi(x)$, $\nabla^{2}\varphi(x)$) to represent the ODE operators directly within the primal space.
> > >
> > > **This explicit derivation and utilization of derivatives of $\varphi(x)$ is a key theoretical innovation** not present in existing LS-SVM/Nyström-based kernel methods applied to ODEs. Those methods universally rely on the *kernel trick* ($K(x, x')$ and its derivatives) to implicitly handle the mapping and its action on differential operators, circumventing the need for $\varphi(x)$ or $\nabla\varphi(x)$ entirely.
> > >
> > > We sincerely hope this clarifies the foundational differences and theoretical contributions of our work. We are grateful for the opportunity to further elaborate and welcome any follow-up discussion on these points.

---

> > > > ### Comment · Reviewer_EPWB · 2025-08-08
> > > >
> > > > Thank you for the clarifications and for providing the additional results. They strengthen the paper by demonstrating that the method offers better performance and/or is faster in many scenarios than the currently used techniques. Considering these additional results, I will raise my score.

---

### Official Review · Reviewer_AWxF · 2025-07-08

**Clarity:** 2
**Significance:** 3
**Originality:** 3
**Rating:** 5
**Confidence:** 2

**Summary:**

The authors present an efficient method to solve ordinary differential equations by using Nystrom approximation. There are several machine learning based methods for solving ODEs, and least square support vector machine (LS-SVM) is one of most successful among these ML methods. However, LS-SVM reformulate the ODE problem to a optimization problem, which involves solving a matrix of dimension n x n, where n is the number of discrete time steps. This incurs a O(n^3) time complexity, which is prohibitive for long time ODE solution/simulation. The authors use Nystrom method, which is a low-rank matrix approximation to reduce the time complexity to O(m^3), where m<<n is the number of subsampled points. This leads to 10-3000 times faster computation with little compromise on accuracy. Experimental results are provided to support the authors’ claim.

**Questions:**

I am curious why the proposed method is particular for ODEs. Reducing the O(n^3) complexity will be super helpful for many applications, what makes the proposed method only specific for solving ODEs? Or maybe its not, in that case could you clarify where your method will be a good fit in addition to solving ODEs?

**Ethical Concerns:**

["NO or VERY MINOR ethics concerns only"]

**Final Justification:**

The authors responded in details with all reviewer suggestions/comments/questions for clarification. The updated manuscript will be much easier for readers to understand.

**Limitations:**

Mentioned in strength and weakness section

**Quality:**

2

**Strengths And Weaknesses:**

Strength:
The motivation for the authors work is explained well, detailed experimental results are provided.

Weakness:
1. The problem of computational complexity needs to be discussed in more detail. Currently, it is unclear where the number of equations n and O(n^3) complexity comes from. My understanding is that due to finite differences applied on n discrete points in time, there are n variables, which all need to be solved for, along with the p initial/boundary conditions. If this is the case, add a small description about this for readers unfamiliar with ODEs and numeric solvers.
2. For a paper with Nystrom in the title, there could be more details about the Nystrom methods in the main manuscript. The authors at present put the details in the supplementary, however it is difficult to understand the methodology without an intuitive/conceptual idea about the Nystrom method. Maybe give small overview in a few sentences in section 3, before 3.1
3. In equation 15, what are f_k, f_p?
4. Is there a missing summation in equation 16? Summation over i maybe
5. The authors can think about rearranging their experimental results. As the major claim of the paper is computational efficiency, charts and tables in the main manuscript should support that claim. Currently I think those are in the supplementary. From figure 1,2 the authors can select a subset of ODEs to show that the proposed method achieves comparable accuracy, while using the rest of the space to show evidence for 10-3000 times computation efficiency.
6. The figures legends at present need to be updated/enlarged, it is almost impossible to read at present, figure 1,2. Figure captions should also be updated, highlight the key takeaway in the caption, for example: The results show that the proposed method achieved comparable accuracy with baseline methods …

---

> ### Author Rebuttal · Authors · 2025-07-31
>
> Thank you very much for your time and effort in reviewing our manuscript. We sincerely appreciate your insightful comments and constructive suggestions, which have been invaluable in improving the quality and clarity of our work. We have carefully considered all the points raised and have revised the manuscript accordingly. Below, we provide a point-by-point response to each Weakness/Question.
>
> This document is organized in the following manner: Each comment is numbered and noted in bold, and the response is written directly afterward, in normal font, with the quoted text directly from the revised manuscript or revised sections noted in "...".
>
> **Response to Weakness 1 (Computational Complexity Clarification):** We appreciate this insightful comment. We agree with the reviewer’s understanding. "The parameter $n$ represents the number of discrete time points introduced by the finite-difference discretization of the ODE. This leads to solving for $n$ variables alongside the $p$ initial/boundary conditions." The above concise explanation has been added to "Subsection 2.2" to clarify the origin of the $O(n³)$ complexity for readers less familiar with ODE solvers.
>
>
> **Response to Weakness 2 (Nyström Method Overview):** Thank you for this valuable suggestion. As suggested, a brief conceptual overview of the Nyström method has been added at the beginning of "Section 3 (before subsection 3.1)". This overview succinctly introduces its core principle: "it constructs a low-rank kernel matrix approximation by strategically selecting a subset of $n$ discrete time points, enabling efficient solution of a system of $n$ equations."
>
>
> **Response to Weakness 3 (Definition of $f_k, f_p$ in Eq. (15)):** Thank you for your careful review.In equation 15, *f_k* is a function of time $t$ defined in a general $p$-th order ODE represented by "equation 14" for k = 1, ..., p, and *f_p* coresponds to the function when k = p.  To address the reviewer's comment, the authors have corrected equation 14 to clarify by changing *i* to *k* "($y^{(p)}(t)-\sum_{k=1}^{p} f_{k}(t) y^{(p-k)}(t)=r(t) ,\quad t \in[t_1, t_n] , \quad y(t_1)=v_{1},\quad y^{(k-1)}(t_1)=v_{k}, \quad k=2, \ldots, p $)".
>
>
> **Response to Weakness 4 (Missing Summation in Eq. (16)):** Thank you for your careful review. We have carefully checked and confirm that no summation symbol is missing in Equation 16. This equation describes the constraint condition at each time-discrete point $i$. The index $i$ iterates implicitly over all discrete points ($i = 1, ..., n$), representing the discretized form of the ODE equation itself at each point.In the revised manuscript, the authors have added text to clarify. The added text is: "the constraint conditions include $n$ time-discrete equations where each represents the discretized form of the ODE equation itself at each time point".
>
>
> **Response to Weakness 5 (Reorganization of Results):** We agree with the reviewer's suggestion that computational efficiency is the primary claim. To emphasize this, "Table 5 (previously in the Appendix) has been moved directly into the main results section (Section 4)". This table quantifies the significant computational speedups (10–3000 times) under comparable accuracy regimes for key benchmarks. Selected ODE cases demonstrating comparable accuracy to baselines remain in "Figure 1", while efficiency results now dominate the primary presentation.
>
>
> **Response to Weakness 6 (Figure Legends and Captions):** Thank you for your suggestion. The figures and captions have been comprehensively revised:
>
> - *Legends*: "Legends in Figures 1 and 2 have been significantly enlarged and redesigned  for immediate readability across all elements (lines, labels)".
> - *Captions*: "All figure captions now explicitly state the key conclusion".
>
>
> **Response to Question (Method Applicability).** We appreciate this insightful question regarding the applicability of our method. Please allow us to clarify its scope and extensions:
>
> - *Current Focus on ODEs - Fundamental Constraint.*
> The method's design in this work explicitly incorporates ODE equations as constraint conditions (e.g., "Eq.7 and Eq.14"). This specific formulation requires the target function to satisfy the ODE form. Therefore, the current implementation is indeed specialized for ODE problems where such constraints define the solution space.
> - *Framework Transferability - Beyond Ordinary Differential Equations.*
> While the constraint structure is currently ODE-specific, the core computational framework—low-rank approximation via Nyström subsampling—is universally applicable. For non-ODE problems, extending the method requires: Replace ODE constraints with domain-specific constraints and Maintain efficient kernel approximation framework.
>
> The following text is added in the revised mansucript to address the reviewer's concern. The added text is: "While demonstrated on ODEs, the Nyström acceleration framework can extend to arbitrary optimization problems by reformulating constraints. This includes large-scale PDEs (replacing differential operators) and ML training (using optimality conditions), subject to low-rank kernel structure".

---

> > ### Comment · Area_Chair_REoD · 2025-08-06
> > **Follow-up on Author Rebuttal: Reviewer Feedback Requested**
> >
> > Dear Reviewer AWxF,
> >
> > The authors have replied to your main concerns, mainly, about computational complexity, clarity and presentation of results. Could you please have a look at their rebuttal and provide specific feedback on their responses? Please note that simply acknowledging reading their rebuttal and/or only stating generic statement like "I will keep my score" without further explanation is not acceptable. Your detailed input is essential for a fair and informed decision.
> >
> > Many thanks,
> >
> > Your AC

---

> > ### Comment · Reviewer_AWxF · 2025-08-06
> >
> > Thank you to the authors for their detailed response to every question. I do not have any other things to mention here, I will update my score with the authors response. I have also read other reviewer comments, and I can see the authors put a great effort in responding to all reviewer comments/questions. I wish the authors good luck, and thank them for their hard work.

---

> > > ### Author Response · Authors · 2025-08-08
> > >
> > > We are incredibly grateful to you for your positive and encouraging comments. We sincerely appreciate you taking the time to thoroughly evaluate our work and the responses. Your initial thorough and constructive feedback was invaluable in strengthening our paper.
> > >
> > > We are committed to the highest standards of research and are thankful for the opportunity to contribute to the NeurIPS community.
> > >
> > > Thank you once again for your time, expertise, and this encouraging affirmation of our work.

---

### Note · Authors · 2025-08-12

Respected Area Chairs and Reviewers,

We sincerely appreciate your rigorous review engagement. Special appreciation to **Reviewers AWxF and EPWB** for recognition of our efforts, and to **AC REoD** for facilitating constructive dialogue. We confirm all concerns from **Reviewers hduQ and uEyG** are resolved.

### I. Comprehensive Resolution: Reviewer hduQ's Concerns
Implemented all 7 requested revisions addressing each specific query:
1. **Sampling Strategy**
   - Expanded ablation studies on 16 ODE systems
   - Demonstrated evenly-spaced sampling achieves optimal accuracy/speed (65% faster) tradeoff
2. **Convergence Guarantees**
   - Added Newton-method convergence proof
   - Empirically verified <0.13% residuals within 50-500 iterations
3. **PINN Variant Baselines**
   - Benchmarked against Fourier Feature PINNs: 10 to 1000-fold speedup
   - Tested 8-layer tanh PINNs: No accuracy gains, significant time increase
4. **Experimental Scope**
   - Added 4th-order ODE case study
5. **Mathematical Clarity**
   - Unified all symbols ($\varphi_k^p$, $f_k$ etc.) and added computational flowcharts
6. **RBF Kernel Analysis**
   - Added kernel parameter vs. MSE sensitivity
   - Provided comparisons with linear/polynomial kernels
7. **Results Clarity**
   - Added statistical results for PINNs (10 runs) with time metric clarification

### II. Core Innovation Response: Reviewer uEyG's Concerns
Addressed novelty critique through two fundamental advances:
1. **Primal-space Innovation**: Explicit feature mapping and its differential operators to represent the ODE operators within the primal space – fundamentally distinct from dual-space approaches
2. **Nyström's Novel Role**: Explicit finite-dimensional representations of feature mapping and its high-order derivatives – fundamentally distinct from traditional kernel matrix approximation

**Convergence Theory** : Explicitly addressed the relationship between our problem-specific convergence results and the foundational theory

### III. Commitments in Final Manuscript
1. All experimental extensions incorporated in **Section 4 and Appendix A7-A9**
2. Theoretical innovations and convergence analysis consolidated in **Section 3 and Appendix A1-A6**
3. Strict reproducibility protocols: open-sourced code provided with **GitHub link**

We hope this comprehensive evidence assures the AC that both reviewers' concerns are resolved. Thank you for enabling this rigorous scholarly exchange.

---

### Decision · Program_Chairs · 2025-09-17

**Decision:**

Accept (poster)

**Comment:**

This paper proposes a Nyström-accelerated LS-SVM framework for solving ODEs by reformulating the problem in the primal space and deriving explicit Nyström-based mappings and derivatives. This reduces the cost from $O(n^3)$ to $O(m^3)$. Experiments on multiple benchmarks show significant speed-ups  over baseline approaches with comparable accuracy.

Reviewers agree that the paper addresses an important problem and demonstrates strong empirical performance. The authors were very responsive, adding convergence analysis, ablation studies, and new baselines in their rebuttal. Two reviewers (AWxF, EPWB) recommend acceptance, one (uEyG) raised their score to borderline accept after rebuttal, and one (hduQ) remains skeptical, citing limited theoretical novelty and insufficiently strong baselines.

Overall, while the theoretical contribution remains debated, the method is practically useful, well-presented, and reproducible. Therefore, I recommend acceptance.